# Astrocytes enable amygdala neural representations supporting memory

Olena Bukalo[1✉], Ruairi O'Sullivan[1], Yuta Tanisumi[2], Adriana Mendez[1], Chase Weinholtz[1], Sydney Zimmerman[1], Victoria Offenberg[1], Olivia Carpenter[1], Hrishikesh Bhagwat[1], Sophie Mosley[1], John J. O'Malley[3], Kerri Lyons[4], Yulan Fang[5,6,7], Jess Goldschlager[1], Linnaea E. Ostroff[5,6,7], Mario A. Penzo[3], Hiroaki Wake[2], Lindsay R. Halladay[4,8] & Andrew Holmes[1✉]

Brain systems mediating responses to previously encountered threats provide critical survival functions. Fear memory and extinction are underpinned by neural representations in the basolateral amygdala (BLA)[1–7], but the contribution of non-neuronal cells, including astrocytes, to these processes remains unresolved. Here, using in vivo calcium ($Ca^{2+}$) imaging and causal astrocyte manipulations, we find that BLA astrocytes dynamically track fear state and support fear memory retrieval and extinction. By combining astrocyte manipulations with in vivo BLA neuronal $Ca^{2+}$ imaging and electrophysiological recordings, we show that astrocyte $Ca^{2+}$ signalling enables neuronal encoding of fear memory retrieval and extinction, and readout through a BLA–prefrontal circuit. Our findings reveal a key role for astrocytes in the generation and adaptation of fear-state-related neural representations, revising neurocentric models of critical amygdala-mediated adaptive functions.

Environmental stimuli experienced during threatening events become associated with danger. On subsequent encounters with these stimuli, a fear memory is retrieved through engagement of dedicated brain circuitry that generates internal fear states and mobilizes defensive behaviours[1,2]. In regions including the BLA, stimulus-elicited fear states are neurally represented and readout by anatomically defined neuronal subpopulations to downstream effector sites[3–8]. However, it is unclear whether non-neuronal BLA cell types enable memory-related representations and the dynamic representational adaptations that occur as memories undergo extinction.

Astrocytes are a highly abundant type of glial cell that make extensive contacts with neurons[9]. Although relatively quiescent under basal conditions, astrocytes exhibit complex intracellular $Ca^{2+}$ dynamics, respond to neurotransmitters and modulators, and regulate transmission through the release of $Ca^{2+}$-evoked gliotransmitters[10–17]. There is growing evidence that astrocytes interact with neurons through dynamic fluctuations in $Ca^{2+}$ activity to encode sensory information, gate state transitions and mediate experience-dependent behavioural adaptations, including those related to fear[18–36]. These findings implicate astrocytes in fear, but there remains a critical gap in our understanding of how astrocytes might support fear memory-related neuronal coding and plasticity through modulation of BLA neuronal representations and behaviourally mediating circuit outputs.

## Astrocytes track memory retrieval and extinction

We first monitored BLA astrocyte activity as mice underwent a cued fear memory paradigm by virally expressing a $Ca^{2+}$ indicator (cyto-GCaMP6f) in the astrocyte cytosol and performing in vivo fibre photometry (Fig. 1a). Immunostaining for astrocytic and neuronal markers confirmed selective GfaABC1D-cyto-GCaMP6f expression in astrocytes (around 2% neuronal expression) (Fig. 1b; details of all statistical tests and results are provided in Supplementary Table 1). Fear conditioning (F-Con) involved three co-presentations of a tone-conditioned stimulus (CS) and footshock unconditioned stimulus (US) and was followed by 2 days of repeated (25×) CS-alone extinction sessions in a novel context (Ext1, Ext2) (Fig. 1c). CS-elicited freezing was high during early extinction (E-Ext, first extinction trial block), that is, during fear memory retrieval, then decreased significantly by late extinction (L-Ext, final extinction trial block) and retrieval (E-Ret), before returning to the pre-extinction levels on retesting in the conditioning context (fear renewal, F-Ren) (Fig. 1d).

Photometry revealed robust $Ca^{2+}$ responses to US, and not CS, presentation during F-Con, and then to CS presentation during E-Ext (Fig. 1e–g). Paralleling changes in freezing across test phases, CS-related astrocyte $Ca^{2+}$ activity (area under the curve (AUC) and transient frequency) decreased with extinction (that is, L-Ext, E-Ret) and recovered on F-Ren (Fig. 1g,h, Extended Data Fig. 1a–d and Supplementary Video 1). Test-stage-wise changes in astrocyte $Ca^{2+}$ activity reflected overall freezing levels during CS presentation (Fig. 1i), rather than episodes of inactivity per se—given astrocyte $Ca^{2+}$ increased at both movement onset and offset[23,27,37–39] during E-Ext (Extended Data Fig. 1g). Thus, these data show that amygdala astrocyte $Ca^{2+}$ activity tracked learned changes in the CS-related fear state.

Comparable photometry data were obtained by measuring $Ca^{2+}$ in astrocyte processes using the membrane-targeted $Ca^{2+}$

[1]Laboratory of Behavioral and Genomic Neuroscience, National Institute on Alcohol Abuse and Alcoholism, NIH, Bethesda, MD, USA. [2]Division of Multicellular Circuit Dynamics, National Institute for Physiological Sciences, National Institutes of Natural Sciences, Okazaki, Japan. [3]Unit on the Neurobiology of Affective Memory, National Institute of Mental Health, NIH, Bethesda, MD, USA. [4]Department of Psychology, Santa Clara University, Santa Clara, CA, USA. [5]Department of Physiology and Neurobiology, University of Connecticut, Storrs, CT, USA. [6]Connecticut Institute for the Brain and Cognitive Sciences, University of Connecticut, Storrs, CT, USA. [7]Institute of Materials Science, University of Connecticut, Storrs, CT, USA. [8]Department of Neuroscience, University of Arizona, Tucson, AZ, USA. ✉e-mail: Olena.Bukalo@nih.gov; Andrew.Holmes@nih.gov

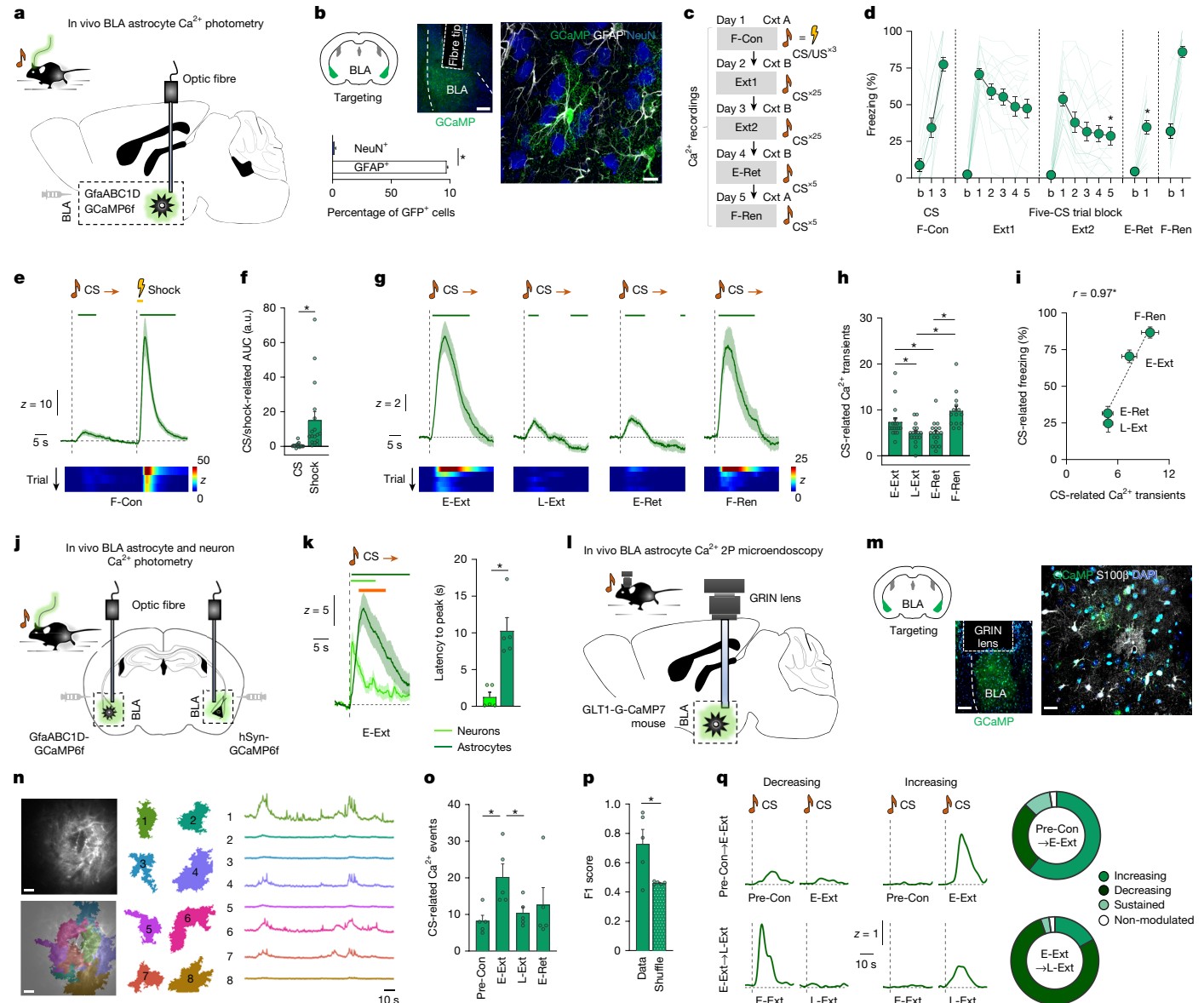

**Fig. 1 | Astrocytes track memory retrieval and extinction. a,** In vivo Ca²⁺ fibre photometry. **b,** Example fibre placement and GCaMP expression in astrocytes (GFAP) and neurons (NeuN) (two-tailed paired *t*-test, *$P < 0.0001$). $n = 16$ sections, 6 mice. Scale bars, 200 μm (left) and 10 μm (right). **c,** Experimental timeline. Cxt, context. **d,** Freezing across testing phases (one-way repeated measures analysis of variance (ANOVA) with Šidák's post hoc test, *$P < 0.0001$, comparing E-Ext (Ext1 block 1) versus L-Ext (Ext2 block 5), and E-Ext versus E-Ret). $n = 17$ mice. **e,** Population trace of CS and shock-related Ca²⁺ activity during F-Con with three trial-wise heat maps shown below. $n = 17$ mice. **f,** AUC analysis of shock-related Ca²⁺ activity during F-Con (two-tailed paired *t*-test versus CS, *$P = 0.0080$). $n = 17$ mice. a.u., arbitrary units. **g,** Population trace of CS-related Ca²⁺ activity during E-Ext, L-Ext, E-Ret and F-Ren; five trial-wise heat maps are shown below. $n = 14$–17 mice per stage. **h,** CS-related Ca²⁺ activity across test phases (one-way repeated measures ANOVA and two-tailed unpaired *t*-tests, comparing E-Ext versus L-Ext (*$P = 0.0231$), E-Ext versus E-Ret (*$P = 0.0262$), L-Ext versus F-Ren (*$P < 0.0001$), E-Ret versus F-Ren (*$P = 0.0002$)). $n = 14$–17 mice per stage. **i,** The relationship between CS-related Ca²⁺ activity and freezing across test sessions (two-tailed Pearson's correlation, *$P = 0.0261$). $n = 14$–17 mice per stage. **j,** Simultaneous in vivo Ca²⁺ fibre photometry recordings of astrocytes (cyto-GCaMP6f) and neurons (hSyn-GCaMP6f) in opposite hemispheres of the same mice. **k,** The latency to peak CS-related Ca²⁺ activity in neurons and astrocytes during E-Ext (two-tailed unpaired *t*-test, *$P = 0.0021$). $n = 5$ mice per group. **l,** In vivo two-photon (2P) Ca²⁺ imaging. **m,** Example GRIN lens placement (left) and GCaMP expression in astrocytes (S100β) (right). Scale bars, 200 μm (left) and 20 μm (right). **n,** Example raw and AQuA[38] processed images (left). Scale bars, 50 μm. Right, example astrocyte events and associated spontaneous Ca²⁺ activity. **o,** The number of Ca²⁺ events per CS across test phases (one-way repeated-measures ANOVA, Pre-Con versus E-Ext (two-tailed unpaired *t*-test, *$P = 0.0154$), E-Ext versus L-Ext (two-tailed unpaired *t*-test, *$P = 0.0376$)). $n = 257$ events, 6 mice. **p,** Decoding CS presentation from astrocyte Ca²⁺ activity (one-tailed paired *t*-test, *$P = 0.0265$). $n = 110$ events, 6 mice. **q,** Example CS-related Ca²⁺ traces illustrating changes induced by fear and extinction learning and quantification of different subpopulation of Ca²⁺ transients changed by extinction. $n = 80$–94 events, 6 mice. Fluorescence values were *z*-scored to the 5-s pre-CS period (photometry) or the entire recording session (two-photon imaging). The horizontal lines above the traces in **e**, **g** and **k** denote the permutation-test-determined significant difference from chance for astrocytes (dark green), neurons (light green), or between astrocytes and neurons (orange). Data are mean ± s.e.m.

indicator lck-GCaMP6f (Extended Data Fig. 1h–l), but there was minimal (cyto-GCaMP6f) CS-related activity during E-Ext when either the US or CS was omitted during F-Con, and no trial-wise decrease in Ca²⁺ activity in non-extinguished mice (Extended Data Fig. 2a–k). Lastly, simultaneous photometry performed in BLA astrocytes and neurons in different hemispheres of the same conditioned and extinguished

animals indicated generally similar test-stage-related changes in Ca²⁺ activity, but with comparatively slower-to-peak and longer-lasting US- and CS-related Ca²⁺ responses in astrocytes than in neurons (Fig. 1j,k and Extended Data Fig. 3a–n).

To gain insight into memory-related astrocyte Ca²⁺ dynamics at the level of individual astrocytic events, we next performed in vivo multiphoton Ca²⁺ imaging of BLA astrocytes of Glt1-G-CaMP7 mice[40] through a chronically implanted gradient index (GRIN) lens (Fig. 1l–q and Extended Data Fig. 4a–c). Imaging data that were processed to isolate discrete activity-related astrocytic Ca²⁺ events[38] in head-fixed animals undergoing fear and extinction testing indicated an overall pattern of changes in Ca²⁺ activity (AUC and transient frequency) across test stages that resembled our photometry data, including elevated CS-related activity during E-Ext (relative to pre-conditioning, Pre-Con) that diminished during L-Ext and E-Ret (Fig. 1o, Extended Data Fig. 4d–h and Supplementary Video 2). Moreover, linear classifiers trained on Ca²⁺ activity from individual astrocyte events during E-Ext could successfully decode CS presentation from unseen data—consistent with the instantiation of CS-related information in BLA astrocytes (Fig. 1p). Moreover, this imaging approach showed that not all astrocyte events developed conditional CS-related-activity from Pre-Con to E-Ext and, furthermore, revealed subsets of events exhibiting either decreasing, increasing, sustained or absent CS-responsivity across extinction (E-Ext–L-Ext) (Fig. 1q and Extended Data Fig. 4i–m). These findings, together with recent evidence of experience-dependent changes within specific astrocyte ensembles in the BLA and hippocampus[26,31,41], suggest that there are heterogeneous subpopulations of astrocytes that acquire and update memory-relevant information.

Overall, these initial imaging and photometry data show amygdala astrocyte Ca²⁺ activity dynamically tracks changes in fear state associated with the retrieval and extinction of fear memory.

## Astrocytes bidirectionally alter memory

Our data so far imply that astrocyte Ca²⁺ dynamics support fear memory retrieval and extinction. To causally test this, we depleted Ca²⁺ by virally expressing the genetically modified plasma membrane Ca²⁺ extruder (hPMCA2w/b, CalEx) in BLA astrocytes. Mice expressing CalEx showed less CS-elicited freezing compared with the viral controls during E-Ext (as in previous work[26]) and on E-Ret, indicating that this chronic disruption of astrocyte Ca²⁺ activity produced a sustained reduction in fear (Extended Data Fig. 5a–l). This led us to manipulate Ca²⁺ activity with greater temporal specificity using a chemogenetic approach, whereby hM3Dq-coupled designer receptors exclusively activated by designer drugs (DREADDs) were virally expressed in BLA astrocytes (Fig. 2a,b).

Gq G-protein-coupled receptor (GPCR) signalling increases astrocyte Ca²⁺ activity through IP3-mediated release of intracellular Ca²⁺ stores[42,43] and hM3Dq actuation causes a Ca²⁺ surge preceded by prolonged quiescence, possibly due to intracellular Ca²⁺ depletion[24,44,45]. Replicating these effects in the BLA, we expressed hM3Dq in BLA astrocytes and used in vivo cyto-GCaMP6f photometry and observed that clozapine-N-oxide (CNO) injection markedly increased Ca²⁺ activity within around 10 min but, thereafter, decreased and remained low for at least 2 h (Fig. 2c and Extended Data Figs. 6a–e and 8e,f). A lower hM3Dq virus concentration or lower CNO dose had modest or negligible effects on Ca²⁺ activity and behaviour (Extended Data Fig. 6h–p). On the basis of these data, we posited that BLA astrocyte Ca²⁺ dynamics would be constrained by hM3Dq actuation at timepoints relevant to behavioural testing. Consistent with this supposition, hM3Dq-actuation essentially abolished Ca²⁺ responses to a potent stimulus (footshock) given 30 min after CNO injection (Extended Data Fig. 6f,g).

We leveraged these effects of hM3Dq actuation to test how constraining astrocyte Ca²⁺ dynamics affected memory acquisition, retrieval, consolidation and extinction by injecting separate groups of animals with 3 mg per kg CNO either before or immediately after F-Con, or

before fear retrieval/extinction training. We found that CNO given before extinction training reduced CS-related freezing during E-Ext—consistent with impaired memory retrieval—in hM3Dq-expressing mice compared with viral controls (Fig. 2d,e). In vivo fibre photometry confirmed that this behavioural effect was accompanied by loss of CS-related astrocyte Ca²⁺ responses (Fig. 2f and Extended Data Fig. 7a–c). In contrast to these memory-retrieval-impairing effects, CNO had no behavioural effect when injected before or after F-Con[26,27] and did not alter uncued freezing, shock-induced flinching or various measures of anxiety-like behaviour (Extended Data Fig. 7d–i). Behavioural effects were also absent when CNO was injected in mice not expressing hM3Dq or when vehicle was injected in hM3Dq-expressing animals, excluding potential non-specific CNO and hM3Dq-virus effects, respectively (Extended Data Fig. 7j–n).

We next compared these effects with those of another DREADD, hM4Di, that produces effects on cortical, striatal and (as we show here; Fig. 2g–i) BLA astrocyte Ca²⁺ activity that mirror those of hM3Dq, that is, increase Ca²⁺ transients[24,46,47]. Accordingly, we found that hM4Di actuation produced effects on memory retrieval that were opposite to hM3Dq: pre-Ext CNO injection produced increases in CS-related freezing and astrocyte Ca²⁺ responses during E-Ext in hM4Di-expressing mice compared with viral controls (Fig. 2j–l and Extended Data Fig. 8a–f). Pre-Ext hM4Di actuation also increased freezing during (CNO-free) E-Ret, indicative of a deficit in extinction memory formation, and attenuated CS-related Ca²⁺ activity during this test stage. This latter effect is notable given that hM3Dq actuation produced a similar extinction deficit and blunted the CS-related Ca²⁺ response on E-Ret (Fig. 2e and Extended Data Fig. 7b), despite the two manipulations having opposite effects on fear retrieval and neither affecting extinction memory when CNO was given before E-Ret (Extended Data Fig. 8g,h). This convergence of extinction-impairing effects suggests that extinction is sensitive to perturbations—whether increases or decreases—in astrocyte Ca²⁺ activity and, by extension, implies an important role for BLA astrocytes in the plastic adaptations underlying extinction memory formation.

Together, these data demonstrate that manipulating amygdala astrocyte Ca²⁺ activity bidirectionally alters fear memory retrieval and disrupts extinction.

## Astrocytes support neuronal memory representation

Astrocytes are integrated within neuronal networks through synaptic contacts[26,31,48,49] (including in the BLA; Extended Data Fig. 9a,b) and synaptic transmission at glutamatergic BLA neurons is required for both fear memory retrieval and extinction (Extended Data Fig. 10a–g). This suggests that astrocytes could support memory and extinction by modulating neuronal coding in the BLA. In a series of in vivo fibre photometry and in vitro electrophysiological experiments, we found that astrocyte hM3Dq actuation did not alter spontaneous neuronal activity in BLA-containing slices or bulk CS-related BLA neuronal Ca²⁺ activity, indicating that this manipulation did not cause substantial alterations in neuronal activity (Extended Data Fig. 11a–l).

To more directly examine the influence of astrocyte Ca²⁺ signalling on neural representation of memory-related fear states[3–7], we combined hM3Dq actuation in BLA astrocytes with freely moving in vivo one-photon cellular-resolution BLA neuronal (GCaMP7f) Ca²⁺ imaging, performed through a chronically implanted GRIN lens (Fig. 3a–c and Extended Data Fig. 12a,b,d). First, we tested whether chemogenetically disrupting astrocyte Ca²⁺ activity (through CNO injection before fear memory retrieval (F-Ret)) affected changes in CS-related neuronal activity occurring as a result of conditioning. Defining neurons on the basis of whether they exhibited an increase (conditionally CS excited), decrease or no change in CS responsivity from Pre-Con to F-Ret, we found that hM3Dq actuation reduced the number of conditionally CS-excited neurons on F-Ret compared with the vehicle-injected hM3Dq-expressing controls (Fig. 3d,e). These data are consistent with a role for astrocytes

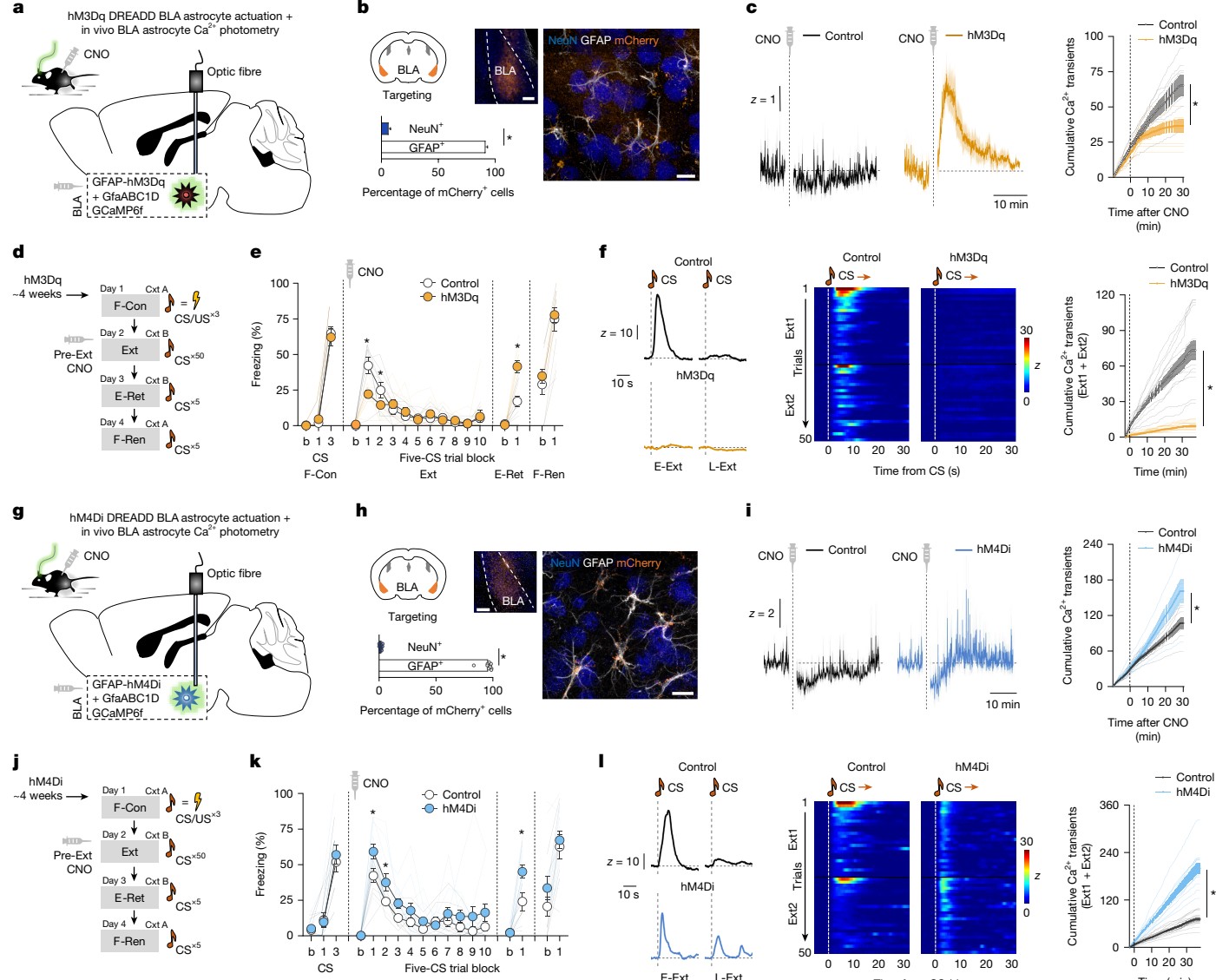

**Fig. 2 | Astrocytes bidirectionally modulate memory retrieval. a**, Combined in vivo Ca²⁺ fibre photometry and astrocyte hM3Dq actuation. **b**, hM3Dq-mCherry expression in astrocytes (GFAP) and neurons (NeuN) (two-tailed paired *t*-test, \**P* < 0.0001). *n* = 12 sections, *n* = 4 mice. Scale bars, 200 µm (left) and 10 µm (right). **c**, CNO effects on neutral-cage Ca²⁺ activity (two-tailed unpaired *t*-tests at minute 29, \**P* = 0.004). *n* = 7 (control) and *n* = 9 (hM3Dq) mice. **d**, Experimental timeline. **e**, CNO effects on CS-related freezing (two-way repeated-measures ANOVA, with Šídák's or Fisher's least significant difference (LSD) post hoc tests, control versus hM3Dq for Ext block 1 (\**P* < 0.0001), Ext block 2 (\**P* = 0.0067) and E-Ret (\**P* < 0.0001)). *n* = 9 (control) and *n* = 11 (hM3Dq) mice. **f**, The effects of CNO on CS-related Ca²⁺ activity during extinction (two-tailed unpaired *t*-test at minute 35, \**P* < 0.0001). *n* = 7 (control) and *n* = 9 (hM3Dq) mice. **g**, Combined in vivo Ca²⁺ fibre photometry and astrocyte hM4Di

actuation. **h**, hM4Di-mCherry expression in astrocytes (GFAP) and neurons (NeuN) (two-tailed paired *t*-test, \**P* < 0.0001). *n* = 10 sections, *n* = 3 mice. Scale bars, 200 µm (left) and 10 µm (right). **i**, CNO effects on neutral-cage Ca²⁺ activity (two-tailed unpaired *t*-test at minute 29, \**P* = 0.0338). *n* = 8 (control), *n* = 6 (hM4Di) mice. **j**, Experimental timeline. **k**, CNO effects on freezing (two-way repeated-measures ANOVA, with Šídák's or Fisher's LSD post hoc tests, comparing control versus hM4Di for Ext block 1 (\**P* = 0.0048), Ext block 2 (\**P* = 0.0250) and E-Ret (\**P* = 0.0005)). *n* = 9 (control) and *n* = 12 (hM4Di) mice. **l**, CNO effects on CS-related Ca²⁺ activity during extinction (two-tailed unpaired *t*-test at minute 35, \**P* < 0.0001). *n* = 8 (control), *n* = 6 (hM4Di) mice. Controls expressed mCherry (**e** and **k**) and mCherry + GfaABC1D GCaMP6f (**c**, **f**, **i** and **l**). Fluorescence values were *z*-scored to the 30-min pre-CNO, 3-min pre-extinction or 5-s pre-CS periods. Data are mean ± s.e.m.

in enabling BLA neurons to express conditioning-related changes in CS-related activity.

To examine this role further, we inspected peri-CS activity on F-Ret to identify neuronal subpopulations that were positively or negatively CS modulated (or non-responsive) (Fig. 3f, g). Although the activity and spatial distribution of CS-modulated neurons was statistically similar in the CNO and vehicle groups, CNO-injected mice had fewer positively modulated neurons (Fig. 3h,i and Extended Data Fig. 12c). Furthermore, decomposing CS-related neuronal activity using principal component analysis (PCA), indicated that more variance was

explained by the top principal components (PCs) in CNO-injected mice compared with vehicle-injected mice and, moreover, that peri-CS neuronal trajectories for PC1 were shorter in the CNO group (Fig. 3j–l). These findings suggest CS-related BLA neuronal activity resided in a lower-dimensional state-space and was less dynamic when astrocyte Ca²⁺ signalling was disrupted by hM3Dq actuation.

We next tested how disrupting astrocyte Ca²⁺ activity affected the signature of BLA neuronal activity during F-Ret. We examined whether a population decoder trained on measures of CS-related neuronal activity could predict, from unseen data, whether hM3Dq-expressing mice

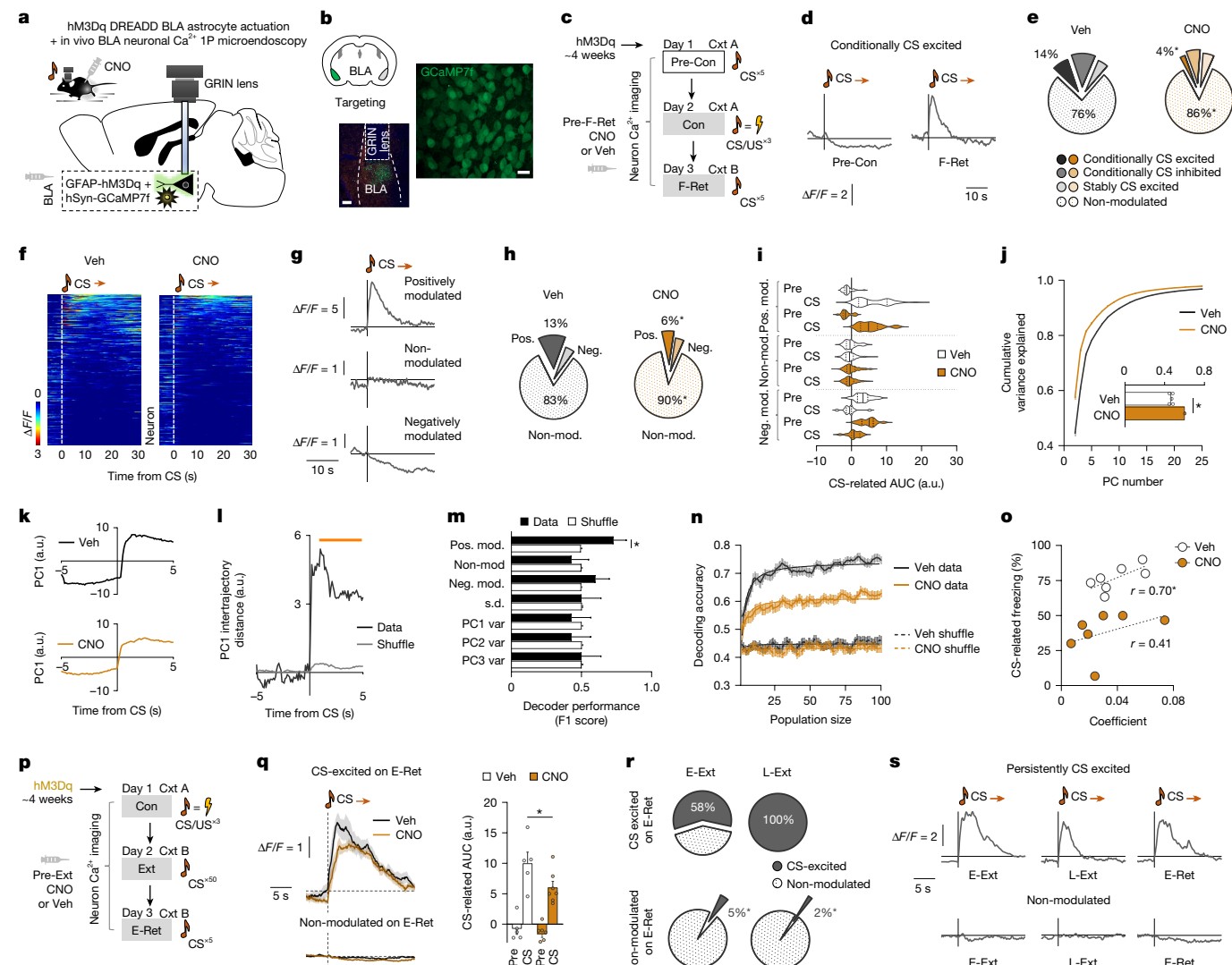

**Fig. 3 | Astrocytes enable neuronal fear memory representation.**
**a**, Combined in vivo one-photon (1P) neuronal Ca²⁺ imaging and astrocyte hM3Dq actuation. **b**, Ex vivo confocal image of hM3Dq-mCherry expression in astrocytes (GFAP) and neurons (NeuN). Lens placement and hSyn-GCaMP7f expression. Scale bars, 200 μm (left) and 10 μm (right). **c**, Experimental timeline. Veh, vehicle. **d**, Example of Ca²⁺ activity in a conditionally CS-excited neuron on Pre-Con and F-Ret. **e**, The cross-stage percentage change of conditionally CS excited, inhibited, sustained and non-responsive neurons (two-sided $\chi^2$ tests, comparing vehicle versus CNO conditionally CS-excited neurons (*$P = 0.0231$) and non-responsive neurons (*$P = 0.0169$)). $n = 182$ neurons, 8 mice (vehicle); and $n = 170$ neurons, 7 mice (CNO). **f**, CS-aligned neuronal Ca²⁺ activity in the vehicle or CNO group. For **f**–**l**, $n = 323$ neurons, 8 mice (vehicle); and $n = 342$ neurons, 7 mice (CNO). **g**, Examples of CS-related neuronal Ca²⁺ activity on F-Ret. **h**, The percentage of CS-modulated neurons (two-sided $\chi^2$ test, comparing vehicle versus CNO conditionally positively modulated neurons (*$P = 0.0015$) and non-modulated neurons (*$P = 0.0053$)). Mod, modulated; neg., negatively; pos., positively. **i**, The effects of CNO on CS-modulated neuronal responses. **j**, The variance (var) explained by PCs. Inset: PC1 variance. Two-tailed unpaired $t$-test, *$P < 0.0001$. **k**, PC1 CS-related neuronal trajectories. **l**, The difference between PC1 CS-related neuronal

trajectories (Benjamini–Hochberg-corrected permutation test, *$P < 0.05$).
**m**, Decoding the treatment group using features of neuronal Ca²⁺ activity (two-tailed paired $t$-test, *$P = 0.0264$). $n = 7$ (vehicle) and $n = 7$ (CNO) mice.
**n**, Decoding CS presentation as a function of neuronal ensemble size (data at each ensemble size are the average decoder performance obtained after randomly selecting neuron subsamples 50×). $n = 7$ (vehicle) and $n = 7$ (CNO) mice. **o**, The relationship between CS-related freezing levels on F-Ret and mouse-averaged decoder coefficient values (one-tailed Pearson's correlation, *$P = 0.039$). $n = 214$ neurons, $n = 7$ mice (vehicle); $n = 227$ neurons, $n = 7$ mice (CNO). **p**, Experimental timeline. **q**, Ca²⁺ activity in CS-excited and non-modulated neurons during (CNO-free) E-Ret after CNO treatment during fear retrieval/extinction training (one-tailed unpaired $t$-test, *$P = 0.034$). For **q** and **r**, $n = 150$ neurons, 7 mice (vehicle); and $n = 126$ neurons, 7 mice (CNO). **r**, The percentage of E-Ret CS-excited and E-Ret non-modulated neurons that were also CS-excited (or non-modulated) on E-Ext and L-Ext (two-sided $\chi^2$ tests, *$P < 0.0001$). **s**, Examples of persistently CS-excited and non-modulated neurons across test stages. The orange horizontal line above the trace in **l** denotes the significant difference between data and shuffle, as determined using a permutation test. Fluorescence values were $z$-scored to the entire recording session. Data are mean ± s.e.m.

had received CNO or vehicle. We found that the treatment group could indeed be successfully decoded from neuronal activity of positively CS-modulated neurons (Fig. 3m). We next tested whether hM3Dq actuation affected the size of the CS-encoding neuronal ensemble by training a population-decoder to distinguish CS from non-CS-related neuronal activity while increasing (from 2 to 100) the number of

randomly selected neurons available to the decoder. We found that, although the decoder performance improved with population size in both treatment groups, CNO-injected mice never attained the levels of the vehicle group (Fig. 3n). Indeed, using a saturating function to calculate the rate at which decoding accuracy increased with the addition of each neuron[50], we estimated that the CS-related information

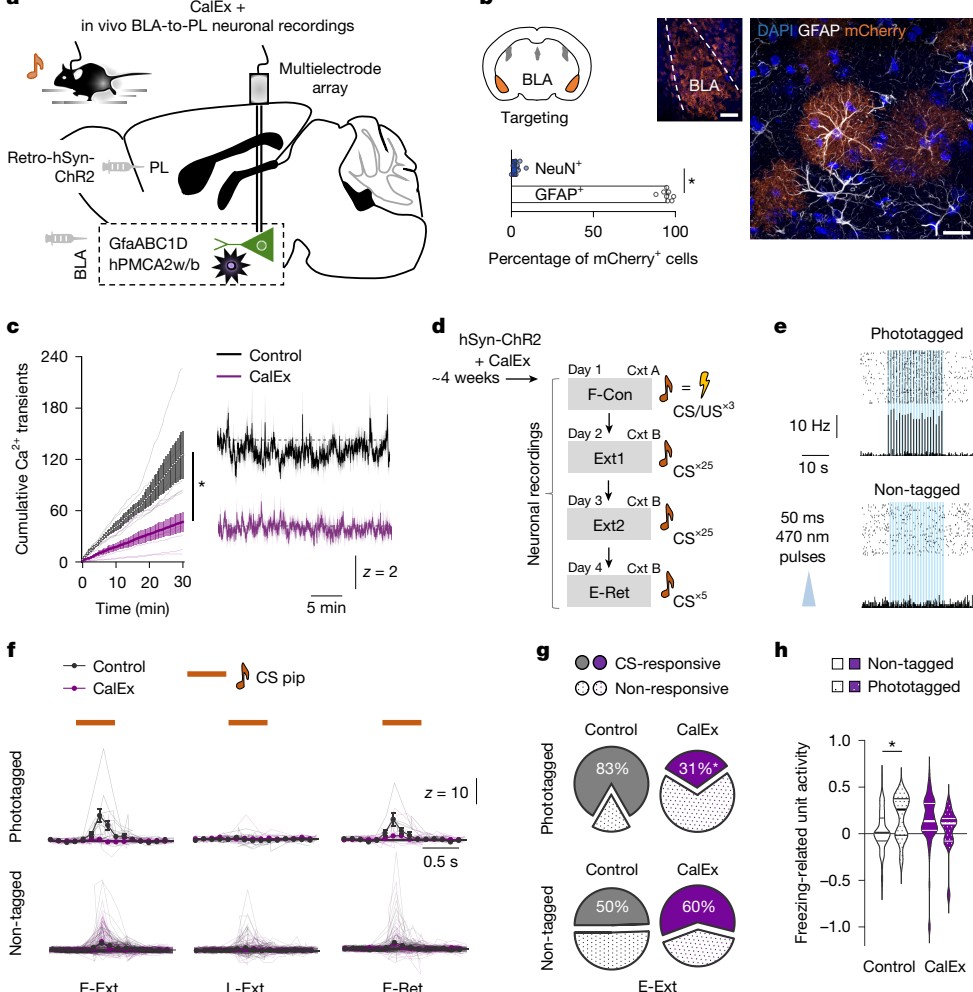

**Fig. 4 | Astrocytes gate memory performance and readout to cortex.**
**a**, CalEx-mediated astrocyte Ca²⁺ depletion and optically targeted in vivo BLA to PL neuronal recordings. **b**, CalEx expression in astrocytes (GFAP) and neurons (NeuN) (two-tailed paired *t*-test, *$P$ < 0.0001). $n$ = 8 sections, $n$ = 4 mice. Scale bars, 200 μm (left) and 10 μm (right). **c**, CalEx effects on neutral-cage Ca²⁺ activity (two-tailed unpaired *t*-test at minute 30, *$P$ = 0.0104). $n$ = 5 (control) and $n$ = 8 (hM3Dq) mice. **d**, Experimental timeline. **e**, Light-evoked BLA-to-PL neuronal responses. **f**, CS-related activity in BLA-to-PL-phototagged ($n$ = 7–12 neurons, 6 mice (control); and $n$ = 14–18 neurons, 4 mice (CalEx)) and non-tagged

($n$ = 82–95 neurons, 8 mice (control); and $n$ = 45–67 neurons, 5 mice (CalEx)) neurons. **g**, The proportion of CS-responsive neurons at E-Ext (two-sided $\chi^2$ test, *$P$ = 0.0062). **h**, Freezing-related unit activity (firing frequency (Hz) rate during freeze − Hz rate during move)/(Hz rate during freeze + Hz rate during move); two-tailed unpaired *t*-test, *$P$ = 0.0275). $n$ = 93–117 neurons per stage, 8 mice (control); $n$ = 61–82 neurons per stage, 5 mice (CalEx). Fluorescence values were $z$-scored to the entire recording period. Neuronal activity values were $z$-scored to the 500 ms pre-pip baseline period. Data are mean ± s.e.m.

carried by individual BLA neurons was lower in the CNO (0.20) than vehicle (0.31) group.

Lastly, we sought to connect the aberrant BLA neuronal encoding at fear memory retrieval caused by astrocyte hM3Dq actuation with the extinction impairment produced by this manipulation (Fig. 2e and Extended Data Fig. 12f). To do so, we performed another microendoscopy neuronal imaging experiment in which BLA astrocytes were hM3Dq actuated through CNO injection during extinction training (but not again before E-Ret) and examined neuronal activity across test stages. Applying unbiased $k$-means clustering to identify neurons that were CS-responsive on E-Ret, but differentially so as a function of treatment, we identified a subpopulation that was CS-excited to a significantly lesser extent in the CNO group than in vehicle controls (Fig. 3q). This reduction in neuronal activity on E-Ret suggests that disrupting astrocyte Ca²⁺ activity impaired plastic changes in neuronal CS encoding that normally occur with extinction.

To further characterize the subset of neurons sensitive to astrocyte hM3Dq actuation, we retrospectively tracked their activity to extinction

training. Notably, we found these neurons were again more likely to be CS excited (relative to neurons non-responsive on E-Ret) during E-Ext and L-Ext, but to a lesser extent in the CNO group (Fig. 3r and Extended Data Fig. 12g). This suggests that disrupting astrocyte Ca²⁺ activity attenuates the activity of a subpopulation of neurons that are persistently CS excited on fear retrieval through extinction. Although the function of these neurons is unclear, they might encode the salience or other associative properties of the CS important for extinction memories to be effectively formed and retrieved; properties that our data suggest are improperly encoded when astrocyte Ca²⁺ activity is disrupted.

Together, these data are consistent with a key role for BLA astrocyte Ca²⁺ signalling in supporting neuronal representations underlying fear memory retrieval and subsequent extinction.

## Astrocytes support extinction and cortical readout

To manifest as behaviourally observable changes in memory retrieval, the BLA neural representation underpinning memory must be

readout to downstream effector regions. This led us to next assess whether astrocytes modulated CS encoding in a population of BLA neurons projecting to the prelimbic subregion of the prefrontal cortex (PL) implicated in CS-related freezing and astrocytic influences on fear memory[26,51–54]. To do so, we expressed the excitatory opsin, channelrhodopsin-2 (ChR2) and, in the same mice, disrupted astrocyte $Ca^{2+}$ dynamics through CalEx expression (avoiding the potential injection stress of CNO-mediated hM3Dq actuation, given the stress sensitivity of the BLA-to-PL pathway)[55] (Fig. 4 and Extended Data Fig. 12h,i). This strategy enabled us to selectively record in vivo, through a chronically implanted multielectrode array, BLA-to-PL neurons identifiable by their short-latency responses to blue light (Fig. 4e). Consistent with earlier work[52,53], we observed robust CS-related activity during E-Ext (that is, fear retrieval) in BLA-to-PL neurons. However, crucially, the proportion of CS-responsive BLA-to-PL (but not non-tagged) neurons and their response magnitude was lower in CalEx-expressing mice compared with viral controls during E-Ext (and remained lower on L-Ext and E-Ret) (Fig. 4f,g and Extended Data Fig. 12j). Moreover, the higher freezing-related activity in BLA-to-PL (versus non-tagged) neurons evident in controls was absent in CalEx-expressing mice (Fig. 4h).

These data suggest that astrocyte $Ca^{2+}$ dynamics support BLA neuronal CS-encoding and readout to downstream effector regions, including PL, to enable fear memory retrieval.

## Conclusion

The instantiation, retrieval and extinction of memories for previously experienced threats have traditionally been viewed as neuronal processes, and previous studies measuring neuronal activity in the BLA have shown that fear states are associated with neural representations in this brain region[1–7]. There is, however, growing evidence of astrocytic contributions to BLA-mediated fear, for example, through fear-memory-related astrocytic engrams[31,36] that may in turn facilitate the selection of specific memory-encoding neuronal engrams[56]. The current findings help to reconcile these two lines of research. We demonstrate that BLA astrocytes dynamically track and causally contribute to fear memory retrieval and extinction and show that astrocytes produce these effects by enabling neuronal representation of fear states in BLA and subsequent readout to a key cortical region mediating the behavioural expression of memory. These findings offer empirical support for the interplay between astrocytes and neurons during neural processes that preserve memory retrieval across contexts and gate experience-dependent adaptations posited to mediate extinction[25,57–59].

One important avenue for future work will be elucidating the molecular mechanisms through which astrocytes modulate fear states and accompanying neuronal representations in BLA and other brain regions. There is currently evidence that, on exposure to salient events, neurotransmitters and neuromodulators, including glutamate, adenosine and noradrenaline, engage astrocytes to support neuronal networks[23,34,36,37,60–65]. For example, noradrenaline signals through astrocyte adrenergic receptors to induce $Ca^{2+}$ activity, prime astrocytic responses to other neurotransmitters and alter synaptic efficacy, thereby mediating neuronal and behavioural flexibility during unproductive actions and repeated exposure to arousing stimuli[17,23,33,34,36,66,67]. Thus, noradrenaline-driven BLA astrocyte responses to threat could support the neuronal computations underlying fear memory retrieval, stabilization and extinction, although it is likely that other astrocyte-modulating neurotransmitters are also integral to these processes[25] and further work will be needed to unravel the complex milieu of mechanisms involved.

Together, we show that BLA astrocytes underpin the generation of memory-supporting neural representations to threat-related environmental stimuli and support plastic changes in these representations that enable extinction. Our findings add to growing evidence that astrocytes exert a potent influence on neuronal circuit function and adaptive state transitions and suggest that targeting astrocytes may be an approach to therapeutically managing abnormal fearful memories in neuropsychiatric disorders.

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

## Methods

### Animals

Mice were male C57BL/6J mice (000664) and male *Vglut1-cre* mice (023527) mice obtained from The Jackson Laboratory (Bar Harbor) and male *Glt1-G-CaMP7*[40] mice obtained from RIKEN BioResource Research Center (G7NG817, RBRC09650). Mice were at least 8 weeks old at the time of surgery, before which they were group-housed in a temperature ($72 \pm 5$ °F) and humidity ($45 \pm 15$%) controlled vivarium under a 12 h–12 h light–dark cycle (lights on 06:00). After surgery, the mice were single housed to prevent cage mates from damaging intracranial implants. Experimental procedures were approved by the NIAAA and National Institute of Natural Sciences Animal Care and Use Committees and followed the NIH guidelines outlined in 'Using Animals in Intramural Research' and the local Animal Care and Use Committees.

### General surgical and histological procedures

**Surgery.** To target injection of viral constructs, optic fibres and GRIN lenses, mice were placed in a stereotaxic alignment system (Kopf Instruments) under isoflurane anaesthesia. Unless stated otherwise, viral constructs (see the relevant sections for details) were unilaterally (one-photon imaging, fibre photometry) or bilaterally (behavioural experiments, single-unit recordings) infused into the BLA in a volume of 0.36 µl over 10 min using a pulled-glass capillary (Drummond Scientific, 2-000-001; tip diameter, ~20 µm) connected to a Nanoject syringe (Drummond Scientific) at the coordinates: anteroposterior (AP) +1.42, mediolateral (ML) +3.27, dorsoventral (DV) −5.15, −4.95 and −4.75 ($1 \times 0.12$ µl injection at each of the three DV coordinates) relative to bregma. After each injection, the glass capillary was left in place for 10 min to ensure diffusion. Testing began no sooner than 4 weeks after surgery, to allow for recovery and virus expression.

For behavioural experiments, the scalp wound was closed with tissue adhesive (GLUture) and a layer of topical antibiotic ointment applied. For fibre photometry experiments, optic fibres (MFC-400/430-0.66-6mm_SM3-FLT, B280-4653.6, Doric Lenses) were unilaterally implanted into the BLA at the coordinates AP +1.40, ML +3.25, DV −4.8 relative to bregma, and affixed to the skull with screws and acrylic dental cement (Coralite Dental Products). For one-photon and two-photon Ca²⁺ imaging experiments, a GRIN lens (ProView GRIN lenses; diameter, 0.6 mm; length, 7.3 mm; 1050-005442, Inscopix) was unilaterally implanted into the BLA at the coordinates AP +1.40, ML +3.25, DV −4.8 relative to bregma, and affixed to the skull with C&B-Metabond Quick Adhesive Cement System (Parkell). For single-unit recordings, a 16-tungsten microelectrode array (35 µm diameter, 150 µm spacing, configured in a semi-circle surrounding the tip of an optic fibre, Innovative Neurophysiology) was unilaterally implanted into the BLA, with the centre of the array at the coordinates: AP −1.50 mm, ML ±3.20 mm, DV −4.95 mm, relative to bregma. Moreover, a ferrule-fibre assembly (Thorlabs) was implanted at a 30° angle, such that the tip was positioned 0.5 mm above the microelectrode tips, with the array and optic fibre affixed to the skull with screws and dental cement (Coralite Dental Products).

**Histology.** On completion of testing, animals were terminally anaesthetized with sodium pentobarbital (50–60 mg per kg) and transcardially perfused with ice-cold PBS followed by ice-cold 4% paraformaldehyde in phosphate buffer. Brains were removed and suspended in 4% PFA overnight and then 4 °C 0.1 M PB for 1–2 days. For fibre photometry and imaging, the head was suspended in 4% PFA overnight before brain extraction. Coronal sections (50 µm) were cut with a vibratome (Leica VT1000 S, Leica Biosystems) in 0.1 M phosphate buffer and/or stored in 2% sodium azide in PBS, then cover slipped with mowiol-based mounting medium (17951, Polysciences) with Hoechst (1 µg ml⁻¹, Thermo Fisher Scientific).

Unless stated otherwise, BLA containing sections were imaged at ×5 (HCX PL FLUOTAR objective, ×5/0.15) using either a Leica slide scanner with an ANDOR Zyla Monochrome camera and Leica DM6 motorized platform controlled by Aperio VERSA software (Leica Biosystems), an Olympus BX41 fluorescence microscope (Olympus America) or a laser-scanning confocal microscope (LSM700, Carl Zeiss) at ×20 (Plan-Apochromat air objective ×20/0.8 NA) and ×63 (Plan-Apochromat oil objective ×63/1.4 NA). Mice with absent/mistargeted viral expression, fibre optic or GRIN lens placement were removed from the analyses. Exceptions to these general histological procedures are described in the relevant sections below.

**Immunohistochemistry.** To verify the specificity of the astrocyte-targeting GCaMP6f, hM3Dq, hM4Di and CalEx viruses, immunostaining for astrocytic (Glial Fibrillary Acidic Protein, S100β) and neuronal (NeuN) markers was conducted, enhanced with primary antibodies to green fluorescent protein (GFP), TdTomato or DsRed. Free-floating sections were washed three times in PBS for 10 min then incubated in blocking solution (0.2% Triton X-100 and 5% normal goat serum in PBS) for 1 h at room temperature. The slices were then incubated overnight at 4 °C with primary antibodies in PBS containing 0.2% Triton X-100 and 1% normal goat serum. The antibodies used were: 1:1,000 chicken anti-GFP (ab13970, Abcam), 1:1000 rat anti-TdTom (EST203, Kerafast), 1:1,000 rabbit anti-DsRed (632496, Clontech), 1:1,000 guinea pig anti-GFAP (173400, Synaptic Systems), 1:100 rabbit anti-S100β (ab52642, Abcam) and 1:1000 rabbit anti-NeuN (ab104225, Abcam).

The next day, the sections were rinsed and incubated for 2 h at room temperature in a cocktail of secondary antibodies (1:500, goat anti-rabbit Alexa 405, A31556; 1:500, goat anti-chicken Alexa 488, A11039; 1:500, goat anti-rabbit Alexa 488, A11034; 1:500, goat anti-guinea pig Alexa 555, A21435; 1:500, goat anti-rat Alexa 555, A21434; 1:500, goat anti-guinea pig Alexa 647, A21450; 1:500, goat anti-rabbit Alexa 680, A21076; all antibodies from Life Technologies) in PBS containing 10% blocking buffer and 0.3% Triton X-100 solution. The sections were then rerinsed, mounted and coverslipped with mowiol-based mounting medium (17951, Polysciences).

BLA containing sections were imaged using a laser-scanning confocal microscope (LSM700, Carl Zeiss) at ×20 (Plan-Apochromat air objective ×20/0.8 NA). For each triple-staining, 2–3 BLA images were acquired per mouse. GFAP-positive cells and NeuN-positive cells, overlapping with virus expression were manually counted by experimenters blinded to experimental group using Fiji software (https://imagej.net/Fiji). The number of GFAP-positive astrocytes and NeuN-positive neurons that were also either GFP- or TdTomato-positive (that is, virus positive) were expressed as the percentage of total virus-expressing cells per 0.01 mm² area.

### Behavioural testing

**Fear conditioning and extinction. Conditioning and extinction apparatus.** For cued fear conditioning and extinction[51,68–70], stimulus presentation was controlled by the Med Associates VideoFreeze system (Med Associates). Conditioning was conducted in a $27 \times 27 \times 11$ cm chamber (80 lux) with opaque metallic walls and a metal rod floor that was cleaned between mice with a 79% water/20% ethanol/1% vanilla-extract solution to provide a distinctive odour. Extinction training and retrieval testing was conducted in a $27 \times 27 \times 14$ cm chamber (20 lux) with transparent walls and a floor covered with wooden chips, sanitized with a solution of 99% water:1% acetic acid.

Conditioning entailed a 180 s baseline period followed by three presentations (60–90 s variable inter-CS interval) of a 30 s, 75 dB, white noise (CS) that co-terminated with a 2 s, 0.4 mA (for chemogenetic experiments, to offset the effect of pre-test injection-stress) or 0.6 mA (all other experiments) scrambled footshock US. There was a 120 s no-stimulus period after the final pairing before the mouse was returned to the home cage.

Extinction training began the day after conditioning, in a single 50× CS session or two 25× CS daily sessions (fibre photometry), bookended

by 180 s pre-CS and post-CS periods. Each CS presentation was 30 s in duration, with a 5 s inter-CS interval except for a 60 s inter-CS interval for fibre photometry and 30 s for one-photon $Ca^{2+}$ imaging to ensure CS-related $Ca^2$ responses returned to baseline between CS-trials.

Extinction retrieval was tested the day after extinction training using the same procedure and context as extinction training, but with only 5× CS presentations.

Fear renewal was tested 1 day later in the same manner as extinction retrieval, but with testing conducted in the conditioning context.

Unless otherwise stated (see the 'In vivo fibre photometry' and 'In vivo one-photon neuronal $Ca^{2+}$ imaging' sections), freezing was scored manually every 5 s as no visible movement except that required for breathing[71], and converted to a percentage ((number of freezing observations/total number of observations) × 100).

**Fibre photometry behavioural control experiments.** Three behavioural control experiments were conducted for fibre photometry. In US-no-CS and CS-no-US experiments, fear conditioning was tested as described above with the exception that, respectively, the US or CS was omitted. There was then a 5× CS test session the next day. In a no-extinction experiment, fear conditioning, extinction training and retrieval was tested as above, with the exception there were only 5× CS presentations at the start of the first extinction training session and another 5× CS presentations at the end of the second training session.

**Anxiety-related behaviours. Novel open field.** The novel open field[72] was a 39 × 39 × 35 cm white Plexiglas square arena (centre of the arena illuminated to around 95 lux). The mouse was placed in a corner and allowed to freely explore the apparatus for 10 min. Total distance travelled and percent time spent in a 20 × 20 cm centre square was measured using the Ethovision automated tracking system (Noldus Information Technology).

**Elevated plus-maze.** The elevated plus-maze[73] was ABS plastic (San Diego Instruments), consisting of 2 × 30 × 5 cm open arms (illuminated to around 95 lux) and two 30 × 5 × 15 cm closed arms (illuminated to around 20 lux) extending from a 5 cm² central square and elevated 20 cm from the ground. The mouse was placed in the centre square to begin a 5 min test. The percentage of time spent in the open arms and total (open + closed) arm entries was measured by the Ethovision automated tracking system (Noldus Information Technology).

**Light/dark exploration test.** The light/dark exploration[74] test apparatus comprised an opaque 39 × 13 × 16 cm black-Plexiglas compartment with a 13 × 8 cm aperture at floor level that allowed access to a larger, 39 × 39 × 35 cm, white-walled square Plexiglas arena (around 95 lux). The mouse was placed at the aperture (facing the opaque compartment) and allowed to explore the apparatus for 10 min. The percentage of time in the dark compartment and the total distance travelled throughout the apparatus was measured by the Ethovision video tracking system (Noldus Information Technology).

## In vivo fibre photometry
**Surgery.** A viral construct containing the genetically engineered $Ca^{2+}$ indicators AAV5-pZac2.1 gfaABC1D-cyto-GCaMP6f (targeting the astrocyte cytosol[75]) (titre, $1.8 \times 10^{13}$ vg per ml, Addgene plasmid 52925; provided by B. Khakh), AAV5-pZac2.1 gfaABC1D-lck-GCaMP6f (targeting astrocyte processes[76]) (titre, $1.8 \times 10^{13}$ vg per ml, Addgene plasmid 52924, provided by B. Khakh) or AAV9-hSyn-jGCaMP7f-WPRE (targeting neurons[77]) (titre $2.5 \times 10^{13}$ vg per ml, Addgene plasmid 104488, provided by D. Kim) was unilaterally injected (0.45 μl) into the BLA, and BLA-targeting optic fibres were unilaterally implanted, as described above in the 'General surgical and histological procedures' section. The same procedure was used for simultaneous photometry of astrocytes and neurons, but with AAV5-pZac2.1 gfaABC1D-cyto-GCaMP6f injected into the BLA of one hemisphere and pAAV.Syn.GCaMP6f.WPRE.SV40 (titre, $2.1 \times 10^{13}$ vg per ml, Addgene plasmid 100837; provided by D. Kim) injected into the BLA of the other hemisphere (counterbalancing which hemisphere received each virus). Before behavioural testing, mice were

handled for 2 min per day for 2 days and habituated to being connected to the optic fibre cables in the home cage for 1 h per day for 3 days.

**Data collection.** To record fluorescence signals via fibre photometry[69,78], a RZ5P Processor acquisition system (Tucker-Davis Technologies) used two continuous sinusoidally modulated LEDs (Thorlabs) at 473 nm (511 Hz) and 405 nm (211 Hz) as a light source to respectively excite GCaMP and an isosbestic autofluorescence signal. The LEDs were connected to a mini cube (Doric Lenses) and each band-pass was filtered before being coupled to a single large core (400 μm), high-NA (0.48) optical fibre patch cord. Emitted light was projected through the same mini cube, passed through a GFP emission bandpass filter (500–525 nm) and focused onto a Newport Visible Femtowatt Photoreceiver (Doric Lenses). Light intensity at the tip of the patch cable (interface between patch cable and fibre implant) was in the 50–100 μW range for each channel. Photometry signals were temporally aligned to US and CS-onset using the Med Associates VideoFreeze system (Med Associates).

**Data analysis.** US- and CS-related changes in GCaMP activity were transformed into $z$ scores of the 5-s period preceding each event ($z = [\Delta F - \text{mean}(\Delta F(t = -5 \text{ to } 0))]/\text{s.d.}$, where s.d. is the standard deviation of $\Delta F$ values during the pre-event period). Periods (bin size = 10 ms) in which peri-event activity differed from null were statistically determined as 95% confidence intervals (bCI) calculated from a 1,000-fold bootstrap estimate, as previously described[79] (https://github.com/philjrdb/FibPhotom). Values for AUC were calculated using MATLAB's built-in trapz function, which uses trapezoidal numerical integration to calculate the AUC ($z$ score) between inputted $x$ values (time) on a graph of $z$ score versus time. US- and CS-related changes were examined in data $z$ scored to the 5 s pre-stimulus period. The number of $Ca^{2+}$ transients was calculated using a peak detection code (https://gist.github.com/antiface/7177333), with each recording visually inspected to ensure the accuracy of the code and the threshold parameter accordingly adjusted to differentiate between peaks and depressions (local maxima and minima) in the input signal. The results were expressed as the mean number of events or cumulative event frequency for a given experimental period and condition. CNO effects on transients were examined in data $z$ scored to the 30 min pre-CNO period.

To examine the relationship between astrocyte $Ca^2$ activity and movement during E-Ext and L-Ext, video recorded behaviour was analysed using the EzTrack software program[80]. $\Delta F/F\ Ca^{2+}$ values were aligned to instances of movement onset and offset as previously described[23] and normalized to the session-average $\Delta F/F\ Ca^{2+}$ value.

## In vivo two-photon $Ca^{2+}$ imaging
**Surgery.** A GRIN lens was unilaterally implanted into the BLA of Glt1-G-CaMP7 mice, as described above in the 'General surgical and histological procedures' section.

**Data collection.** Data were acquired using a Nikon A1R MP+ multiphoton laser-scanning microscope (Nikon Instech) controlled by NIS-Elements software (Nikon Instech). During imaging[81], the mice were head-fixed but, to reduce the stress of immobilization, were able to ambulate on a rotating stainless-steel disc connected to a footshock generator (SG-1000S, Melquest). US and CS presentation was controlled by the Bpod HiFi Module HD State Machine r0.5 (Sanworks) and aligned to imaging frames via a multifunction I/O device (USB6343, National Instruments) running a custom script written in LabVIEW. A laser (Ti:sapphire, $\lambda = 920$ nm, Mai Tai DeepSee; Spectra-Physics) used as a light source to excite GCaMP was controlled by a galvanometer scanning an $xy$ plane (512 × 512 pixels) at 500 ms frames using a water immersion ×16 objective (NA 0.8; Nikon). Fear conditioning and extinction was conducted as described above for fibre photometry, except for the omission of fear renewal and the addition of a pre-conditioning

(Pre-Con) habituation session entailing 5× CS presentations, conducted the day before fear conditioning.

**Data analysis.** Images were analysed using ImageJ (v.1.37, National Institutes of Health) and MATLAB. Images were corrected for focal/$xy$ plane displacements using StackReg and TurboReg and open-source code for 3D two-photon imaging registration (https://github.com/atakehiro/TurboReg_macro). Relative changes in $Ca^{2+}$ fluorescence $F$ were calculated by $\Delta F/F_0 = (F - F_0)/F_0$ (where $F_0$ is a 25-frame moving average) using Astrocyte Quantitative Analysis software (AQuA)[38]. Overall $Ca^{2+}$ changes across testing phases in AUC $\Delta F/F_0$ GCaMP values within regions of interest (ROIs) were identified using AQuA and custom MATLAB code from data concatenated across all five testing phases. Changes in $Ca^{2+}$ event number were identified using AQuA and custom MATLAB code applied to individual testing phases. 11 ROIs with $\Delta F/F_0$ values >7 or a single AUC duration longer than 60 frames (2 fps) were excluded from the analyses. To assess plasticity across test stages, $Ca^{2+}$ activity around a 20-s peri-CS period was used to determine CS-modulated astrocyte events (post-CS > pre-CS comparison using paired $t$-test (alpha, $P < 0.05$) on a given test stage and then segregate events into those exhibiting increasing, decreasing, sustained or consistent non-responsivity to the CS as a function of conditioning (Pre-Con to early extinction in cross test stage co-registered events) or extinction (early to late extinction).

**CS decoding.** A response vector was created by demarking the all-stage concatenated data according to presentation of the CS around a 5-s pre-CS period and 30-s post-CS period and decoded using a logistic regression model with an L2 penalty. To equate the length of the pre- and post-CS periods for decoding, a 5-s window from the post-CS period was constructed from activity in 10× non-contiguous 0.5-s intervals randomly sampled from the 30-s window. This construction process was repeated 1,000 times and the decoder performance was estimated by the mean average of the resultant F1 scores on held-out trials (trial-grouped fivefold cross-validation). Values were compared to the mean F1 scores obtained from circularly rotating the data (shuffle) 1,000 times with respect to CS presentation.

**Histology.** Histology was performed as described above in the 'General surgical and histological procedures' section.

## Astrocyte manipulations

### hPMCA2w/b Ca$^{2+}$ extruder (CalEx). Surgery and behavioural testing.
A viral construct containing the plasma membrane $Ca^{2+}$ pump, CalEx[82] (AAV5-pZac2.1-GfaABC1D-mCherry-hPMCA2w/b, titre $2.0 \times 10^{13}$ vg per ml, Addgene plasmid 111568, provided by B. Khakh) or a tdTomato-expressing control virus (AAV5-pZac2.1 gfaABC1D-tdTomato, titre $7.0 \times 10^{12}$ vg per ml, Addgene plasmid 44332, provided by B. Khakh) was bilaterally injected (0.36 µl per hemisphere) into the BLA, as described above in the 'General surgical and histological procedures' section. For simultaneous fibre photometry, AAV5-pZac2.1 gfaABC1D-cyto-GCaMP6f was unilaterally injected into the BLA, and BLA-targeting optic fibres were unilaterally implanted, as described above in the 'In vivo fibre photometry' and 'General surgical and histological procedures' sections. Before behavioural testing, mice were handled for 2 min per day for 3 days.

### Chemogenetics. Surgery.
A viral construct containing either hM3Dq (AAV5-GFAP-hM3D(Gq)-mCherry, titre $2.0 \times 10^{13}$ vg per ml, Addgene plasmid 50478, provided by B. Roth), hM4Di (AAV-GFAP-hM4D(Gi)-mCherry, titre $1.0 \times 10^{13}$ vg per ml, Addgene plasmid 50479, provided by B. Roth) or mCherry (AAV5-GFAP104-mCherry, titre $1.7 \times 10^{13}$ vg per ml, Addgene plasmid 58909, provided by E. Boyden), was bilaterally injected into the BLA (0.36 µl per hemisphere), as described above in the 'General surgical and histological procedures' section. For simultaneous fibre

photometry, AAV5-pZac2.1 gfaABC1D-cyto-GCaMP6f was unilaterally injected into the BLA and BLA-targeting optic fibres were unilaterally implanted, as described above in the 'In vivo fibre photometry' and 'General surgical and histological procedures' sections. Before behavioural testing, mice were handled for 2 min per day for 2 days and then injected with 0.1 ml of sterile saline intraperitoneally for 3 days to habituate to injection stress.

For DREADD actuation, CNO was injected intraperitoneally at a dose of 3.0 mg ml$^{-1}$ per kg body weight, except in dose-comparison experiments in which CNO was injected at a dose 0.1 mg ml$^{-1}$ per kg body weight. In a separate experiment to examine behavioural effects of a lower concentration hM3Dq virus, CNO was injected at a dose of 3 mg per kg in animals expressing a 1:8 diluted concentration of the hM3Dq-expressing virus (titre, $2.5 \times 10^{12}$ vg per ml).

**Behavioural testing.** CNO was injected in separate groups of hM3Dq-expressing animals either (1) 30 min before fear conditioning; (2) immediately after fear conditioning; (3) 30 min before extinction training; or (4) 30 min before E-Ret. CNO was injected in hM4Di-expressing animals 30 min before extinction or 30 min before E-Ret. $Ca^{2+}$ activity was measured in hM3Dq- and hM4Di-expressing animals, using in vivo fibre photometry, in the home cage for 30 min before CNO injection and then for the 30-min post-injection period up to the beginning of behavioural testing, as described above in the 'In vivo fibre photometry' section. After completion of behavioural testing, $Ca^{2+}$ activity was recorded for 2 h after injection in a subset of mCherry-, hM3Dq- and hM4Di-expressing mice. For anxiety-related behaviour tests, CNO was injected into hM3Dq- and hM4Di-expressing animals 30 min before testing, with at least a 1-week interval between tests.

**Chemogenetic control experiments.** In an experiment to test for potential behavioural effects of CNO per se, mice bilaterally expressing a control virus in the BLA were injected before extinction with either vehicle or CNO. In an experiment to test for potential behavioural effects of hM3Dq expression per se, mice bilaterally expressing a hM3Dq or control virus in the BLA were injected before extinction with vehicle. A replicate experiment to match the design of the one-photon neuronal imaging (that is, hM3Dq actuation before F-Ret) was conducted in the same manner: that is, vehicle or CNO was injected before F-Ret in mice bilaterally expressing a hM3Dq or control virus in the BLA. Finally, the effects of hM3Dq actuation on shock-related freezing and flinching (the magnitude of response was manually scored on a scale of 0–5) were tested 30 min after CNO injection by presentation of five 0.4 mA footshocks each separated by a 60 s interval. Moreover, shock-related $Ca^{2+}$ activity was measured during five 0.4 mA footshocks or, in a separate experiment, five 1.0 mA footshocks, as described above in the 'In vivo fibre photometry' section.

## Electron microscopy

**Fixation and immunohistochemistry.** Mice were transcardially perfused with 0.1% glutaraldehyde/4% paraformaldehyde in 0.1 M phosphate buffer, pH 7.4, rinsed in the same buffer. Sections (40 µm) were cut with a vibratome (Leica VT1000 S, Leica Biosystems) and those containing BLA were rinsed in PBS consisting of 0.01 M phosphate buffer at pH 7.4 with 154 mM sodium chloride. The sections were then blocked for 15 min in 0.1% sodium borohydride in PBS, rinsed in PBS and blocked for 1 h in 1% BSA (Jackson ImmunoResearch Labs). The sections were incubated overnight in mouse anti-GFP (A-11120, 1859591; 1:500, Invitrogen) in PBS with 1% BSA at room temperature, rinsed in PBS and incubated for 30 min in biotinylated goat anti-mouse antibody (31802, 1:200, Invitrogen).

After rinsing in PBS, the sections were incubated for 30 min with an avidin/biotin complex (ABC) peroxidase reagent (Vectastain Elite ABC Kit PK-6100, Vector Labs) according to the manufacturer's instructions, rinsed in PBS and reacted for 5 min with 1 mM 3,3-diaminobenzidine and 0.0015% hydrogen peroxide in PBS. After rinsing, the sections were

incubated in 1% hydrogen peroxide in PBS for 15 min to quench the remaining peroxidase, then a second round of labelling was performed as above with rabbit anti-DsRed primary antibody (632496, 1904182; 1:500, Takara Bio), goat anti-rabbit secondary antibody (A32731, 1:200, Invitrogen) and ABC. The second antibody was detected with the Vector VIP peroxidase substrate (Vector Labs).

**Microscopy preparation.** BLA-containing sections were post-fixed in 2.5% glutaraldehyde/4% paraformaldehyde in 0.1 M cacodylate buffer, pH 7.4, rinsed in the same buffer, fixed for 1 h in reduced osmium (1% osmium tetroxide with 1.5% potassium ferrocyanide), rinsed in buffer and fixed for 1 h in 1% osmium tetroxide. After buffer rinses, the sections were rinsed quickly in water then dehydrated in a series of ethanol dilutions in water (50%, 70%, 90% and 100%) containing 1.5% uranyl acetate. The samples were transferred into acetone, infiltrated with LX-112 resin (Ladd Research) in acetone, embedded in pure resin and cured for 48 h at 60 °C.

**Serial sectioning and imaging.** One block from each of two mice was trimmed to expose the medial half of the basal amygdala. The sections were cut at 45 nm on a Leica UC7 ultramicrotome (Leica Biosystems) and collected on pioloform-coated slot grids. Serial sections were also cut from one block. Sections were imaged at ×5,000 on a JEOL 1400 TEM at 120 kV with an AMT NanoSprint43 digital camera. For 3D reconstruction, 11 serial sections were imaged. Reconstruct software[83] was used for registration and segmentation and 3DS Max software (Autodesk) was used for rendering.

### Inhibition of synaptic transmission
A tetanus toxin virus was used to inhibit synaptic release in BLA principal neurons, as previously described[84]. A viral construct containing tetanus toxin light chain[85] (AAV1-DIO-GFP:TeNT, titre $4.0 × 10^{12}$ vg per ml, provided by L. Zweifel), was bilaterally injected in the BLA (0.36 µl per hemisphere) of *Vglut1-cre*+ mice or *Vglut1-cre*− controls, as described above in the 'General surgical and histological procedures' section. Then, 4 weeks later, fear conditioning and extinction was conducted as described above.

### In vitro slice neuronal electrophysiological recordings
**Slice preparation.** A viral construct containing hM3Dq (AAV5-GFAP-hM3D(Gq)-mCherry, titre $2.0 × 10^{13}$ vg per ml, Addgene plasmid 50478, provided by B. Roth) was bilaterally injected into the BLA (0.36 µl per hemisphere), as described above in the 'General surgical and histological procedures' section.

At least 4 weeks after surgery, brains were removed and sliced as previously described[86]. In brief, the mice were deeply anaesthetized using isoflurane and then intracardially perfused with an ice-cold *N*-methyl-D-glutamine based solution (NMDG) (92 mM NMDG, 2.5 mM KCl, 1.25 mM NaH$_2$PO$_4$, 10 mM MgSO$_4$, 0.5 mM CaCl$_2$, 30 mM NaHCO$_3$, 20 mM glucose, 20 mM HEPES, 2 mM thiouera, 5 mM Na-ascorbate and 3 mM Na-pyruvate, pH 7.3–7.4, 300–310 mOsm and saturated with 95% O$_2$, 5% CO$_2$). Brains were removed and coronal slices (300 µm) cut in ice-cold NMDG-based solution using a VT1200 S vibratome (Leica Biosystems). Slices were then held in NMDG-based solution maintained at 35 °C for around 12 min before being transferred to a room-temperature HEPES artificial cerebrospinal fluid (ACSF) (92 mM NaCl, 2.5 mM KCl, 1.25 mM NaH$_2$PO$_4$, 2 mM MgSO$_4$, 2 mM CaCl$_2$, 30 mM NaHCO$_3$, 25 mM glucose, 20 mM HEPES, 2 mM thiouera, 5 mM Na-ascorbate and 3 mM Na-pyruvate, 300–310 mOsm, and saturated with 95% O$_2$ and 5% CO$_2$).

**Data collection.** Slices were transferred from the modified HEPES ACSF to a standard ACSF (118 mM NaCl, 2.5 mM KCl, 26.2 mM NaHCO$_3$, 1 mM NaH$_2$PO$_4$, 20 mM glucose, 2 mM MgCl$_2$ and 2 mM CaCl$_2$, at 25 °C, pH 7.4, 300–310 mOsm, and gassed with 95% O$_2$ and 5% CO$_2$). Neurons were visualized using an infrared differential interference contrast camera

on a BX51WI system (Olympus). Whole-cell patch clamp recordings were performed using micropipettes pulled from a borosilicate glass capillary tube using a PMP102 Micropipette Puller (MicroData Instruments). Electrode tip resistance was 3–5 MΩ. Recordings were acquired using an Axon Multiclamp 700 and Axon Digidata 1550 A (Molecular Devices), sampled at 10 kHz, low-pass filtered at 3 kHz and analysed in Igor Pro 8 (WaveMetric). Access resistance was continuously monitored and changes greater than 20% from the initial value were excluded from the analysis. Recordings were made in 1–2 cells per mouse.

To assess the effects of hM3Dq actuation of BLA astrocytes on nearby neuronal activity, a potassium gluconate-based intracellular solution (155 mM K-gluconate, 4 mM KCl, 10 mM HEPES, 4 mM MgATP, 0.3 mM Na$_3$GTP, 10 mM phosphocreatine, pH 7.3, 285–290 mOsm) was used. Neurons were first current clamped and the membrane potential was maintained at −70 mV while a 500 ms current step was applied every 5 s, starting at −200 pA to 600 pA increasing by 50 pA with every step. The neurons were then held in voltage clamp at −70 mV and spontaneous EPSCs (sEPSCs) measured. After a 5 min baseline period, 10 µM CNO was bath applied and maintained in CNO for 35 min and sEPSCs again measured as above.

### In vivo one-photon neuronal Ca$^{2+}$ imaging
**Surgery.** A viral vector containing the Ca$^{2+}$ indicator jGCaMP7f[77] (pGP-AAV-syn-jGCaMP7f-WPRE, titre $2.5 × 10^{13}$ vg per ml, 1:5 dilution in Dulbecco's PBS, Thermo Fisher Scientific, Addgene plasmid 104488, provided by D. Kim and the GENIE Project) was unilaterally infused (0.50 µl) into the BLA, as described above in the 'General surgical and histological procedures' section. Then, 3–4 weeks later, before behavioural testing, mice were handled for 2 min per day for 3 days and then habituated to being connected to the microscope and tether for 30 min per day for 3 days.

**Data collection.** Fear conditioning and extinction was conducted as described above, with 3 mg per kg CNO or vehicle injected into mice bilaterally expressing hM3Dq in BLA astrocytes 30 min before F-Ret in one experiment or, in a separate experiment, 30 min before extinction training. BLA neurons were imaged using a miniature microscope (nVistaTM 3.0, Inscopix) and IDAS HD software (Inscopix) at a frame rate of 20 Hz with an LED power of 10–60% (0.9–1.7 mW at the objective, 475 nm), analogue gain 1, 1,080 × 1,080 pixels. Timestamped imaging frames were collected for temporal alignment using the Med Associates VideoFreeze system (Med Associates).

**Data analysis.** For data preprocessing, imaging frames were downsampled to 10 Hz, spatially filtered and motion-corrected using Inscopix software, then putative neurons were identified using constrained non-negative matrix factorization (CNMF-E)[54]. Traces for each neuron were manually inspected to exclude false-positive or false-negative cell mask allocation. Raw Ca$^{2+}$ traces were obtained by averaging pixel values in each mask. Slow drift of the baseline signal over the course of minutes was removed using a low-cut filter (Gaussian, cut-off 2–4 min). Relative changes in Ca$^{2+}$ fluorescence $F$ were calculated by $\Delta F/F_0 = (F − F_0)/F_0$ (where $F_0$ is the median fluorescence of the entire session trace). Ca$^{2+}$ activity data were analysed using the Python programming language (https://www.python.org/). To compute CS-related Ca$^{2+}$ responses, time-normalized AUC fluorescence values ($z$ scored to the entire session) were calculated for each stimulus presentation. To examine the relationship between astrocyte Ca$^{2+}$ activity and freezing, video-recorded behaviour was analysed using the EzTrack software program, as described above in the 'In vivo fibre photometry' section.

**CS-modulated activity.** CS modulation was defined by calculating the delta in activity for the 5-s post-CS relative to the 5-s pre-CS period and a corresponding distribution of 500 null values obtained from activity circularly rotated with respect to CS presentation. Modulated

neurons were statistically defined based on the proportion of null values greater than the observed value ($P < 0.05$). CS modulated neurons were statistically defined based on the proportion of null values greater than the observed value ($P < 0.05$) on a given test stage. These results were used to segregate cross test stage co-registered neurons into subsets exhibiting increasing, decreasing, sustained, or consistent non-responsivity to the CS as a function of fear conditioning (Pre-Con to F-Ret).

**Neuronal topography.** To determine the spatial organization of CS-modulated neurons within the field of view, the Euclidean distance (arbitrary units) between individual CS-related neurons and their respective centroid was calculated, as previously described[54].

**Decoding treatment group identity by animal.** CNO treatment group identity was decoded using a logistic regression model with an L2 penalty (balancing the number of animals per group, $n = 7$) and creating a response vector based on specific features of F-Ret CS-related neuronal activity during a 10-s post-CS period demarked according to group. To ensure that the same number of neurons was used for each mouse, a randomly selected (through sampling with replacement) subset of $n = 5$ neurons was sampled 100 times and the average computed. The decoder performance was then evaluated using a specific neural feature (see below) from mean F1 scores across 14 cross-validation folds (that is, 1-fold per mouse) and compared to the mean F1 scores obtained from circularly rotating the data (shuffle) with respect to group identity, 100 times. The following features were examined: mean activity of positively CS-modulated, negatively CS-modulated and non-modulated neurons (as defined above), the s.d. of activity of all neurons during the 5-s post-CS period and the decomposed activity values of the three highest-eigenvalue principal components computed by PCA on $z$-scored, trial-averaged and mean-normalized activity from a 10-s peri-CS period.

**Decoding CS as a function of population size.** A response vector was created by demarking data according to presentation of the CS around a 10-s peri-CS period and decoded from the neuronal data using a logistic regression model with an L2 penalty (balancing the number of animals per group, $n = 7$). Decoding was conducted with the number of neurons available to the decoder (randomly selected by sampling without replacement) increasing in one-neuron increments from 2 to 100. The performance at each population size was estimated as the mean held-out F1 score of 50 decoding models, each trained on a random sample of neurons of the corresponding size and compared to values of 50 iterations of circularly rotated data (shuffle) with respect to CS presentation.

To model the relationship between population size and decoding performance, a saturating function[50,87] was fit to the data. The saturating function was defined as $y = bn/a + n$, where $y$ is the decoder performance, $n$ is the population size, $b$ is the performance at an infinite number of neurons and $a$ is the saturation rate. The saturating function was fit to F1 scores using the curve_fit function from the SciPy Python package. To estimate the variance of model fits, the function was fit to the data 1,000 times, each time with a randomly selected combination of the 50 model fits (50 random samples with replacement) per population size.

**Relationship between freezing and CS presentation decoding.** CS presentation was decoded from neuronal activity during F-Ret as described in the 'Decoding CS as a function of population size' section, but here using the entire population of neurons available (balancing the number of animals per group, $n = 7$). The resultant decoder coefficient values for each neuron were averaged by mouse and the mouse-averaged correlated with the respective F-Ret CS-related freezing values.

**$k$-Means clustering.** To identify neurons that were CS-responsive on extinction and differentially activated by CNO treatment, neuronal activity in the 5-s pre-CS through 30-s post-CS period during L-Ext and E-Ret was subject to $k$-means clustering ($n = 4$ clusters).

## In vivo neuronal electrophysiological recordings and phototagging

**Surgery.** A viral vector containing CalEx was bilaterally infused into the BLA, as described above in the 'hPMCA2w/b Ca$^{2+}$ extruder' and 'General surgical and histological procedures' sections. Additionally, the retrograde-travelling excitatory opsin, channelrhodopsin (retroAAV-hSyn-hChR2(H134R)-eYFP, titre $2.5 \times 10^{13}$ vg per ml, Addgene plasmid 26973, provided by K. Deisseroth) was bilaterally infused (0.20 µl) into dmPFC at a 20° angle at the coordinates: AP +1.95 mm, ML ±1.00 mm, DV −1.90 mm, relative to bregma, as described above in the 'General surgical and histological procedures' section. During the same surgery, a microelectrode array and optic fibre were unilaterally implanted into the BLA[88], as described above in the 'General surgical and histological procedures' section. Before testing, the mice were habituated to the recording tether and patch cable for 20 min in the home cage handled for 2 days. During habituation, 20 Hz blue light pulses (fifty 1 s trains of 20 Hz light pulses with a 3 s intertrial interval) were delivered through the optic fibre to identify light-responsive isolated single units. Light responsivity was reconfirmed after each behavioural test session.

**Data collection.** Fear conditioning, extinction training and retrieval were tested as described above with the exception that the CS consisted of a 20 s train of white noise pips[89] (500 ms pips delivered at 1 Hz, 70 dB) and the extinction context was a clear acrylic, cylindrical 30 cm diameter chamber with an open top to accommodate the tether connecting the head-stage. Electrophysiological and behavioural videos were acquired using the Omniplex Neural Data Acquisition System and Cineplex Behavioural Research System (Plexon). Radiant Software (Plexon) was used to generate TTL pulses to control stimulus-presentation. Plex Control Software (Plexon) recorded TTL signals to synchronize electrophysiological recordings with stimulus-presentation and videos of behaviour.

**Data analysis.** Single units were sorted manually using Offline Sorter v.4.0 (Plexon) and analysed using NeuroExplorer v5 (Nex Technologies). To identify CS-responsive units, data during a 500 ms window following the onset of each pip during each of the 5 CS presentations of E-Ext, L-Ext and E-Ret were binned in 100 ms bins and $z$-score normalized to the 500 ms pre-pip baseline period, as follows: (CS bin value − baseline mean value)/(s.d. of the baseline mean value). Units with at least 1 of 5 bins with a value of >2.58 during the post-pip period were deemed significantly different from baseline ($P < 0.01$) and classified as pip-responsive. To identify freezing-related unit activity, firing rates during freezing epochs were compared to firing rates during movement epochs on the 5 CS presentations and associated ITI periods of E-Ext, as follows: (Hz rate during freeze − Hz rate during move)/(Hz rate during freeze + Hz rate during move).

**Histology.** To verify electrode placements, mice were anaesthetized with 2% isoflurane and marking lesions made by passing current (40 µA for 2 s) through each of the microelectrodes (S48 Stimulator and Model CCU1, Grass Technologies). At least 1 day later, mice were euthanized with 150 mg per kg Euthasol (Henry Schein) and transcardially perfused with 4% PFA solution. The brains were extracted and stored in PBS. Coronal sections (50 µm thick) were cut with a VT1000 vibratome (Leica), mounted on slides and cover slipped with Invitrogen Fluoromount-G mounting medium with DAPI. Viral and fibre placements and marking lesion sites were documented with the aid of a Keyence BZ-X microscope (Keyence).

## Statistical analysis

Data were analysed using Python (https://www.python.org), MATLAB (MathWorks) and Prism (GraphPad) software. Group effects were examined using two-tailed unpaired or paired Student's $t$-tests or ANOVA, depending on the number of independent variables. Post hoc tests (Šídák's and Fisher's LSD) were conducted after ANOVA. $\chi^2$ tests were used to analyse group proportion differences. Sample sizes were based on pilot and previous (similar types of) experiments and were not statistically predetermined. Experiments were powered to match sample sizes typical of the technique reported in the field, although no formal power analysis was performed a priori. Images shown were selected from multiple independent samples (that is, different animals). $P < 0.05$ was considered the threshold for statistical significance, and $P < 0.0001$ is the lowest $P$ value reported. Experiments were biologically replicated across multi-animal batches (animal $N$ and $n$ numbers for each experiment are provided in the corresponding figure legends and Supplementary Table 1). Representative example micrographs depicted in Figs. 1b,m,n, 2b,h, 3b and 4b and Extended Data Figs. 1a,h, 3a,i, 4a, 5k, 9a,b, 10b, 11g and 12i) were selected from at least three independent animals, except for electron microscopy, which was based on two mice. Experimental groups were randomized, and experimenters were blinded to group allocation whenever possible.

## Reporting summary

Further information on research design is available in the Nature Portfolio Reporting Summary linked to this article.

## Data availability

The data supporting the findings of this study are available in Article and its Supplementary Information, and from the corresponding authors on reasonable request. Source data are provided with this paper.

## Code availability

Custom-written codes used to analyse data from this study are available on reasonable request from the corresponding authors.

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

**Acknowledgements** We thank B. Averbeck for discussions; and M. Abril and J. Perez-Garza for technical assistance. Resources from the Bioscience Electron Microscopy Laboratory at the University of Connecticut were used for electron microscopy. Research was supported by Grants-in-Aid for Transformative Research Areas (A) grant 20H05899 (H.W.), JST CREST grant JPMJCR22P6 (H.W.), the NIMH Intramural Research Program (M.A.P.) and NIAAA Intramural Research Program (A.H.).

**Author contributions** Conceptualization: O.B. and A.H. Methodology, data collection and formal analysis: O.B., R.O., Y.T., A.M., C.W., S.Z. V.O., O.C., H.B., S.M., J.J.O., K.L., Y.F., J.G., L.E.O., M.A.P., H.W., L.R.H. and A.H. Funding acquisition and project administration: M.A.P., L.R.H., H.W. and A.H. Writing—original draft: O.B. and A.H. Writing—review and editing: O.B., R.O., Y.T., A.M., C.W., S.Z. V.O., O.C., H.B., S.M., J.J.O., K.L., Y.F., J.G., L.E.O., M.A.P., H.W., L.R.H. and A.H.

**Competing interests** The authors declare no competing interests.

**Additional information**
**Correspondence and requests for materials** should be addressed to Olena Bukalo or Andrew Holmes.

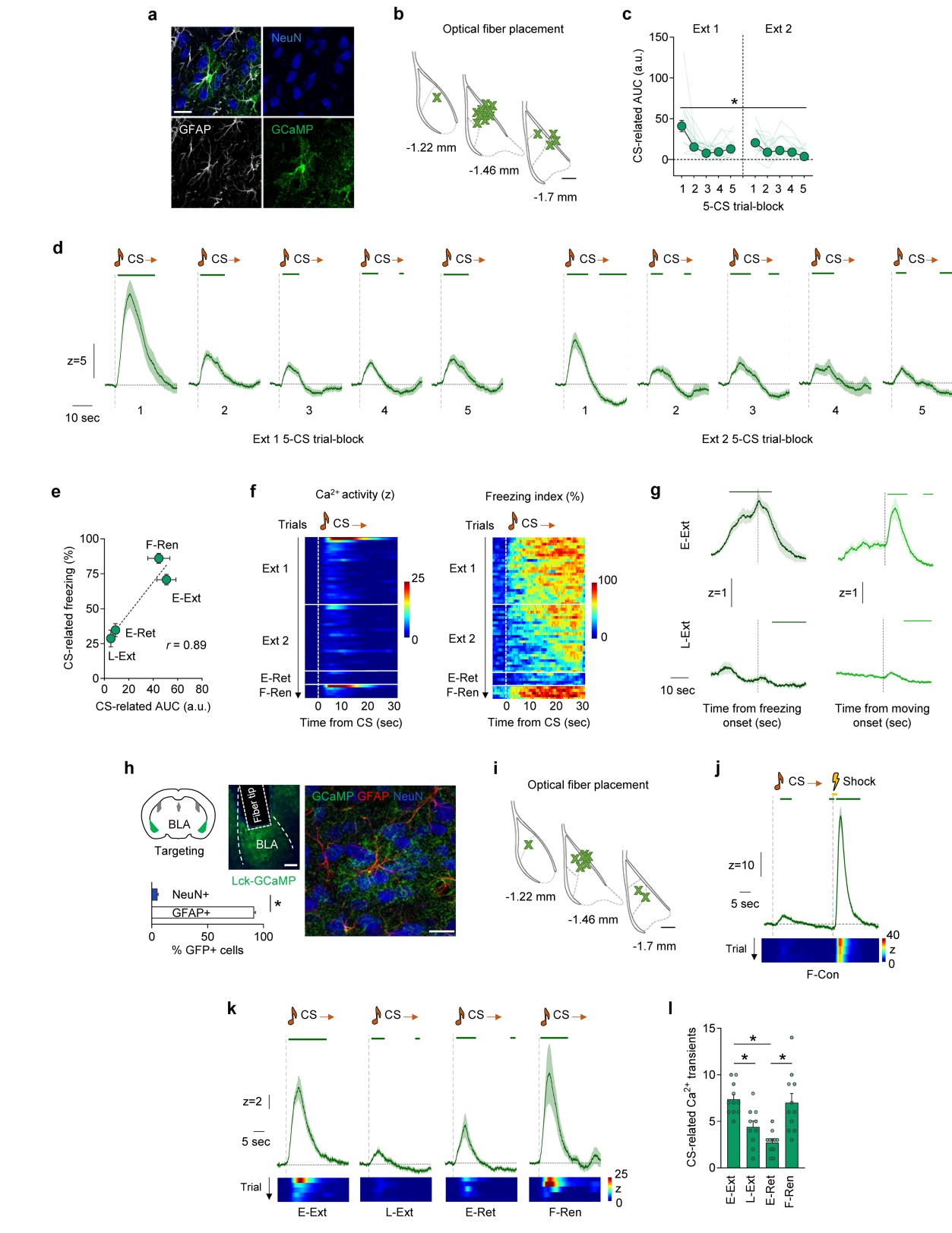

**Extended Data Fig. 1** | See next page for caption.

**Extended Data Fig. 1 | Fibre photometry measurement of astrocyte Ca²⁺ correlates of fear. a**, Representative GCaMP expression in astrocytes (GFAP) and neurons (NeuN), scale bar: 20 μm. **b**, Optical fibre tip location for cyto-GCaMP6f in vivo astrocyte Ca²⁺ fibre photometry. **c**, AUC quantification CS-related Ca²⁺ activity during each 5-CS trial-block of extinction session 1 (Ext1) and 2 (Ext 2) shown in panel **d** (1-way repeated measures ANOVA and Šídák's *post hoc* tests, E-Ext (Ext1 block 1) vs L-Ext (Ext2 block 5) *$P = 0.0003$, n = 17 mice/stage. **e**, Relationship between AUC of CS-related Ca²⁺ transients and freezing across test-sessions (2-tailed Pearson's correlation, $P = 0.0551$, n = 14–17 mice). **f**, Heatmaps for CS-related Ca²⁺ activity (left) and EzTrack-detected freezing (right) across test phases (n = 14 mice). **g**, Ca²⁺ activity aligned to freezing and movement during early (E-Ext, first 5 trials of extinction session 1) and late (L-Ext, last 5 trials of extinction session 2) extinction (n = 133–135 onset episodes/stage, n = 158–200 offset episodes/stage, 14 mice). **h**, Example fibre placement and expression of the membrane-targeted Ca²⁺ indicator lck-GCaMP6f in astrocytes (GFAP) and neurons (NeuN) (2-tailed paired *t*-test, $P < 0.0001$*, n = 13 sections/6 mice) (scale bars: 200 μm (left), 10 μm (right). **i**, Optical fibre tip location for lck-GCaMP6f in vivo astrocyte Ca²⁺ fibre photometry. **j**, Population CS-related and shock-related Ca²⁺ activity during F-Con, with 3-trial-wise heatmap below (n = 11 mice). **k**, Population CS-related Ca²⁺ activity during E-Ext, L-Ext, E-Ret, and F-Ren, with 5-trial-wise heatmaps below (n = 10–11 mice/stage). **l**, AUC quantification CS-related Ca²⁺ activity shown in panel **k** (1-way repeated measures ANOVA and Šídák's *post hoc* tests, E-Ext vs L-Ext *$P = 0.0365$, E-Ext vs E-Ret *$P = 0.0003$, E-Ret vs F-Ren *$P = 0.0121$, n = 10–11 mice/stage). Horizontal lines above traces in d,g,j,k denote permutation test-determined significant difference from chance. Fluorescence values z-scored to the 5-s pre-CS, pre-freezing or pre-movement periods. Data mean±SEM.

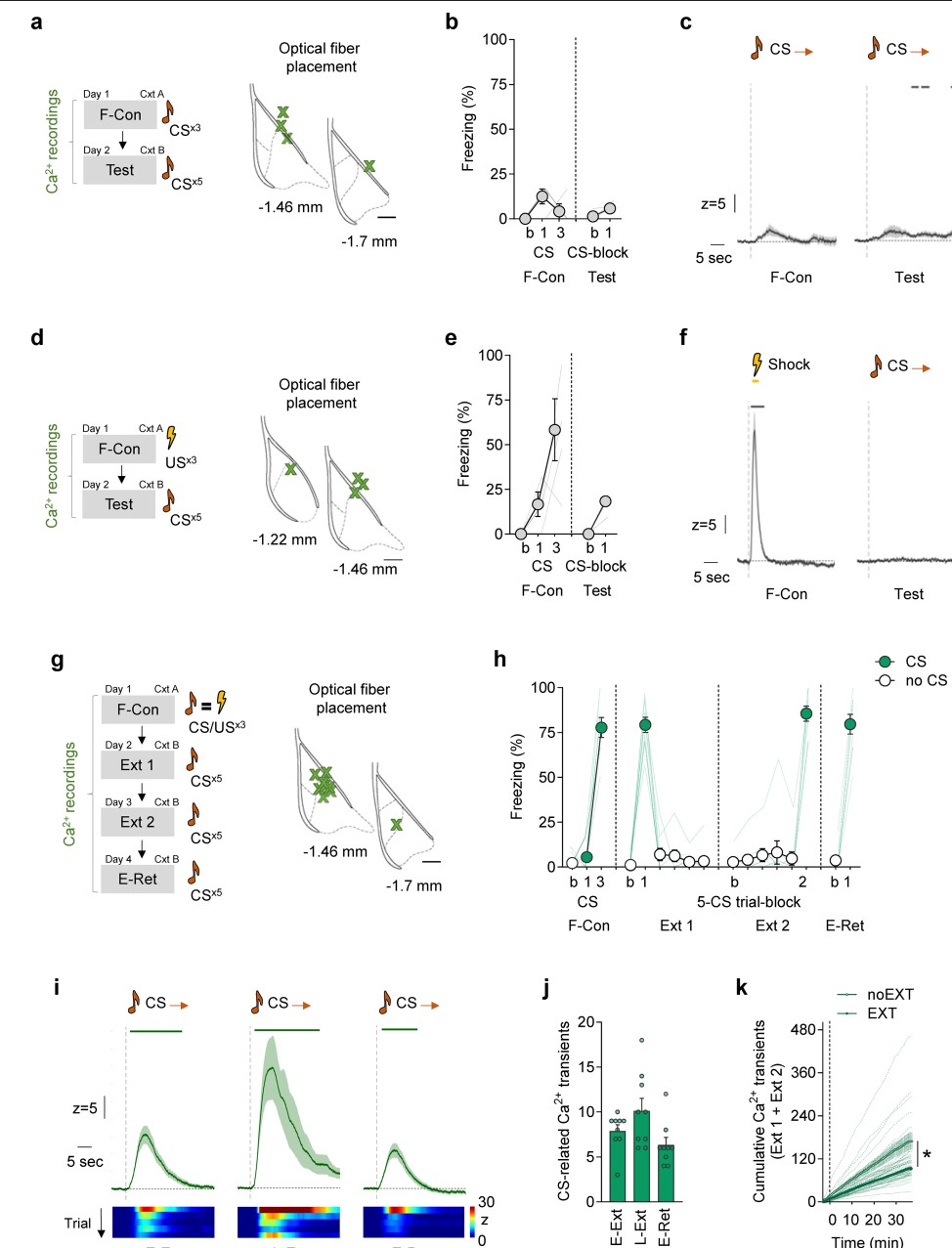

**Extended Data Fig. 2 | Controls for astrocyte Ca²⁺ correlates of fear.**
**a**, Experimental timeline for cyto-GCaMP6f fibre photometry in animals presented with the CS, but no US, during 'fear conditioning' (F-Con), with accompanying fibre tip placement map. **b**, Freezing levels in CS/no-US animals (1-way repeated measures ANOVA and Šídák's *post hoc* tests, $P > 0.05$, n = 4 mice). **c**, Population CS-aligned Ca²⁺ responses in CS/no-US animals (n = 4 mice). **d**, Experimental timeline for cyto-GCaMP6f fibre photometry in animals presented with the US, but no CS, during F-Con, with accompanying fibre tip placement map. **e**, Freezing levels in US/no-CS animals (1-way repeated measures ANOVA and Šídák's *post hoc* tests, $P > 0.05$, n = 4 mice). **f**, Population CS-aligned Ca²⁺ responses in US/no-CS animals (n = 4 mice. **g**, Experimental timeline for cyto-GCaMP6f fibre photometry in non-extinguished animals

presented with the CS during the first and last 5 trials of extinction only, with accompanying fibre tip placement map. **h**, Freezing levels in non-extinguished animals (1-way repeated measures ANOVA and Šídák's *post hoc* tests, $P > 0.05$, n = 9 mice). **i**, Population CS-aligned Ca²⁺ responses in non-extinguished animals, with 5-trial-wise representative heatmap below (n = 9 mice). **j**, Quantification of CS-related Ca²⁺ transients shown in panel **i** (1-way repeated measures ANOVA and Šídák's *post hoc* tests, $P > 0.05$, n = 9 mice). **k**, Cumulative Ca²⁺ activity across extinction (Ext 1 and 2 combined) (2-tailed unpaired *t*-test, *$P < 0.0001$, extinguished n = 17 vs non-extinguished n = 8 mice/session). Horizontal lines above traces in c,f,i denote permutation test-determined significant difference from chance. Fluorescence values z-scored to the 5-s pre-CS or 3-min pre-extinction period. Data mean ± SEM.

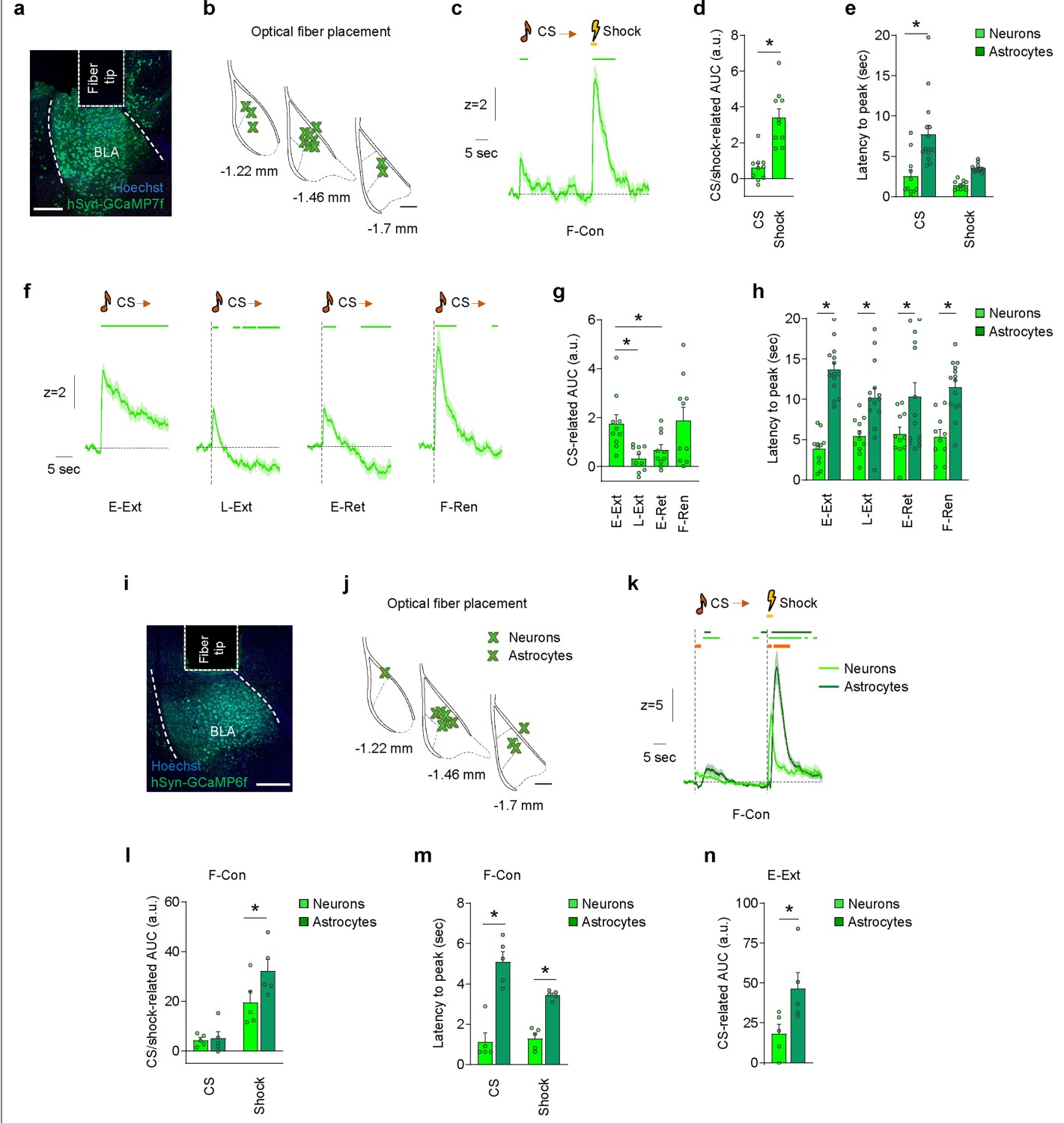

**Extended Data Fig. 3** | See next page for caption.

**Extended Data Fig. 3 | Comparison of astrocyte and neuronal Ca²⁺ correlates. a**, Example optical fibre placement and Ca²⁺ indicator GCaMP7f expression in neurons (hSyn-GCaMP7f) (scale bar: 200 μm). **b**, Optical fibre tip location for in vivo neuronal Ca²⁺ fibre photometry using hSyn-GCaMP7f. **c**, Population CS-related and shock-related Ca²⁺ activity during fear conditioning (F-Con) (n = 10 mice). **d**, AUC of CS- and shock-related Ca²⁺ activity shown in panel **c** (2-tailed paired *t*-test, *P = 0.0002, n = 10–15 mice/group). **e**, Latency to peak CS- and shock-related Ca²⁺ activity in neurons and astrocytes (astrocyte data from photometry experiment shown in Fig. 1a–i) during F-Con (2-way ANOVA and Fisher's LSD *post hoc* tests, during CS *P < 0.0001, n = 10–15 mice/group). **f**, Population CS-related Ca²⁺ activity during E-Ext, L-Ext, E-Ret, and F-Ren (n = 10 mice/stage). **g**, AUC quantification of CS-related Ca²⁺ activity shown in panel **f** (2-way repeated measures ANOVA and Šídák's *post hoc* tests, E-Ext vs L-Ext *P = 0.0495, E-Ext vs E-Ret *P = 0.0378, n = 10 mice/stage). **h**, Latency to peak CS-related Ca²⁺ activity in neurons and astrocytes (astrocyte data from photometry experiment shown in Fig. 1a–i) across stages (2-way ANOVA and Šídák's *post hoc* tests, all stages *P < 0.0001, n = 10–15 mice/group). **i**, Example optical fibre placement and Ca²⁺ indicator GCaMP6f expression in neurons for simultaneous in vivo Ca²⁺ fibre photometry of astrocytes (cyto-GCaMP6f) and neurons (hSyn-GCaMP6f) in opposite hemispheres of the same animals (scale bar: 200 μm). **j**, Optical fibre tip location for simultaneous neuronal and astrocyte fibre photometry. **k**, Population CS-related and shock-related Ca²⁺ activity during F-Con in astrocytes and neurons from simultaneous neuronal and astrocyte fibre photometry (n = 5 mice). **l**, AUC of CS- and shock-related Ca²⁺ activity in astrocytes and neurons shown in panel **k** (2-way repeated measures ANOVA and Fisher's LSD *post hoc* tests, during shock *P = 0.0210, n = 5 mice). **m**, Latency to the peak of CS- or shock-related Ca²⁺ activity in astrocytes and neurons shown in panel **k** (2-way repeated measures ANOVA and Fisher's LSD *post hoc* tests, during CS *P < 0.0001 and shock *P = 0.0006, n = 5 mice). **n**, AUC of CS-related Ca²⁺ activity during E-Ext from simultaneous neuronal and astrocyte fibre photometry (2-tailed paired *t*-test, *P = 0.0252, n = 5 mice). Horizontal lines above traces in c,f,k denote permutation test-determined significant difference from chance for astrocytes (dark green), neurons (light green), or between astrocytes and neurons (orange). Fluorescence values z-scored to the 5-s pre-CS period. Data mean ± SEM.

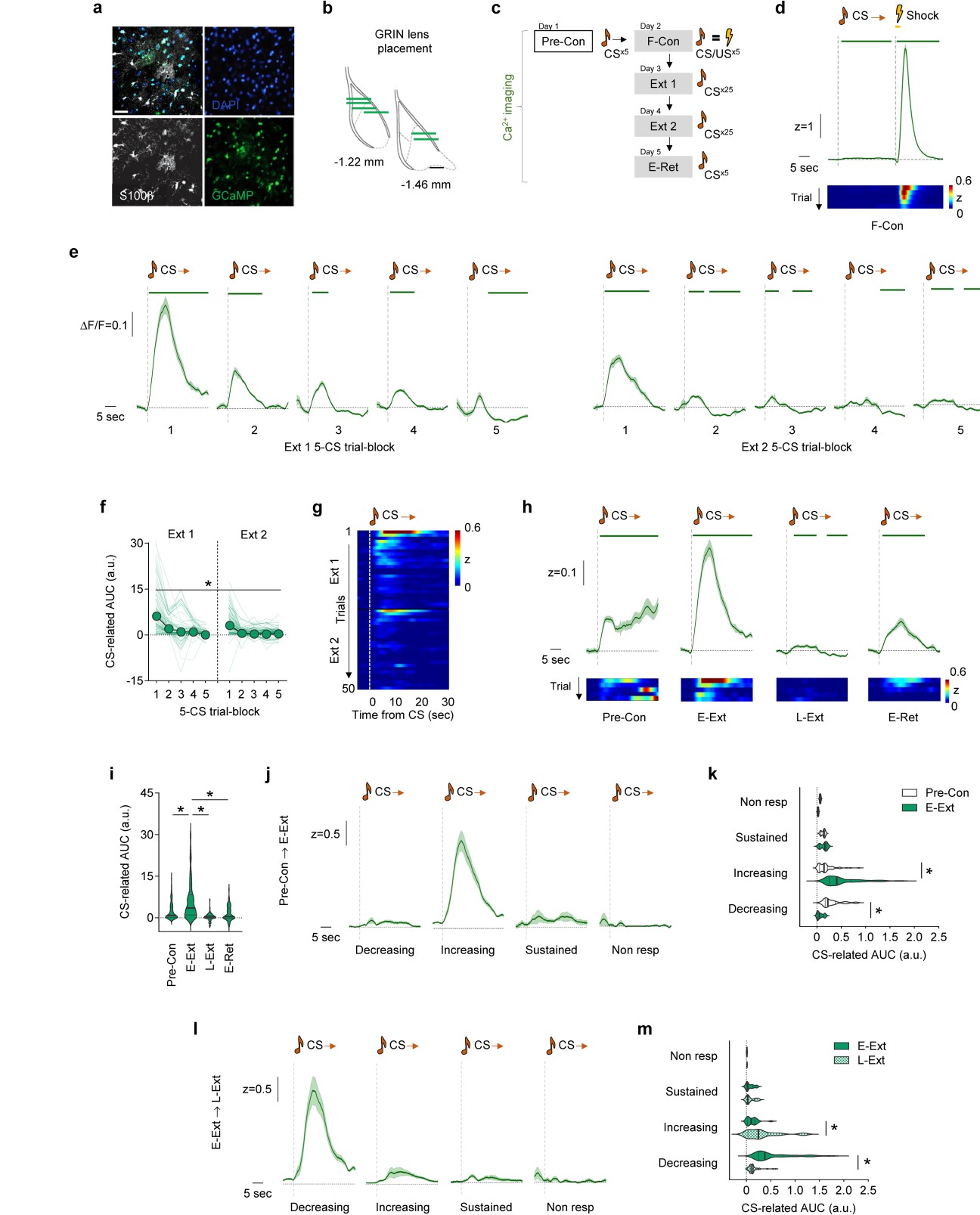

**Extended Data Fig. 4** | See next page for caption.

**Extended Data Fig. 4 | Two-photon imaging of astrocyte Ca²⁺ correlates of fear. a**, Representative GCaMP expression in astrocytes (S100β), scale bar: 50 μm. **b**, GRIN lens location for Glt1-G-CaMP7 in vivo 2-photon Ca²⁺ imaging in astrocytes. **c**, Experimental procedure. **d**, CS-related and shock-related Ca²⁺ activity during fear conditioning (F-Con), with 5-trial-wise cumulative heatmap below (n = 110 events/6 mice). **e**, Traces of CS-related Ca²⁺ activity during each 5-CS trial-block of extinction session 1 (Ext 1) and 2 (Ext 2). **f**, AUC quantification CS-related Ca²⁺ activity during Ext1 and Ext2 shown in panel **e** (1-way repeated measures ANOVA and Šídák's *post hoc* tests, E-Ext (Ext1 block 1) vs L-Ext (Ext2 block 5) *P < 0.0001, n = 110 events/6 mice). **g**, Heatmap of CS-related Ca²⁺ activity across extinction training (n = 110 events/6 mice). **h**, CS-related Ca²⁺ activity during Pre-Con, early extinction (E-Ext, first 5 trials of extinction session 1), late extinction (L-Ext, last 5 trials of extinction session 2), and extinction retrieval (E-Ret), with 5-trial-wise heatmaps below (n = 110 events/6 mice).

**i**, AUC quantification of CS-related Ca²⁺ activity shown in panel **h** (1-way repeated measures ANOVA and Šídák's *post hoc* tests, Pre-Con vs E-Ext, E-Ext vs L-Ext, E-Ext vs E-Ret *P < 0.0001 for all, n = 110 events/6 mice). **j**, CS-related Ca²⁺ activity grouped by events exhibiting decreased, increased, sustained, or no response to the CS from Pre-Con to E-Ext. **k**, AUC quantification of CS-related Ca²⁺ activity shown in panel **j** (2-tailed paired *t*-test, events with decreasing activity *P < 0.0001, increasing activity *P < 0.0001 and non-responsive *P = 0.0155, n = 96 events/6 mice). **l**, CS-related Ca²⁺ activity grouped by events exhibiting decreased, increased, sustained, or no response from E-Ext to L-Ext. **m**, AUC quantification of CS-related Ca²⁺ activity shown in panel **l** (2-tailed paired *t*-test, events with decreasing activity *P < 0.0001, increasing activity *P = 0.0045, n = 80 events/6 mice). Horizontal lines above traces in d,e,h denote permutation test-determined significant difference from chance. Fluorescence values z-scored to the 5-s pre-CS period. Data mean ± SEM.

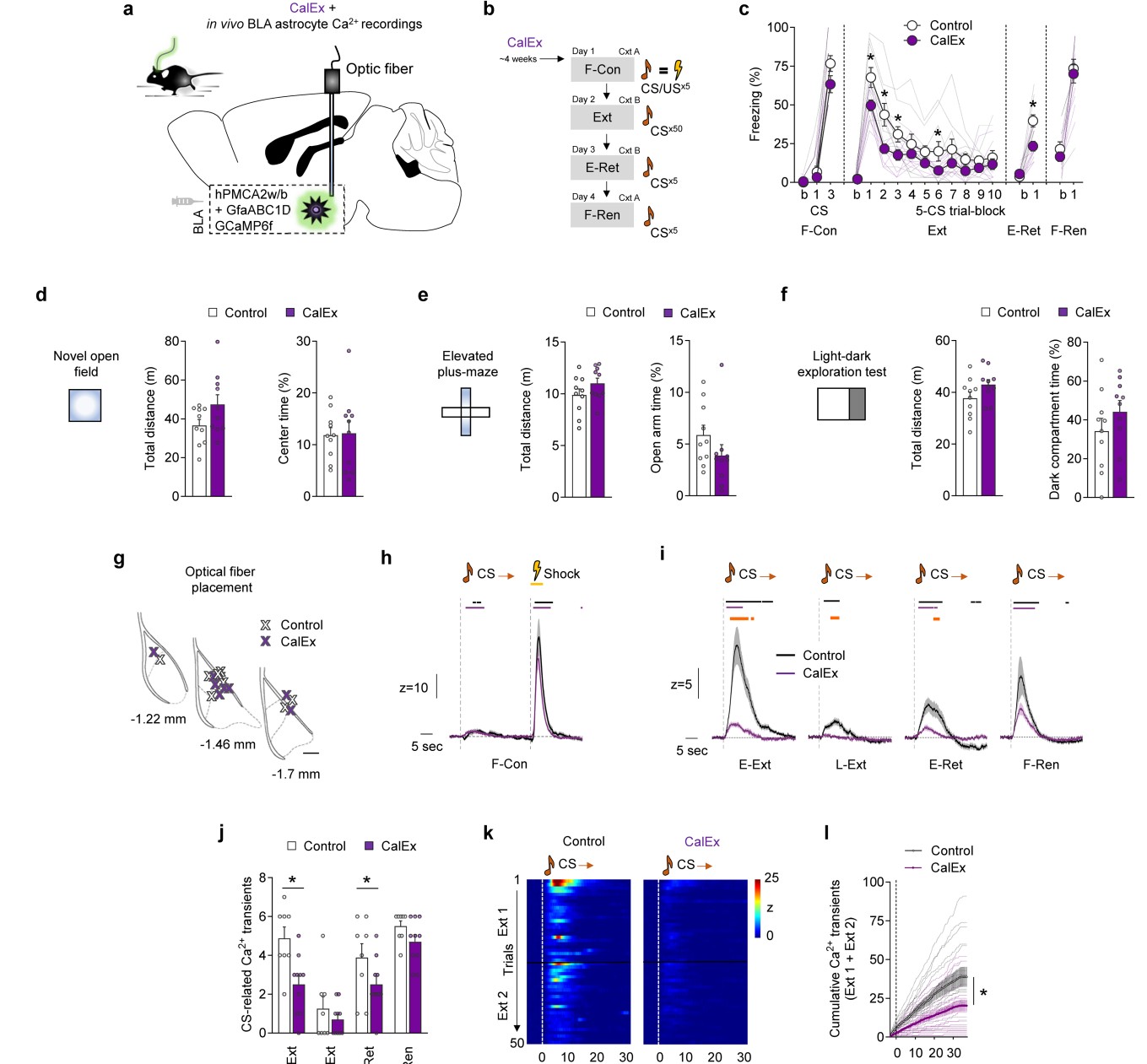

**Extended Data Fig. 5 | Effects of astrocyte Ca²⁺ extruder. a**, cyto-GCaMP6f fibre photometry in animals expressing CalEx. **b**, Experimental timeline. **c**, Effects of CalEx on freezing levels during fear conditioning (F-Con), extinction (Ext), extinction retrieval (E-Ret), and fear renewal (F-Ren) in CalEx or TdTom control virus-expressing animals (2-way repeated measures ANOVA, Šídák's or Fisher's LSD *post hoc* tests, Ext block 1 *P = 0.0024, block 2 *P = 0.0002, block 3 *P = 0.0238, block 6 *P = 0.0364, E-Ret *P = 0.0001, Control n = 10, CalEx n = 10 mice). **d**, Effects of CalEx on novel open field total distance and centre time (2-tailed unpaired *t*-tests, P > 0.05, Control n = 10, CalEx n = 10 mice). **e**, Effects of CalEx on elevated plus-maze total distance and open arm time (2-tailed unpaired *t*-tests, P > 0.05, Control n = 10, CalEx n = 10 mice). **f**, Effects of CalEx on light-dark exploration test total distance and dark compartment time (2-tailed unpaired *t*-tests, P > 0.05, Control n = 10, CalEx n = 10 mice). **g**, Optical fibre tip location for cyto-GCaMP6f in vivo astrocyte Ca²⁺ fibre photometry in CalEx-expressing animals. **h**, Effects of CalEx on CS-related and shock-related Ca²⁺ activity during F-Con as shown by population activity traces (Control n = 8, CalEx n = 10 mice). **i**, Effects of CalEx on CS-related Ca²⁺ activity during Ext, E-Ret and F-Ren as shown by population activity traces (Control n = 8, CalEx n = 10 mice). **j**, Quantification of CS-related Ca²⁺ activity shown in panel **i** (2-way repeated measures ANOVA and Šídák's *post hoc* tests, Control vs CalEx during E-Ext *P = 0.0008 and E-Ret *P = 0.0449, Control n = 8, CalEx n = 10 mice). Heatmaps (**k**) and quantification (**l**) of cumulative Ca²⁺ activity during extinction (Ext 1 and 2 combined) (2-tailed unpaired *t*-test on minute 35, *P = 0.0132, Control n = 8, CalEx n = 10 mice). Horizontal lines above traces in h,i denote permutation test-determined significant difference from chance for CalEx (purple) and Control (black) or significant difference between groups (orange). Fluorescence values z-scored to the 5-s pre-CS or 3-min pre-extinction period. Data mean±SEM.

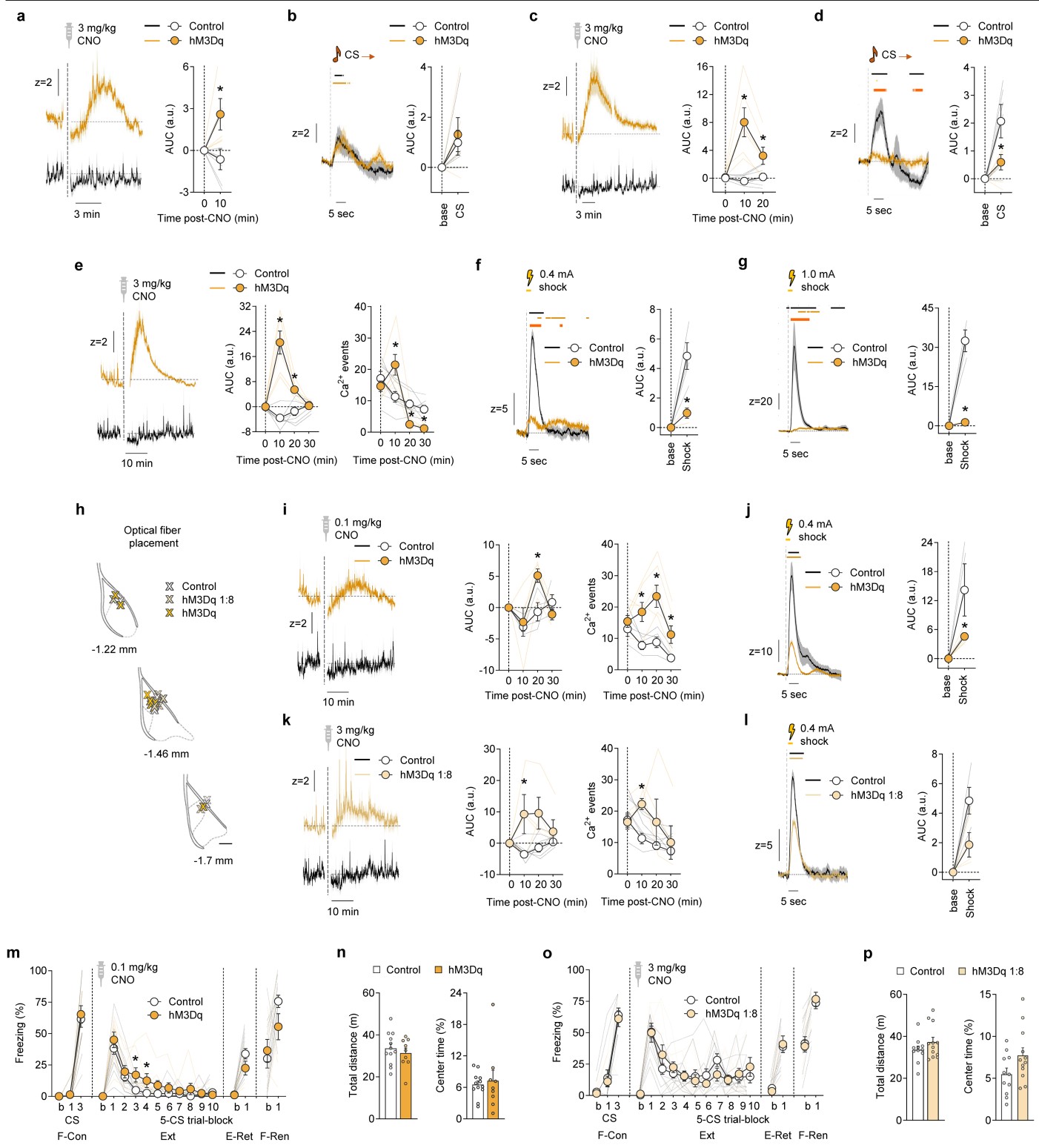

**Extended Data Fig. 6 | See next page for caption.**

**Extended Data Fig. 6 | Chemogenetic manipulation of astrocyte Ca$^{2+}$ via hM3Dq-actuation. a**, In vivo astrocyte Ca$^{2+}$ fibre photometry measuring effects of 3 mg/kg CNO in animals expressing hM3Dq or a control mCherry virus over 10 min post-CNO injection, as shown by population traces (left) and AUC quantification (right) (2-way repeated measures ANOVA and Fisher's LSD *post hoc* tests, *$P$ = 0.0022, Control n = 7, hM3Dq n = 6 mice). **b**, Effects of 3 mg/kg CNO on CS-related astrocyte Ca$^{2+}$ activity tested beginning 3 min after CNO-injection, as shown by population traces (left) and AUC quantification (right) (2-way repeated measures ANOVA and Fisher's LSD *post hoc* test on AUC values, *$P$ < 0.05, Control n = 7, hM3Dq n = 6 mice). **c**, In vivo astrocyte on Ca$^{2+}$ fibre photometry measuring effects of 3 mg/kg CNO in animals expressing hM3Dq or a control mCherry virus over 20 min post-CNO injection, as shown by population traces (left) and AUC quantification (right) (2-way repeated measures ANOVA and Šídák's *post hoc* tests on AUC values, at 10 min *$P$ < 0.0001, 20 min *$P$ = 0.0306, Control n = 7, hM3Dq n = 6 mice). **d**, Effects of 3 mg/kg CNO on CS-related astrocyte Ca$^{2+}$ activity tested 10 min post-CNO injection, as shown by population traces (left) and AUC quantification (right) (2-way repeated measures ANOVA and Fisher's LSD *post hoc* test on AUC values, *$P$ = 0.0069, Control n = 7, hM3Dq n = 6 mice). **e**, In vivo astrocyte Ca$^{2+}$ fibre photometry measuring effects of 3 mg/kg CNO on Ca$^{2+}$ activity in animals expressing hM3Dq or a mCherry control virus over 30 min post-CNO injection, as shown by population traces (left) and AUC quantification (centre) and Ca$^{2+}$ events (right) (2-way repeated measures ANOVA and Šídák's *post hoc* tests for AUC values at 10 min *$P$ < 0.0001, 20 min *$P$ = 0.0013, and Ca$^{2+}$ transients at 10 min *$P$ = 0.0002, 20 min *$P$ = 0.0138, 30 min *$P$ = 0.0200, Control n = 7, hM3Dq n = 6 mice). **f**, Effects of 3 mg/kg CNO on 0.4 mA shock-related Ca$^{2+}$ activity tested 30 min post-CNO injection, as shown by population traces (left) and AUC quantification (right) (2-way repeated measures ANOVA and Fisher's LSD *post hoc* test for AUC values, *$P$ = 0.0006, Control n = 4, hM3Dq n = 3 mice). **g**, Effects of 3 mg/kg CNO on 1.0 mA shock-related Ca$^{2+}$ activity tested 30 min post-CNO injection, as shown by population traces (left) and AUC quantification (right) (2-way repeated measures ANOVA and Fisher's LSD *post hoc* test for AUC values, *$P$ < 0.0001, Control n = 3, hM3Dq n = 3 mice). **h**, Optical fibre tip placement for in vivo astrocyte Ca$^{2+}$ fibre photometry assessment of Ca$^{2+}$ activity in animals expressing a diluted 1:8 concentration of hM3Dq-expressing virus (hM3Dq 1:8), undiluted hM3Dq-expressing virus (hM3Dq), or a mCherry control virus (Control) administered 0.1 or 3 mg/kg CNO. **i**, Effects of a lower (0.1 mg/kg) CNO dose on Ca$^{2+}$ activity in animals expressing hM3Dq or a mCherry control virus over 30 min post-CNO injection, as shown by population traces (left) and AUC quantification (centre) and Ca$^{2+}$ events (right) (2-way repeated measures ANOVA and Šídák's *post hoc* tests for AUC values at 20 min *$P$ = 0.0026 and Ca$^{2+}$ transients at 10 min *$P$ = 0.0058, 20 min *$P$ = 0.0003, 30 min *$P$ = 0.0458, Control n = 4, hM3Dq n = 5 mice). **j**, Effects of 0.1 mg/kg CNO on 0.4 mA shock-related Ca$^{2+}$ activity tested 30 min post-CNO injection, as shown by population traces (left) and AUC quantification (centre) and Ca$^{2+}$ events (right) (2-way repeated measures ANOVA and Fisher's LSD *post hoc* test, *$P$ = 0.0133, Control n = 4, hM3Dq n = 5 mice). **k**, Effects of 3 mg/kg CNO on Ca$^{2+}$ in animals expressing a diluted 1:8 concentration of hM3Dq-expressing virus or a mCherry control virus over 30 min post-CNO injection, as shown by population traces (left) and AUC quantification (centre) and Ca$^{2+}$ events (right) (2-way repeated measures ANOVA and Šídák's *post hoc* tests for AUC values at 10 min *$P$ = 0.0459 and Ca$^{2+}$ transients at 10 min *$P$ = 0.0448, Control n = 4, hM3Dq 1:8 n = 4 mice). **l**, Effects of 3 mg/kg CNO on 0.4 mA shock-related Ca$^{2+}$ activity in animals expressing a diluted 1:8 concentration of hM3Dq-expressing virus or a mCherry control virus tested 30 min post-CNO injection, as shown by population traces (left) and AUC quantification (centre) and Ca$^{2+}$ events (right) (2-way repeated measures ANOVA and Fisher's LSD *post hoc* tests, $P$ > 0.05, Control n = 4, hM3Dq 1:8, n = 4 mice). **m**, Effects of 0.1 mg/kg CNO injected prior to extinction training on freezing levels during F-Con, Ext, E-Ret, and F-Ren in animals expressing hM3Dq or a mCherry control virus (2-way repeated measures ANOVA and Šídák's *post hoc* tests, Ext block 3 *$P$ = 0.0165, block 4 *$P$ = 0.0432, Control n = 12, hM3Dq n = 9 mice). **n**, Effects of 0.1 mg/kg CNO on novel open field total distance and centre time (2 tailed unpaired *t*-tests, $P$ > 0.05, Control n = 12, hM3Dq n = 9 mice). **o**, Effects of 3 mg/kg CNO injected prior to extinction training on freezing levels in animals expressing a diluted 1:8 concentration of hM3Dq-expressing virus or a mCherry control virus (2-way repeated measures ANOVA and Šídák's *post hoc* tests, $P$ > 0.05, Control n = 11, hM3Dq 1:8 n = 12 mice). **p**, Effects of 3 mg/kg CNO on novel open field total distance and centre time in animals expressing a diluted 1:8 concentration of hM3Dq-expressing virus or a mCherry control virus (2-tailed unpaired *t*-tests, $P$ > 0.05, Control n = 11, hM3Dq 1:8 n = 12 mice). Horizontal lines above traces in b,d,f,g,j,l denote permutation test-determined significant difference from chance for hM3Dq (yellow) and Control (black) or significant difference between groups (orange). Fluorescence values were z-scored to either the 30-min or 3-min pre-CNO period, the 5-s pre-CS period, or the 5-s pre-shock period. Data mean ± SEM.

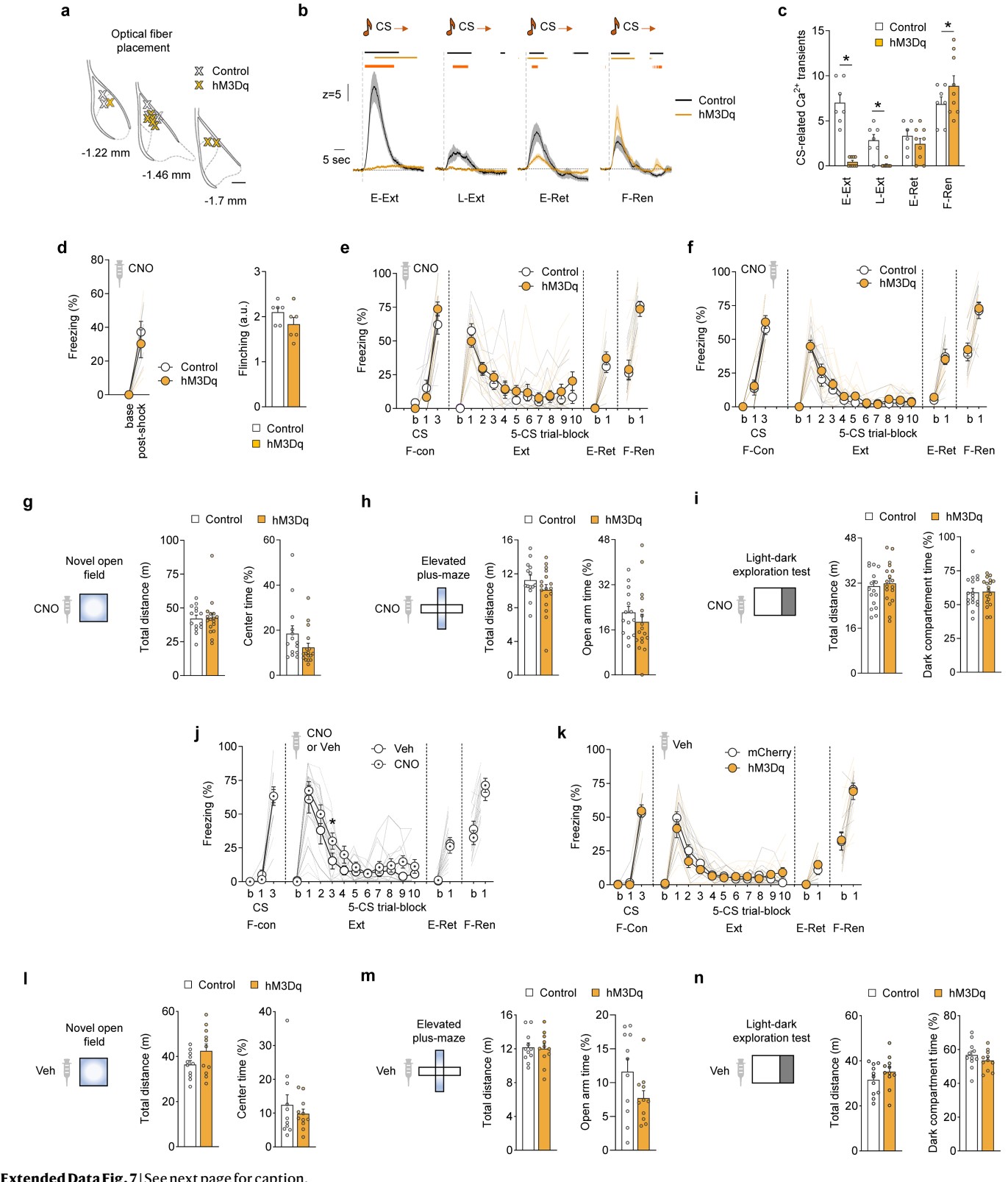

**Extended Data Fig. 7 | See next page for caption.**

**Extended Data Fig. 7 | Behavioural effects of chemogenetic manipulation of astrocyte Ca²⁺ via hM3Dq-actuation. a**, Optical fibre tip placement map for cyto-GCaMP6f in vivo astrocyte Ca²⁺ fibre photometry following pre-extinction CNO in animals expressing hM3Dq or a mCherry control virus. **b**, Population CS-related Ca²⁺ activity during early extinction (E-Ext, first 5 trials of extinction session 1), late extinction (L-Ext, last 5 trials of extinction session 2), extinction retrieval (E-Ret), and fear renewal (F-Ren) (Control n = 7, hM3Dq n = 9 mice). **c**, Quantification of CS-related Ca²⁺ activity shown in panel **b** (2-way repeated measures ANOVA and Šídák's *post hoc* tests, during E-Ext *$P < 0.0001$, L-Ext *$P = 0.0072$, F-Ren *$P = 0.0435$, Control n = 6–7, hM3Dq n = 9 mice/stage). **d**, Effects of pre-test 3 mg/kg CNO on freezing levels (left) at baseline (base) and after exposure to footshock, with associated flinching scores (right), in animals expressing hM3Dq or a mCherry control virus (2-tailed unpaired *t*-test, $P > 0.05$, Control n = 6, hM3Dq n = 6 mice). **e**, Effects of 3 mg/kg CNO prior to fear conditioning on freezing levels during F-Con, Ext, E-Ret, and F-Ren in animals expressing hM3Dq or a mCherry control virus (2-way repeated measures ANOVA, Šídák's or Fisher's LSD *post hoc* tests, $P > 0.05$, Control n = 11, hM3Dq n = 12 mice). **f**, Effects of 3 mg/kg CNO immediately after F-Con on freezing levels during F-Con, Ext, E-Ret, and F-Ren in animals expressing hM3Dq or a mCherry control virus (repeated measures 2-way ANOVA, $P > 0.05$, Šídák's or Fisher's LSD *post hoc* tests, Control n = 11, hM3Dq n = 13 mice). **g**, Effects of 3 mg/kg CNO on novel open field total distance and centre time (2-tailed unpaired *t*-tests, $P > 0.05$, Control n = 14, hM3Dq n = 17 mice). **h**, Effects of 3 mg/kg CNO on elevated plus-maze total distance and open arm time (2-tailed unpaired *t*-tests, $P > 0.05$, Control n = 14, hM3Dq n = 17 mice). **i**, Effects of 3 mg/kg CNO on light-dark exploration test total distance and dark compartment time (2-tailed unpaired *t*-tests, $P > 0.05$, Control n = 17, hM3Dq n = 18 mice). **j**, Effects of pre-extinction 3 mg/kg CNO or Veh on freezing levels during F-Con, Ext, E-Ret, and F-Ren in animals expressing mCherry control virus (2-way repeated measures ANOVA, Šídák's or Fisher's LSD *post hoc* tests, Ext block 3 *$P = 0.0424$, Veh n = 10, CNO n = 10 mice). **k**, Effect of pre-extinction vehicle (Veh) administration on freezing levels during F-Con, Ext, E-Ret, and F-Ren in animals expressing hM3Dq or a mCherry control virus (2-way repeated measures ANOVA, Šídák's or Fisher's LSD *post hoc* tests, $P > 0.05$, Control n = 11, hM3Dq n = 11 mice). **l**, Effects of Veh administration on novel open field total distance and centre time (2-tailed unpaired *t*-tests, $P > 0.05$, Control n = 11, hM3Dq n = 11 mice). **m**, Effects of Veh administration on elevated plus-maze total distance and open arm time (2-tailed unpaired *t*-tests, $P > 0.05$, Control n = 11, hM3Dq n = 11 mice). **n**, Effects of Veh administration on light-dark exploration test total distance and dark compartment time (2-tailed unpaired *t*-tests, $P > 0.05$, Control n = 11, hM3Dq n = 11 mice). Horizontal lines above traces in b denote permutation test-determined significant difference from chance for hM3Dq (yellow) and Control (black) or significant difference between groups (orange). Fluorescence values z-scored to the 5-s pre-CS period. Data mean ± SEM.

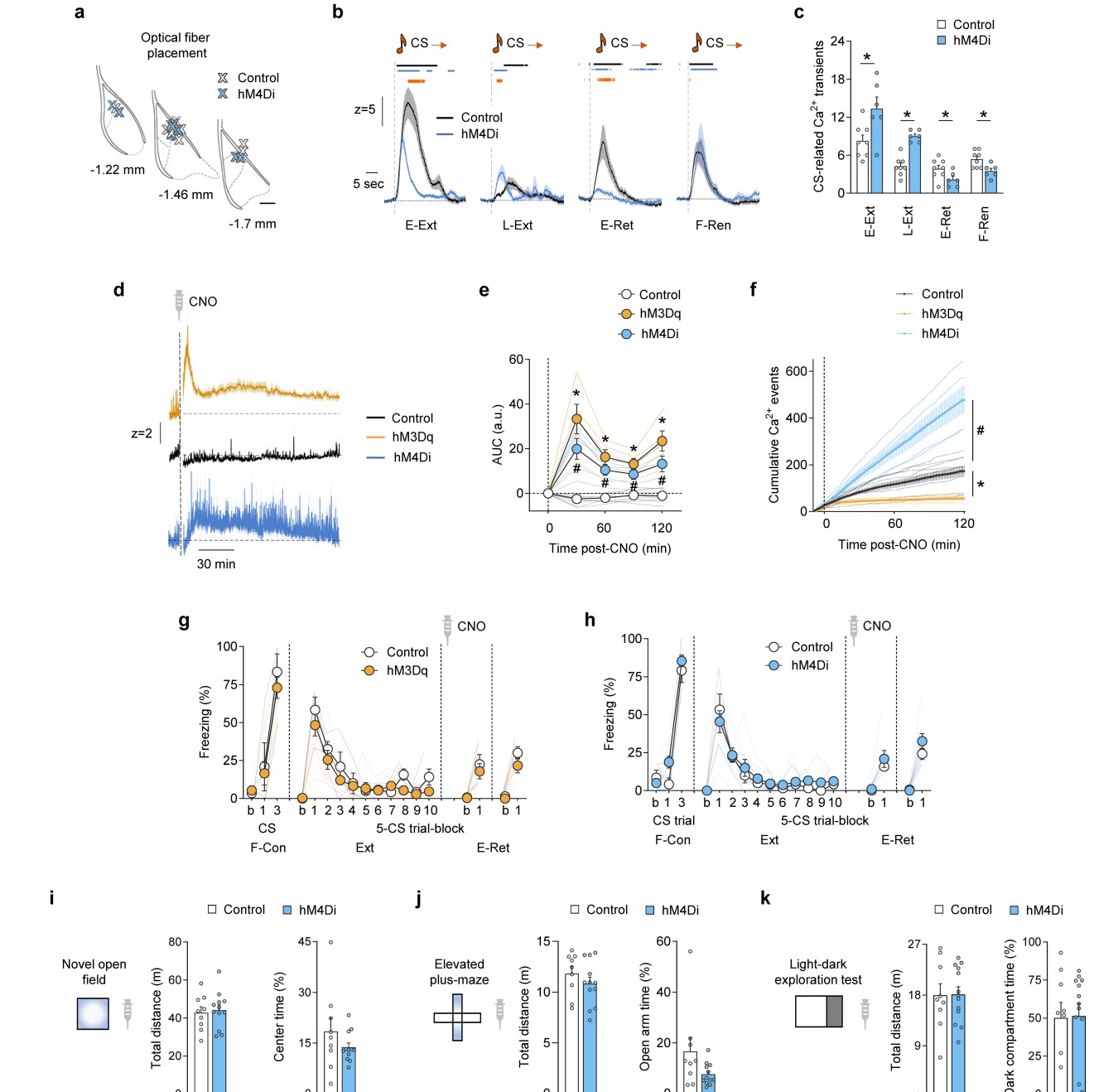

**Extended Data Fig. 8 | Chemogenetic manipulation of astrocyte Ca²⁺ via hM4Di-actuation. a**, Optical fibre tip location for cyto-GCaMP6f in vivo astrocyte Ca²⁺ fibre photometry following pre-extinction CNO in animals expressing hM4Di or a mCherry control virus. **b**, Population CS-related Ca²⁺ activity during early extinction (E-Ext, first 5 trials of extinction session 1), late extinction (L-Ext, last 5 trials of extinction session 2), extinction retrieval (E-Ret), and fear renewal (F-Ren) (Control n = 8, hM4Di n = 6 mice). **c**, Quantification of CS-related Ca²⁺ activity shown in panel **b** (2-way repeated measures ANOVA and Šídák's *post hoc* tests, during E-Ext *P = 0.0421, L-Ext *P < 0.0001, E-Ret *P = 0.0368, F-Ren *P = 0.0115, Control n = 8, hM4Di n = 6 mice). Extended timeline measuring effects of 3 mg/kg CNO on Ca²⁺ activity in animals placed in a neutral-cage, with corresponding quantification of **d**, area under the curve (AUC) **e** (2-way repeated measures ANOVA and Šídák's *post hoc* tests, Control vs hM3Dq at all intervals *P < 0.0001, Control vs hM4Di 30 min #P < 0.0001, 60 min #P = 0.0005, 90 min #P = 0.0093, 120 min #P < 0.0001) and Ca²⁺ events **f** (unpaired *t*-test for transients at minute 120, *P = 0.0025, #P < 0.0001, Control n = 9, hM3Dq n = 5, hM4Di n = 5 mice). **g**, Effects of pre-retrieval 3 mg/kg CNO on

freezing levels during fear conditioning (F-Con), Ext, E-Ret, and F-Ren in hM3Dq or mCherry control virus-expressing animals (2-way repeated measures ANOVA, Šídák's or Fisher's LSD *post hoc* tests, P > 0.05, Control n = 4, hM3Dq n = 8 mice). **h**, Effects of pre-retrieval 3 mg/kg CNO on freezing levels during F-Con, Ext, E-Ret, and F-Ren in animals expressing hM4Di or mCherry control virus (2-way repeated measures ANOVA, Šídák's or Fisher's LSD *post hoc* tests, P > 0.05, Control n = 4, hM4Di n = 8 mice). **i**, Effects of 3 mg/kg CNO on novel open field total distance and centre time (2-tailed unpaired *t*-tests, P > 0.05, Control n = 9, hM4Di n = 12 mice). **j**, Effects of 3 mg/kg CNO on elevated plus-maze total distance and open arm time (2-tailed unpaired *t*-tests, P > 0.05, Control n = 9, hM4Di n = 12 mice). **k**, Effects of 3 mg/kg CNO on light-dark exploration test total distance and dark compartment time (2-tailed unpaired *t*-tests, P > 0.05, Control n = 9, hM4Di n = 13 mice). Horizontal lines above traces in b denote permutation test-determined significant difference from chance for hM4Di (blue) and Control (black) or significant difference between groups (orange). Fluorescence values z-scored to the 30 min pre-CNO or 5-s pre-CS periods. Data mean±SEM.

**a**

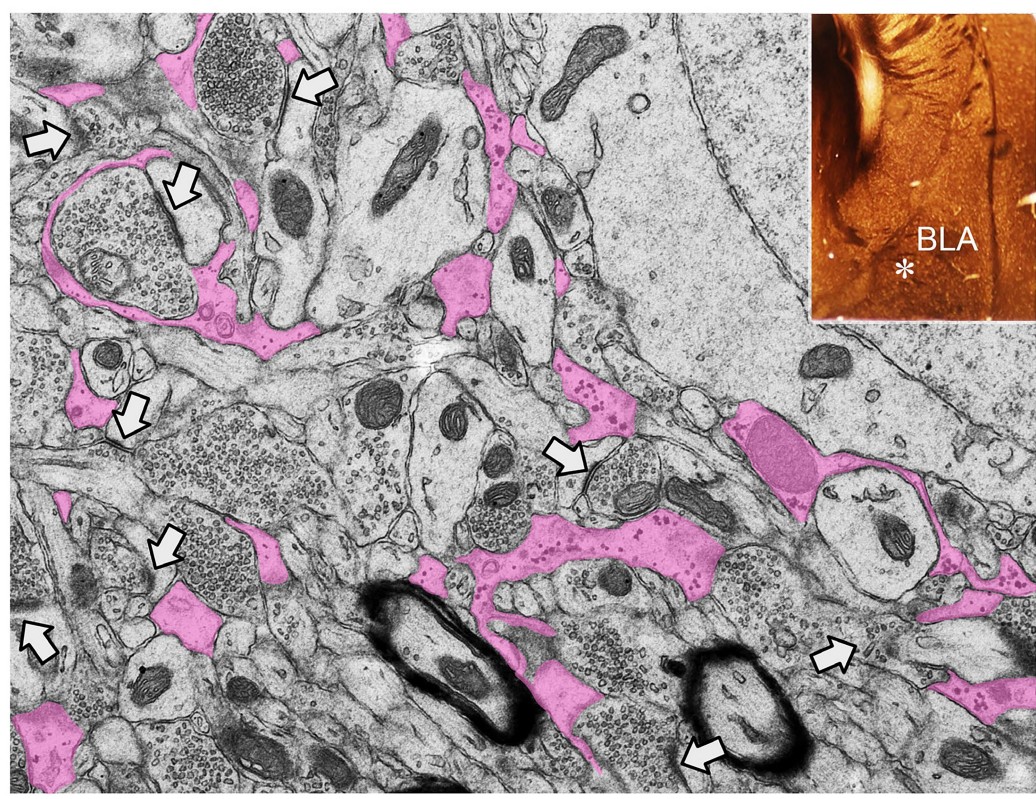

**b**

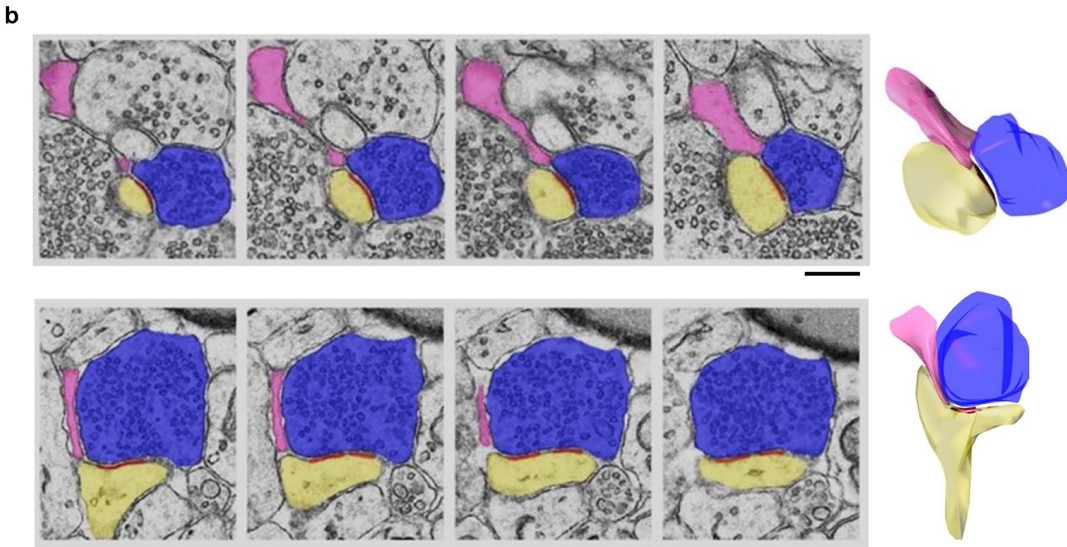

**Extended Data Fig. 9 | Electron microscopy of astrocyte processes at BLA synapses. a**, Electron micrograph of astrocytic processes (pink) distributed through the BLA neuropil. Synapses indicated by arrows. Inset: BLA section after embedding for electron microscopy showing imaging location (denoted by asterisk) (scale bar: 1 µm). **b**, Four serial sections through each of two excitatory synapses (upper and lower panels) in BLA, with corresponding 3D renderings showing dendritic spine (yellow), post-synaptic density (red), axonal bouton (blue), and astrocytic process (pink) (scale bars: 0.25 µm).

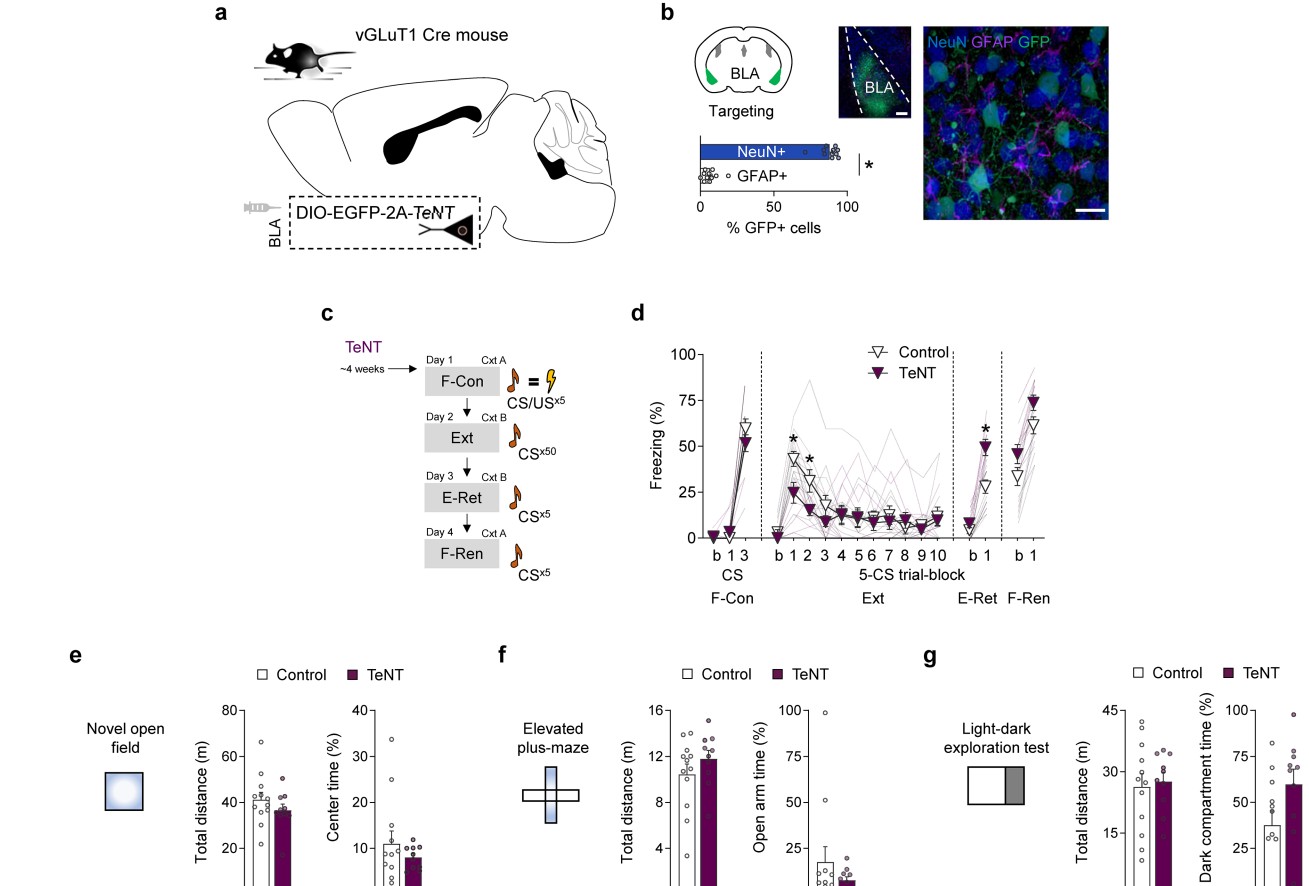

**Extended Data Fig. 10 | Effect of inhibiting neuronal synaptic transmission.** **a**, Inhibition of glutamatergic synaptic transmission via tetanus toxin light chain (TeNT) expression in VGlut1-Cre mice. **b**, Representative TeNT expression in astrocytes (GFAP) and neurons (NeuN) (2-tailed paired *t*-test, $P < 0.0001*$, n = 11 sections/3 mice) (scale bar: 200 μm (left), 10 μm (right)). **c**, Experimental timeline. **d**, Effects of TeNT on freezing levels during fear conditioning (F-Con), extinction (Ext), extinction retrieval (E-Ret), and fear renewal (F-Ren) in Cre+ mice or Cre- controls expressing TeNT virus (2-way repeated measures ANOVA,

Šídák's or Fisher's LSD *post hoc* tests, Ext block 1 *$P = 0.0025$, block 2 *$P = 0.0092$, E-Ret *$P < 0.0001$, Control n = 12, TeNT n = 10 mice). **e**, Effects of TeNT on novel open field total distance and centre time (2-tailed unpaired *t*-tests, $P > 0.05$, Control n = 12, TeNT n = 10 mice). **f**, Effects of TeNT on elevated plus-maze total distance and open arm time (2-tailed unpaired *t*-tests, $P > 0.05$, Control n = 12, TeNT n = 10 mice). **g**, Effects of TeNT on light-dark exploration test total distance and dark compartment time (2-tailed unpaired *t*-tests, $P > 0.05$, Control n = 12, TeNT n = 10 mice).

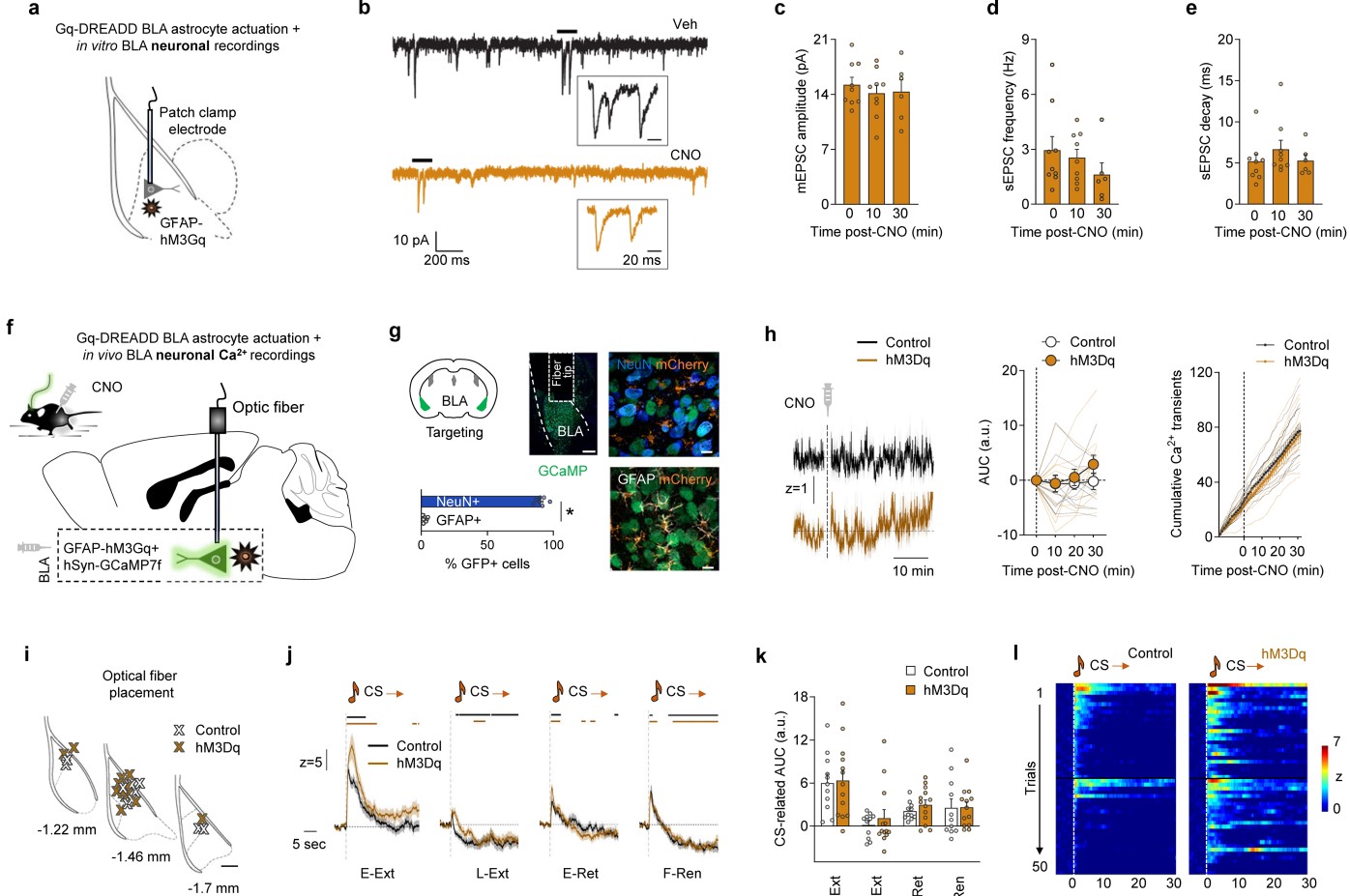

**Extended Data Fig. 11 | Chemogenetic manipulation of astrocyte Ca²⁺ on neuronal activity. a**, In vitro slice electrophysiology recordings following bath application of CNO in animals expressing hM3Dq in astrocytes. **b**, Representative traces of BLA neuronal activity before (Veh) and after (CNO) (insets show periods denoted by black lines). CNO effects on sEPSC amplitude **c**, frequency **d** and decay **e** in BLA neurons (1-way repeated measures ANOVA and Šídák's *post hoc* tests, *P* > 0.05, n = 6–9 cells/7 mice). **f**, Combined in vivo fibre photometry neuronal Ca²⁺ imaging and astrocyte hM3Dq-actuation. **g**, Example optic fibre placement and hSyn-GCaMP7f expression in astrocytes (GFAP) and neurons (NeuN) (2-tailed paired *t*-test, *P* < 0.0001*, n = 7 sections/3 mice) (scale bars: 200 µm (left), 10 µm (right). **h**, In vivo fibre photometry measurement of hM3Dq effects on BLA neuronal Ca²⁺ activity in a neutral-cage in animals expressing hM3Dq or a mCherry control virus in astrocytes, as shown by population activity traces (left), AUC values (centre) and Ca²⁺ transients (right) (2-way repeated measures ANOVA and Šídák's *post hoc* tests,

*P* > 0.05, Control n = 11, hM3Dq n = 13 mice). **i**, Optical fibre tip location for in vivo GCaMP7f neuronal Ca²⁺ fibre photometry following pre-extinction CNO in animals expressing hM3Dq or a mCherry control virus in astrocytes. **j**, Population CS-related neuronal Ca²⁺ activity during early extinction (E-Ext, first 5 trials of extinction session 1), late extinction (L-Ext, last 5 trials of extinction session 2), extinction retrieval (E-Ret), and fear renewal (F-Ren) following 3 mg/kg CNO pre-extinction administration in animals expressing hM3Dq or a mCherry control virus in astrocytes (Control n = 11, hM3Dq n = 13 mice). **k**, AUC quantification of CS-related Ca²⁺ transients shown in panel **j** (2-way repeated measures ANOVA and Šídák's *post hoc* tests, *P* > 0.05, Control n = 11, hM3Dq n = 13 mice). **l**, Heatmaps of cumulative Ca²⁺ activity across the extinction (Control n = 11, hM3Dq n = 13 mice). Horizontal lines above traces in **j** denote permutation test-determined significant difference from chance for hM3Dq (yellow) and Control (black). Fluorescence values z-scored to the 30 min pre-CNO, 3-min pre-extinction, or 5-s pre-CS period. Data mean ± SEM.

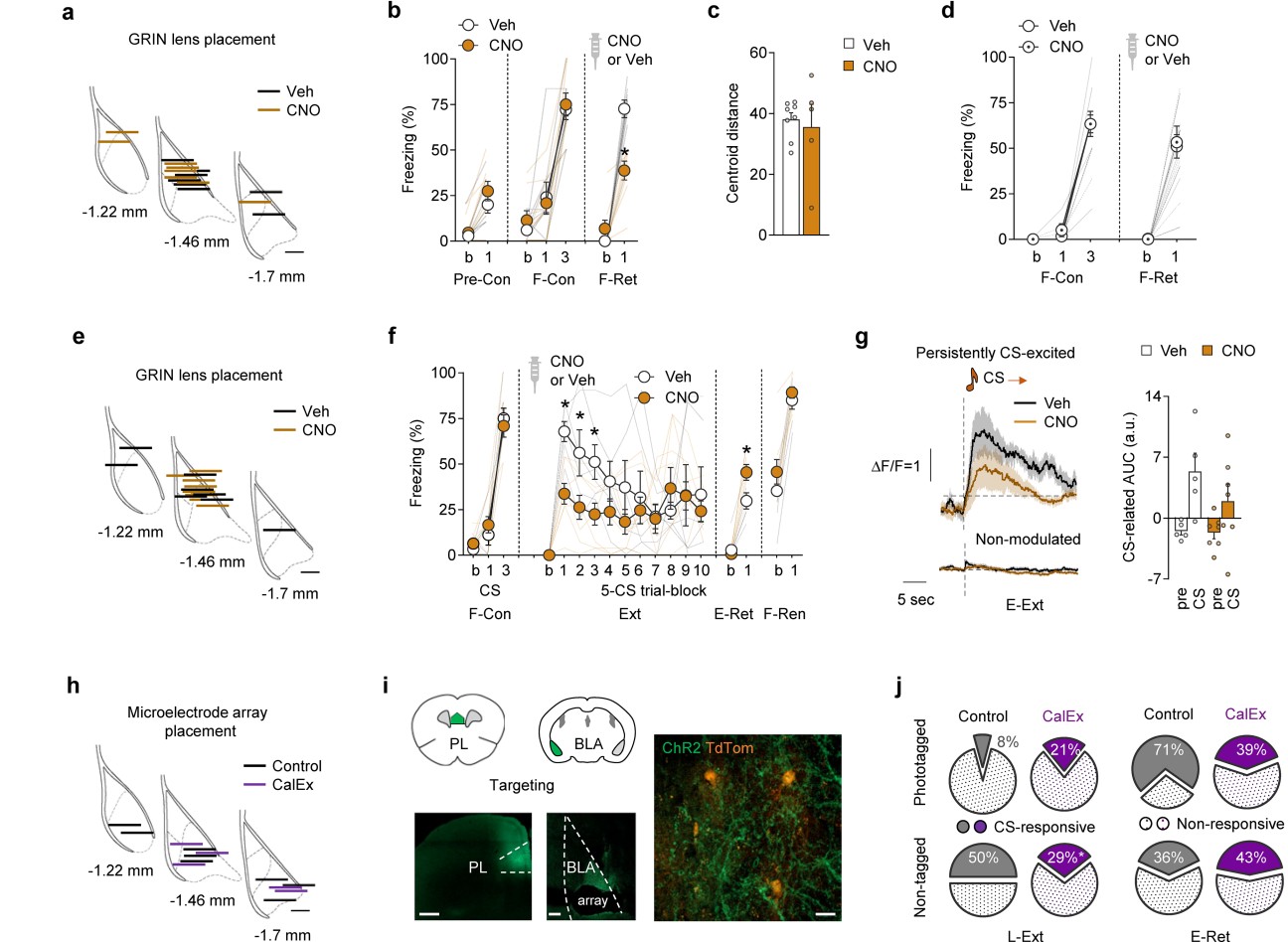

**Extended Data Fig. 12 | Effects of astrocyte Ca²⁺ manipulations on neuronal representations. a**, GRIN lens location for in vivo GCaMP7f neuronal Ca²⁺ imaging in animals expressing hM3Dq in astrocytes, to assess the effects of astrocyte hM3Dq-actuation on neuronal activity at fear retrieval (F-Ret). **b**, Effects of CNO on CS-related freezing levels during pre-conditioning (Pre-Con), fear conditioning (F-Con) and F-Ret in animals undergoing neuronal Ca²⁺ imaging and astrocyte hM3Dq-actuation (2-way repeated measures ANOVA and Fisher's LSD *post hoc* tests, Veh vs CNO during F-Ret: $P < 0.0001$*, Veh n = 8, CNO n = 7 mice). **c**, Distance from centroid of CS-responsive neurons in the imaging FOV (2-tailed unpaired *t*-test, $P > 0.05$, Veh=8, CNO = 5 mice). **d**, Effects of CNO on CS-related freezing levels during F-Con and F-Ret in animals expressing a mCherry control virus (2-way repeated measures ANOVA and Fisher's LSD *post hoc* tests, $P > 0.05$, Veh n = 10, CNO n = 10 mice). **e**, GRIN lens location for in vivo GCaMP7f neuronal Ca²⁺ imaging in animals expressing hM3Dq in astrocytes, to assess the effect of astrocyte actuation on neuronal

activity during extinction. **f**, Effects of pre-extinction 3 mg/kg CNO or Veh on freezing levels during F-Con, Ext, extinction retrieval (E-Ret), and fear renewal (F-Ren) in animals expressing hM3Dq in astrocytes (2-way repeated measures ANOVA, Šídák's or Fisher's LSD *post hoc* tests, Ext block 1 *$P = 0.0008$, block 2 *$P = 0.0197$, block 3 *$P = 0.0253$, E-Ret *$P = 0.0032$, Veh n = 7, CNO n = 7 mice). **g**, Effects of CNO on E-Ret CS positively-modulated neurons during E-Ext (1-tailed unpaired *t*-test, $P > 0.05$, Veh n = 150 neurons/7 mice, CNO n = 126 neurons/7 mice). **h**, Multi-electrode array location for in vivo neuronal recordings in animals expressing CalEx in astrocytes. **i**, Example multi-electrode array placement and ChR2 expression in BLA→PL projecting neurons in CalEx-expressing mice (scale bars: 500 μm (left), 200 μm (middle), 10 μm (right)). **j**, Effect of CalEx on the proportion of CS-responsive neurons during late extinction (L-Ext) and E-Ret (2-sided Chi-squared test, non-tagged neuronal activity at L-Ext *$P = 0.0203$, Control n = 92–94 neurons/stage/8 mice, CalEx n = 61–62 neurons/stage/5 mice). Data mean±SEM.

# Reporting Summary

## Statistics

For all statistical analyses, confirm that the following items are present in the figure legend, table legend, main text, or Methods section.

| n/a | Confirmed | |
|---|---|---|
| ☐ | ☒ | The exact sample size (*n*) for each experimental group/condition, given as a discrete number and unit of measurement |
| ☐ | ☒ | A statement on whether measurements were taken from distinct samples or whether the same sample was measured repeatedly |
| ☐ | ☒ | The statistical test(s) used AND whether they are one- or two-sided<br>*Only common tests should be described solely by name; describe more complex techniques in the Methods section.* |
| ☐ | ☒ | A description of all covariates tested |
| ☐ | ☒ | A description of any assumptions or corrections, such as tests of normality and adjustment for multiple comparisons |
| ☐ | ☒ | A full description of the statistical parameters including central tendency (e.g. means) or other basic estimates (e.g. regression coefficient) AND variation (e.g. standard deviation) or associated estimates of uncertainty (e.g. confidence intervals) |
| ☐ | ☒ | For null hypothesis testing, the test statistic (e.g. *F*, *t*, *r*) with confidence intervals, effect sizes, degrees of freedom and *P* value noted<br>*Give P values as exact values whenever suitable.* |
| ☒ | ☐ | For Bayesian analysis, information on the choice of priors and Markov chain Monte Carlo settings |
| ☒ | ☐ | For hierarchical and complex designs, identification of the appropriate level for tests and full reporting of outcomes |
| ☒ | ☐ | Estimates of effect sizes (e.g. Cohen's *d*, Pearson's *r*), indicating how they were calculated |

*Our web collection on statistics for biologists contains articles on many of the points above.*

## Software and code

Policy information about availability of computer code

| Data collection | Fiber photometry, cellular resolution, in vivo and in vitro electrophysiology data was acquired using Tucker-Davis Technologies Synapse v88 or higher, Inscopix IDAS HD v2.0.0 or higher, Plexon Omniplex1.21 and Cineplex3.5, Molecular Devises pCLAMP11.0 software. Behavioral data was acquired using Med Associates VideoFreeze2.7.0 and MedPC IV, Noldus Ethovision14 software. Images were acquired using Leica Aperio VERSA12.5, Zeiss ZEN3.10, Olympus CellSens Standard 1.15 and VS200 ASW 3.4.1, Nikon NIS-Elements 5.22.x, Keyence BZ-X Viewer software. |
|---|---|
| Data analysis | Data were analyzed using MATLAB2018a or higher, Python3.10 or higher, Inscopix IDPS v1.6, NeuroExplorer v5 , Prism8.1.0 or higher, EzTrack1.3.5, AQuA, Fiji v2.9.0 and ImageJ v1.37 software |

For manuscripts utilizing custom algorithms or software that are central to the research but not yet described in published literature, software must be made available to editors and reviewers. We strongly encourage code deposition in a community repository (e.g. GitHub). See the Nature Portfolio guidelines for submitting code & software for further information.

## Data

Policy information about availability of data

All manuscripts must include a data availability statement. This statement should provide the following information, where applicable:
- Accession codes, unique identifiers, or web links for publicly available datasets
- A description of any restrictions on data availability
- For clinical datasets or third party data, please ensure that the statement adheres to our policy

All source data are available on request to the corresponding author and are also available in Figshare. Custom code and program-provided code in MATLAB, Inscopix and Python software used for analysis, are available upon request from the corresponding author.

# Research involving human participants, their data, or biological material

Policy information about studies with human participants or human data. See also policy information about sex, gender (identity/presentation), and sexual orientation and race, ethnicity and racism.

| | |
|---|---|
| Reporting on sex and gender | N/A |
| Reporting on race, ethnicity, or other socially relevant groupings | N/A |
| Population characteristics | N/A |
| Recruitment | N/A |
| Ethics oversight | *Identify tN/he organization(s) that approved the study protocol.*          N/A |

Note that full information on the approval of the study protocol must also be provided in the manuscript.

# Field-specific reporting

Please select the one below that is the best fit for your research. If you are not sure, read the appropriate sections before making your selection.

☒ Life sciences        ☐ Behavioural & social sciences        ☐ Ecological, evolutionary & environmental sciences

For a reference copy of the document with all sections, see [nature.com/documents/nr-reporting-summary-flat.pdf](http://nature.com/documents/nr-reporting-summary-flat.pdf)

# Life sciences study design

All studies must disclose on these points even when the disclosure is negative.

| | |
|---|---|
| Sample size | Sample size were based on pilot and previously (similar types of) experiments and were not statistically predetermined. Experiments were replicated in multiple subjects and powered to match sample sizes typical of the technique reported in the field, though no formal power analysis was performed a priori. Images shown were selected from multiple independent samples (i.e., different animals) |
| Data exclusions | Mice with freezing scores greater than three standard deviations above the group mean were excluded from the analysis. Mice with absent viral expression or improperly targeted implantations were excluded from analysis. |
| Replication | Experiments were replicated across multi-animal batches (animal N and n numbers for each experiment are provided in the corresponding figure legends and Supplementary Table 1) |
| Randomization | Experiments groups were randomized. |
| Blinding | Experimenters were blinded to group allocation whenever possible. |

# Reporting for specific materials, systems and methods

We require information from authors about some types of materials, experimental systems and methods used in many studies. Here, indicate whether each material, system or method listed is relevant to your study. If you are not sure if a list item applies to your research, read the appropriate section before selecting a response.

## Materials & experimental systems

| n/a | Involved in the study |
|---|---|
| ☐ | ☒ Antibodies |
| ☒ | ☐ Eukaryotic cell lines |
| ☒ | ☐ Palaeontology and archaeology |
| ☐ | ☒ Animals and other organisms |
| ☒ | ☐ Clinical data |
| ☒ | ☐ Dual use research of concern |
| ☒ | ☐ Plants |

## Methods

| n/a | Involved in the study |
|---|---|
| ☒ | ☐ ChIP-seq |
| ☒ | ☐ Flow cytometry |
| ☒ | ☐ MRI-based neuroimaging |

## Antibodies

**Antibodies used**

Anti-GFP (chicken, Abcam catalog # ab13970), anti-TdTom (rat, Kerafast catalog # EST203), anti-DsRed (rabbit, Clontech catalog # 632496), anti-GFAP (guinea pig, Synaptic Systems catalog # 173400), Anti-S100 beta (rabbit, Abcam catalog # ab52642), anti-NeuN (rabbit, Abcam catalog # ab104225), goat anti-rabbit Alexa 405 (catalog # A31556, Life Technologies), goat anti-guinea pig Alexa 555 (catalog # A21435, Life Technologies), goat anti-rat Alexa 555, (catalog # A21434, Life Technologies), goat anti-guinea pig Alexa 647 (catalog # A21450, Life Technologies), goat anti-rabbit Alexa 680 (catalog # A21076, Life Technologies), mouse anti-GFP (catalog # A-11120, lot # 1859591, Invitrogen), anti-DsRed (rabbit, catalog # 632496, lot # 1904182; Takara Bio), biotinylated goat anti-mouse antibody (catalog # 31802, Invitrogen), goat anti-rabbit secondary antibody (catalog # A32731, Invitrogen)

**Validation**

Reactivity with mouse in fluorescent immunohistochemistry was confirmed by the manufacture and by several previous publication (Adamsky et al., 2018; Vaidyanathan et al., 2021; Hagihara et al., 2021)

## Animals and other research organisms

Policy information about studies involving animals; ARRIVE guidelines recommended for reporting animal research, and Sex and Gender in Research

**Laboratory animals**

Mus musculus, males C57/Bl6J (Jakson Lab catalog #00664), Vglut1-Cre (Jakson Lab catalog #023527), and Glt1-G-CaMP7 (RIKEN BioResource Research Center catalog # G7NG817) aged 2-3 months at the time of injection (2-7 months at time of the experiment) were used.

**Wild animals**

This study did not involve wild animals

**Reporting on sex**

Males animals were used in this study.

**Field-collected samples**

This study did not involve samples collected from the field.

**Ethics oversight**

Experimental procedures were approved by the NIAAA and National Institute of Natural Sciences Animal Care and Use Committees and followed the NIH guidelines outlined in 'Using Animals in Intramural Research' and the local Animal Care and Use Committees.

Note that full information on the approval of the study protocol must also be provided in the manuscript.

## Plants

**Seed stocks** — N/A

**Novel plant genotypes** — N/A

**Authentication** — N/A

