## [Peer Review File · Nature]

Astrocytes enable amygdala neural representations supporting memory

Corresponding Author: Dr Andrew Holmes

Version 0:

Reviewer comments:

Referee #1

(Remarks to the Author)

Bukalo et al investigate the functional involvement of basolateral amygdala (BLA) astrocytes in fear learning in mice, finding that astrocytic calcium dynamics correlate with the presentation of threat-predicting stimuli in a fear state dependent manner, as well as demonstrating that astrocyte calcium dynamics are causally involved in fear and extinction behavior and fear memory coding in BLA. The authors use a combination of imaging tools, electrophysiology, chemogenetics and behavioral analysis to show that a) BLA astrocyte Ca²⁺ activity is increased during threat prediction in high fear states but not after fear extinction, b) BLA astrocyte Ca²⁺ activity manipulation affect fear retrieval bidirectionally and perturb fear extinction retrieval, c) manipulations of astrocyte Ca²⁺ activity affect BLA neuronal population representations and neuronal information content and d) that BLA astrocyte Ca²⁺ dynamics are crucial for BLA encoding of freezing and information transfer from BLA to mPFC.

Overall, this is an important study that will be interesting to a wide audience. Converging evidence from many independent experiments reveal the role of BLA astrocytes and their potential impact on neurons in state-dependent threat prediction. However, several major issues remain:

- 1) It is unclear how the astrocyte data should be compared to the neuronal data. The authors chose different behavioral protocols for measurements of astrocytic and neuronal activity patterns, i.e. a fear extinction protocol (astrocytes) vs. a fear retrieval protocol (neurons). In some cases, extinction protocols run for 1-day, in other cases for 2 days. It is thus currently unclear how to reconcile the neuronal and astrocyte data throughout the paper given the difference in behavioral paradigms.
- 2) BLA neurons are known to exhibit plasticity upon fear learning and extinction. Although it is somewhat implied in the abstract and conclusions of the manuscript that astrocytes affect the state-dependent adaptive neuronal representations in fear learning and extinction, it is not measured directly, given that no habituation or pre-learning CS responses were included in the measurements. It is thus unclear which role astrocytes play in BLA neuronal plasticity during fear learning and extinction.
- 3) hM3Dq activation in astrocytes leads to reduced freezing (Fig. 3d). How can it be ruled out that the changes in neuronal CS coding upon hM3Dq activation (Fig. 3i, 3m) are not an indirect effect of changes in behavioral state (i.e. reduced freezing) that in turn affects BLA neuronal activity. I.e. the change in BLA activity is dependent on the behavioral state and not due to a direct impact of astrocytes. It is thus currently not clear how astrocytes exert their function on BLA principal neuron coding. Do the authors propose that this effect is direct (neurochemical, homeostatic, synaptic etc) or indirect, second order via a behavioral state change (modulation of freezing)?
- 4) As the authors report correctly, astrocyte signaling has been previously implicated in memory-related brain function. The authors argue that they revealed a novel mechanism through which astrocytes modulate threat-related memory and BLA coding. However, the mechanism how astrocytes exert their effects on neuronal function in BLA is not addressed (see also

direct vs. indirect effects in Major Point 3) and remains unknown.

Minor:

- 1) In one set of experiments, hM4Di is used to inhibit astrocytes, while other experiments use CalEx. The authors provide a rationale for using CalEx (to avoid stress), but the use of different inhibition methods weakens the comparability of the data, especially considering that the mechanism of hM4Di is unknown.
- 2) Do astrocytes follow neuronal activity or vice versa? A latency analysis (even if just as mean CS / US responses in different animals) could shine light on this.
- 3) Suppl Fig 1e, heat map: the calcium responses over trials during extinction 1 (Ext 1) and fear renewal (F-Ren) exhibit very similar dynamics, i.e. they decay very quickly with increasing numbers of trials. However, the freezing pattern is remarkably stronger and consistent during F-Ren. If BLA astrocyte calcium events are threat predictors, shouldn't the calcium responses in the heat map also be consistent during fear renewal instead of decaying, given that freezing is stable?
- 4) Ca²⁺ transients, e.g. Fig. 2f: The difference between the global Ca²⁺ wave and the quantification of Ca²⁺ transients is not quite clear to the reader, yet essential for the conclusions of the paper as the transients are quantified. It would be important to demonstrate a few examples of Ca²⁺ transient in the control and CNO data.
- 5) In Fig 2c, traces last ca. 30 min. Following an initial Ca²⁺ peak, the signal decays close to baseline. The authors mention that behavioral testing starts after 30 minutes, so it would be important to plot the activity in 2c and 2i beyond 30 min.
- 6) During fear conditioning, it seems that the CS leads to a continuous activation of astrocytes (ED Fig. S3b). During early extinction the CS responses decay during the tone (ED Fig. S3e). Maybe this is a misperception due to different scaling but if not, this difference in the time course of the CS responses during conditioning and early extinction is interesting.
- 7) Fig. 3i,j vs. Fig. 3m. Inhibition of astrocyte Ca²⁺ signaling leads to a lower dimensional state-space of neuronal CS responses on the population level, yet, the ability to decode decreases. This seems counter-intuitive. Would a lower dimensional representation not lead to higher decode-ability?
- 8) Fig. 1b,k: The GFAP, S100b staining is not visible. It would be easier for the reader to judge the data if the channels would be split.
- 9) Fig. 1e,g: The horizontal labels for the CS and US periods seem to be misaligned / broken-up.
- 10) Fig. 1n: ROI 6 appears twice in the ROI map.
- 11) Fig. 2e: A correction for multiple comparisons would seem appropriate for the statistical tests on CS blocks.
- 12) Fig 2f: The Ca²⁺ traces are reported as Ext. 1 and Ext. 2. However, the behavioral data is measured in 1 session. There seems to be a disconnect.
- 13) Fig. 3h: Did the authors perform a statistical test to reach the stated conclusion?
- 14) Fig. 3f: The CS presentation periods are unclear. Error bars / variance bars are missing.
- 15) Fig. 3n: It is currently unclear if the result holds true across animals. A statistical test should be performed to demonstrate that the results are not just driven by 1 mouse.
- 16) The text should be revised for typos and clarity, as well as incorrect referencing of figure panels, e.g. lines 206-207, 209-210, 230, 138, 212.

(Remarks on code availability)

Referee #2

(Remarks to the Author)

Overall, this is a novel and impressive study on the role of astrocyte activity in the amygdala on fear memory coding. They report that astrocytic activity tracks fear responses in the amygdala, and correlates with fear conditioning, extinction, and reinstatement.

The show that bidirectional modulation of astrocyte function during extinction learning modulates fear extinction learning, and memory, in a dissociable manner. They then show the effect of this astrocytic manipulation on neuronal ensemble activity with miniscope recordings. They found that amygdala astrocyte signaling maintains the neuronal representations that correlates with memory recall, as measured by population decoding analysis. Lastly, through impressive multi-electrode array based recording of amygdala-prelimbic cortex projections, the authors show that astrocytic activity in the amygdala modulate CS-induced responses of the prelimbic region.

The study is novel and important. The question is highly relevant to a broad audience. The methods are integrative and appropriate. The experiments are well controlled and the analysis is rigorous. The Figures are well constructed (though in the wrong order, which was confusing....). The study is balanced, very elegant, and concise. The conclusions are very concise, but clear and well justified.

However, I am surprised that the authors have not used their experimental preparation to investigate the role of astrocyte activity in forming memories at the time of training. Indeed, I would also wonder how this process effects the formation of new engrams. Further, it would be interested to test what astrocyte disruption does to existing fear engrams.

This could be important and valuable, because it would test whether the ensemble decoding changes discovered by the authors is in fact functionally important for real engram coding from the perspective of the animal (and not just from the perspective of the investigator).

Lastly, I am not confident that the authors genetic tools are astrocyte specific. This could be better justified and explained.

These are just two broad suggestions for improvement and clarification. The authors may have other improvements they want to prioritize or substitute for the above when considering the other Reviewers' comments.

(Remarks on code availability)

Referee #3

(Remarks to the Author)

This study by Bukalo and colleagues seeks to deepen our understanding of how astrocytes contribute to the processes of memory formation, extinction, and recall in learned threat associations. The research contains many technically impressive and conceptually intriguing aspects. However, I believe the current manuscript lacks clarity in presenting its findings, and its novelty could be enhanced by addressing some unclear aspects in the results.

Below are my main considerations:

1. **Fiberphotometry and Single-Cell Tracking in the BLA:** The authors first examine overall astrocytic activity in the basolateral amygdala (BLA) using fiber photometry, finding a pronounced response to the conditioned stimulus (CS) during early extinction and fear renewal phases, which aligns with elevated freezing behaviors. Subsequently, they apply endoscopic methods to analyze single-cell activity in the same region using the same behavioral protocol. However, a significant opportunity is missed here: additional analyses could provide richer insights into BLA neuronal reorganization during these phases. Specifically, longitudinal tracking of individual cells or examining single-cell responses and contributions to the CS would add depth. Instead, the authors primarily confirm the fiber photometry results without further exploiting these data. For instance, in Fig. 1n, numerous cells appear silent—do they have distinct roles in these dynamics? Do they remain inactive at all times, or are they conditionally activated? A closer analysis here could greatly enhance the paper's novelty by offering more than what is already known about astrocytes' contribution to memory processes.

2. **Chemogenetic Manipulation of BLA Astrocytes:** The authors use chemogenetics to manipulate BLA astrocytes, but I find it hard to be convinced that CNO exposure inactivates the targeted population within 30 minutes. Furthermore, the chemogenetic effect on fiber photometry data is unclear: there are discrepancies between Z-scores (which need clarification within the methods and the text explanation), area under the curve (AUC), and cumulative calcium transients. This raises questions about whether data interpretation might vary depending on the analysis approach. Does the calcium signal uniformly increase or decrease across these methods? What effect is observed beyond the initial 30 minutes? A later testing time post-CNO injection might yield clearer data, particularly concerning effects on extinction retrieval (E-Ret; see next point).

3. **Unresolved E-Ret Dynamics:** The findings related to extinction retrieval (E-Ret) remain ambiguous. Testing E-Ret with specific chemogenetic or optogenetic manipulations could help clarify astrocyte involvement at this time point. Additionally, including E-Ret data in calcium recordings (e.g., Fig. 2f) might improve understanding of astrocytic roles during retrieval.

4. **Differential Effects of Chemogenetic Activation:** Chemogenetic activation appears to exert opposite effects on extinction related to Chemo-inactivation. However, both manipulations lead to an increase in freezing behavior during E-Ret. This intriguing result, however, would benefit from additional analyses to explore its underlying mechanisms. Calcium recordings, once again, could enhance interpretability.

5. **Calcium Recordings Across Extinction Phases:** The calcium data from chemogenetically activated astrocytes reveal a generally flat activation profile across extinction sessions, lacking the modulation or decline observed in vehicle-treated animals. It appears that activity decreases during the initial extinction sessions (Ext 1 and 2) compared to vehicle-treated animals. This suggests that hM4D treatment dampens astrocytic activity rather than simply increasing it, as the authors suggest, which may impair normal extinction dynamics by preventing adaptive modulation of activity. What insights do we gain from this manipulation? Additionally, to better understand the fiber photometry responses and differences from the vehicle group, I recommend showing the zero-value calibration for Z-scores on the heatmap in Fig. 2l.

6. **Analysis of Treated vs. Untreated Mice:** The authors then examine differences between vehicle- and CNO-hM3D-treated mice during fear conditioning and retrieval, employing exciting and persuasive analysis techniques. However, interpreting the significance of comparing baseline astrocyte activity with chemogenetically manipulated activity is challenging, particularly given the widespread activity reduction shown in Fig. 2f. As expected with this reduced activity, the analysis reveals lower dimensionality, fewer units responding to the CS, and overall decreased informational content. This comparison, however, may fall short in illuminating the specific contributions of BLA astrocytes to memory processes. Instead, comparing single-cell activation patterns, dimensionality, and decoding capabilities across stages—such as early extinction, late extinction, retrieval, and renewal—might yield more meaningful insights.

7. **Correlation of Freezing Levels with Decoder Coefficients:** In Fig. 4a-c, the authors suggest a correlation between freezing and decoder coefficients driven by vehicle data. To validate this, they should display separate graphs and statistical analyses for vehicle-only data.

8. **Notable Findings Using Optogenetics and Multi-Electrode Recordings:** The most compelling findings involve optogenetic and multi-electrode recordings, which demonstrate how BLA astrocytes influence projections affecting prelimbic (PL) neuronal dynamics. These results highlight novel insights into how BLA contributes to PL recruitment during extinction and

how optogenetic astrocyte inactivation impacts this process. This approach holds substantial promise for further studies on the memory-related dynamics of this pathway.

In summary, this study shows considerable potential but requires additional data analysis to clarify some unresolved dynamics (e.g., E-Ret) and to integrate single-cell identification. Additionally, BLA-PL dynamics analysis remains preliminary and could benefit from further investigation. I am enthusiastic about this research but believe the current version is still preliminary and requires further analysis and interpretation.

Minor points:

Fig. 2f: If this uses the same protocol as Fig. 2a, consider omitting the redundant drawing to avoid confusion.

Fig. 3n: Can the authors statistically assess group differences and variability?

Fig. 3a: The term "activation" could be misleading; "manipulation," as used in Fig. 2, may be clearer.

(Remarks on code availability)

Version 1:

Reviewer comments:

Referee #1

(Remarks to the Author)

The authors have addressed major issues. A few minor issues remain:

In general: It would be preferable if the authors would choose data graphing methods that reveal the distribution of the data instead of bar plots.

Fig. 1d. Did a post-hoc test reveal significant differences between E-Ext., L-Ext, E-Ret?

Fig. 1f. Label ambiguous. Does the star indicate significant difference between CS and shock? I assume that the CS and US are always recorded in the same animal. Would this not warrant a paired test design?

Fig. 1g. Graphing issues of the green CS bar. It appears discontinuous. Similar general issue in several other figures.

Fig. 1i. Vertical and horizontal error bars are missing on the individual summary data points.

Fig. 1k. The stack bar design deviates from the general bar design of the other figures and is harder to read.

Fig. 1q. It is not clear how the different plasticity types were determined (minimum threshold for change magnitude, statistical test?).

Fig. 2. It is not immediately clear to the reader how the control was setup. Are these mCherry-only expressing mice or was only GCaMP6f expressed in these animals?

Fig. 3a. The microscope is not mentioned in the methods. An experimental timeline similar to 2d would help the reader to understand the experimental design.

Fig. 3b. Is the GCaMP7f image a miniature microscope image or an ex vivo confocal image. This should be clearly labelled in the figure.

Fig. 3b. How many neurons were imaged per mouse / day?

Fig. 3c,f,r: Unless I misunderstood the methods, the baseline of the traces should average around 0. However, this doesn't seem to be the case in the example traces.

Fig. ED 11,a,f: actuation vs. activation

Fig. 3n, comment to Minor Point 19. Using pseudo-populations seems unnecessary given that the authors should be able to record from large numbers of BLA neurons in individual mice. The analysis would benefit from a demonstration of the robustness of the results within animals.

Fig. 3o. The analysis balances for the number of animals. Was the number of neurons also balanced to train the pseudo-population decoders on the same number of neurons pooled from all mice? In general, I would propose to test the decoder accuracy / single neuron coefficients on an animal-by-animal basis and relate the decoding results of each mouse to its behavior.

Fig. 4h. "Moreover, the degree to which CS-responsivity on E-Ext was diminished by CalEx correlated with the reduction in CS-related freezing (Fig. 4h)." It is unclear how the figure panel 4h supports this statement.

Referee #2

(Remarks to the Author)

I have reviewed manuscript with fresh eyes, including carefully going through the other two Reviewers' comments.

Overall, there can be no doubt that the authors have substantially improved their body of work with new experiments, new analyses, and improvements to the manuscript text.

This effort is more than what is more than sufficient for justifying publication in a high impact journal. Many of these experiments represent incremental improvements, but are valuable nonetheless. Some of the experiments are quite substantial in impact, owing partly to the suggestions of the other Reviewers' comments (but not mine). I believe the core messages from the initial submission were sufficiently novel and robust enough to be published in a high impact journal, but the new additions give the manuscript and extra edge.

However, on an aesthetic level I believe the original arrangement of the main Figures was better, and far more easy to read and comprehend. I believe this matters to scientific dissemination, and is certainly more important than the "we did a lot of work in review" maxim. I think that some of the content placement makes sense in review, but less to a reader of the published version. However, this is for the Editors and Authors to decide.

The Extended Figures are only stored as single files rather than in the Supplementary Information, which was a little irritating in review.

The authors have satisfactorily answered both of my major points during the first round of review. I have no further comments there. I emphasize that I am not working in astrocyte biology and I have no particular stance in this sub-field, but I believe the authors have done an excellent job and I don't think this manuscript should be restrained from publication any further. However, I would like to see a couple of sentences in the Discussion on how engram specificity might be modulated by astrocytic activity.

Reading through the other Reviewers' comments, here are my further comments.

Reviewer 1:

Reviewer 1 Point 1 is an fair parametric criticism. The authors have engaged with it extensively through these two experiments. I particularly admire the bihemispheric approach, and I think this makes a strong impact on the reader.

Reviewer 1 Point 2 seems to be the most valuable criticism of the entire review process. The authors engage with this point extremely well, thereby also somewhat nullifying one of my own comments. The authors show that astrocytic activity is necessary for the learning-induced CS-specificity of BLA neurons. The new experiments in Fig. 3p-r are very welcome to this effect, as they show clear and robust evidence that astrocytic activity is crucial for extinction learning via BLA ensemble responsiveness. Fig. 3r reads a bit like cherry picking, but I don't think that is a major problem here.

Reviewer 1 Point 3 is an important scientific point, and the authors go to a large effort to engage with it. Indeed, some crucial new experiments have been added. I believe the TeNT experiment is very important. However, I also think that one could argue that these synaptic inputs are simply necessary for indirect effects that behavioral changes could have on ensemble plasticity. Nevertheless, I emphasize that there is no definitive experiment that can fully answer this criticism, and indeed the criticism itself assumes a scenario that while parametrically possible, does not have a plausible mechanistic basis itself. In other words, the authors have a stronger theoretical basis for their interpretation than the Reviewer, and indeed I would be very surprised if differences in freezing behavior would cause suggest a robust plasticity effect. I think the solution here is for the authors to simply state this caveat in the Discussion.

Reviewer 1 Point 4 is also important. It is difficult to resolve these issues in such a concise publishing format, but the authors' changes make good sense. I believe the authors have identified a strong 'mechanism' at an ensemble level. There may well be enabling molecular mechanisms at the level of astrocyte/neuron interactions, but that is a different question. Further, it would be inappropriate to investigate a molecular mechanism of this phenomenon in tandem with such detailed and frontier analysis of BLA neuronal activity and coding. I believe it makes much more sense for a study that is already highly integrative (in terms of methods) to focus on one crucial level of analysis to explain behavior.

Reviewer 1 Minor Points are all well managed, and I think Extended Data Fig. S8f is particularly important.

Reviewer 3:

Reviewer 3 Point 1 is well addressed by the 2-P analysis and this enhances the manuscript.

Reviewer 3 Point 2 is clearly in part a communication issue, but the new experiments are welcome. The time courses of CNO effect are, in my opinion, very valuable. The authors explanations for their FP analysis is clear and reasonable.

Reviewer 3 Point 3 is a fair criticism, and the authors respond with a clear argument based on new and previously existing data. The lack of acute effect on extinction memory persistence is clear. However, if the authors wanted to push this further they could of course look at reinstatement, but I think is well outside of the current scope.

Reviewer 3 Point 4 is perhaps expecting more symmetry than one can reasonably expect from biological systems. However, the Reviewer is fair to point it out. The authors experiments and interpretation adequately deal with this criticism overall. I think the authors deserve credit for showing the behavioral effects as they are, and not trying to force them to fit a rigid structure.

Reviewer 3 Point 5 This is an insightful criticism, and a point of caution that I missed. But the authors response and analysis is a clear counter to this view.

Reviewer 3 Point 6 There seems to be have been a communication issue here, but the authors' response is more than appropriate, and I believe sufficient.

Reviewer 3 Point 7 Minor but valuable addition to the manuscript.

Reviewer 3 Point 8 Fully agree.

Reviewer 3 Point 9 I agree, and the authors have substantially improved their thesis.

Reviewer 3 Minor Points are all well managed.

- end of review -

Referee #3

(Remarks to the Author)

The authors have adequately addressed all of my previous concerns. They have improved the manuscript significantly through the addition of new experiments and analyses, as well as by providing clear and satisfactory explanations where necessary. I am satisfied with the revisions and find the current version of the manuscript suitable for publication.

Referee #4

(Remarks to the Author)

Version 2:

Reviewer comments:

Referee #1

(Remarks to the Author)

I thank the authors for the revised version of the manuscript. All my remaining minor points were addressed and clarified in the rebuttal letter and manuscript.

Andrew Holmes, PhD
Chief, Laboratory of Behavioral and Genomic Neuroscience
National Institute on Alcohol Abuse and Alcoholism
National Institutes of Health
5625 Fishers Lane, 2N09, Rockville MD 20592
Telephone: +1 301 402 3519

July 8, 2025

[REDACTION]

Based on the highly positive evaluation of our initial submission, we are delighted to submit a substantially revised and expanded version of manuscript entitled '*Astrocytes enable amygdala neural representations supporting memory*,' by Bukalo et al. The revised manuscript is greatly improved as a result of the review process and we would like to thank you and the Referees for the positive assessment of the work, remarking that it is 'an important study' and 'interesting to a wide audience' (Referee 1), 'novel and impressive study' and 'well controlled and...rigorous' (Referee 2), and 'technically impressive and conceptually intriguing aspects' (Referee 3). In the spirit of the Referees' constructive comments and requests, we have endeavored to faithfully address each of them with substantive new experimentation and analyses, as well as modifications to the text of the manuscript, which we detail below.

Referee 1

Bukalo et al investigate the functional involvement of basolateral amygdala (BLA) astrocytes in fear learning in mice, finding that astrocytic calcium dynamics correlate with the presentation of threat-predicting stimuli in a fear state dependent manner, as well as demonstrating that astrocyte calcium dynamics are causally involved in fear and extinction behavior and fear memory coding in BLA. The authors use a combination of imaging tools, electrophysiology, chemogenetics and behavioral analysis to show that a) BLA astrocyte Ca²⁺ activity is increased during threat prediction in high fear states but not after fear extinction, b) BLA astrocyte Ca²⁺ activity

manipulation affect fear retrieval bidirectionally and perturb fear extinction retrieval, c) manipulations of astrocyte Ca²⁺ activity affect BLA neuronal population representations and neuronal information content and d) that BLA astrocyte Ca²⁺ dynamics are crucial for BLA encoding of freezing and information transfer from BLA to mPFC. Overall, this is an important study that will be interesting to a wide audience. Converging evidence from many independent experiments reveal the role of BLA astrocytes and their potential impact on neurons in state-dependent threat prediction. However, several major issues remain.

Response: We thank the Referee for considering our findings ‘*an important study that will be interesting to a wide audience*’ and for recognizing the ‘*converging evidence from many independent experiments.*’ We have addressed the Referee’s issues with additional experimental work and analyses, along with revisions to the manuscript text and figures. The study is significantly improved as a result.

1. It is unclear how the astrocyte data should be compared to the neuronal data. The authors chose different behavioral protocols for measurements of astrocytic and neuronal activity patterns, i.e. a fear extinction protocol (astrocytes) vs. a fear retrieval protocol (neurons). In some cases, extinction protocols run for 1-day, in other cases for 2 days. It is thus currently unclear how to reconcile the neuronal and astrocyte data throughout the paper given the difference in behavioral paradigms.

Response: We used the 2-day extinction protocol for astrocyte Ca²⁺ fiber photometry experiments (only) based on our concerns that slow bulk CS-evoked responses might not have sufficient time to return to baseline between CS presentations. Hence, we used a longer (60-second) inter-CS interval and, as a result, split extinction training into 2 daily sessions to avoid excessively lengthy sessions and the potential for photobleaching and mouse (cable) fatigue. In our 2-photon astrocyte Ca²⁺ imaging experiment (wherein we did not have the same concerns as with photometry), a 1-day extinction protocol was indeed used, and here we found the same pattern of change in Ca²⁺ activity across test-phases as we had found with the 2-day fiber photometry experiment.

To further address the Referee’s question about the comparability of the astrocyte and neuron protocols, we have performed 2 new experiments. In the first experiment, we performed fiber photometry in which Ca²⁺ responses were measured in neurons (via hSyn-GCaMP7f) using the same 2-day fear extinction protocol we used for astrocytes. This experiment showed that stage-

wise patterns of activity change were largely similar in neurons as we had seen in astrocytes (e.g., increased CS-related activity with conditioning, that subsequently decreases with extinction), but with comparatively slower-to-peak and smaller overall responses in neurons than in astrocytes (hence, these results also address the Referee’s question about stimulus-response latencies in astrocytes versus neurons in point #6 below). These new data are now shown in Extended Data Fig. 3a-h, and discussed in the revised manuscript as follows: ‘*Additionally, while we observed similar stage-related Ca^{2+} activity changes in neurons (Extended Data Fig. 3a-h)...*’

In the second experiment, we performed photometry in which Ca^{2+} responses were simultaneously measured in BLA astrocytes and neurons of the same animals. Specifically, we expressed the Ca^{2+} indicator GCaMP6f in neurons of one hemisphere (hSyn-GCaMP6f) and in astrocytes of the other hemisphere (GfaABC1D-cyto-GCaMP6f) and performed *in vivo* fiber photometry. This strategy allowed for a direct comparison of stimulus-related CS-related Ca^{2+} responses in the two cell types and demonstrated similar activity patterns across test-phases in the two cell-types, but (as in the aforementioned experiment) with comparatively larger and slower responses in astrocytes. These new data are now shown in Fig. 1j,k (for E-Ext, and pasted below) and Extended Data Fig. 3i-n, and discussed in the revised manuscript as follows: ‘*...simultaneous photometry performed in BLA astrocytes and neurons in different hemispheres of the same animals indicated comparatively slower-to-peak, but longer lasting, US and CS-related Ca^{2+} responses in astrocytes (Fig. 1j,k and Extended Data Fig. 3e-h).*’

2. BLA neurons are known to exhibit plasticity upon fear learning and extinction. Although it is somewhat implied in the abstract and conclusions of the manuscript that astrocytes affect the state-dependent adaptive neuronal representations in fear learning and extinction, it is not measured directly, given that no habituation or pre-learning CS responses were included in the measurements. It is thus unclear which role astrocytes play in BLA neuronal plasticity during fear learning and extinction.

Response: To address this point, we have added two new datasets related to neuronal plasticity. First, in our original 1P neuronal imaging experiment, we had obtained neuronal measurements of CS responses during an habituation/pre-conditioning (Pre-Con) test (1 day prior to conditioning, in the absence of CNO) but did not analyze these at the time of initial submission. We have now analyzed these data and used them to define neurons based on whether they exhibit an increase, decrease or no change in CS-responsivity as a function of conditioning (i.e., from Pre-Con to fear retrieval (F-Ret)). We now show that our chemogenetic manipulation of astrocyte Ca^{2+} activity during F-Ret (via CNO in hM3Dq-expressing mice) attenuates (relative to Veh controls) the CS-responsivity of neurons exhibiting increased CS-responsivity due to conditioning. These data provide further evidence that astrocytes are crucial to the expression of learning-related plastic changes occurring in BLA neurons and we thank the Referee for directing us to analyze these data. These new findings are now added as Fig. 3c,d (also pasted below) and discussed in the revised manuscript as follows: ‘*First, we tested whether chemogenetically disrupting astrocyte Ca^{2+} activity (via CNO-injection prior to fear memory retrieval (F-Ret) affected changes in CS-related neuronal activity occurring as a result of conditioning. On defining neurons based on whether they exhibited an increase (conditionally CS-excited), decrease or no change in CS-responsivity from Pre-Con to F-Ret, we found that hM3Dq-actuation had fewer conditionally CS-excited neurons than Veh-injected hM3Dq-expressing controls on F-Ret (Fig. 3c,d). These data indicate a role for astrocytes in enabling BLA neurons to express conditioning-related changes in CS-related activity.*’

Second, we have added another new neuronal 1P imaging experiment in which CS-responses were measured across fear retrieval and extinction after chemogenetically manipulating astrocyte Ca^{2+} activity via pre-extinction training CNO versus Veh in hM3Dq-expressing mice. Using unbiased K-means clustering, we identified a subpopulation of neurons exhibiting a CS-related increase (CS-excited) in activity on extinction retrieval that was significantly blunted in the CNO group – thereby providing a correlate of the CNO-induced deficit in extinction retrieval. Of further interest, we found that this subpopulation was CS-excited across the earlier (fear retrieval and extinction training) test-stages. Taken together these observations reveal how disrupting astrocyte activity perturbs extinction-related adaptations in a subset of neurons that exhibit persistent CS-encoding, possibly reflecting signaling of CS salience. We thank the Referee for prompting us to add these new findings, which are added as Fig. 3p-r (also pasted below) and discussed in the revised manuscript as follows: ‘*Next, we sought to connect the aberrant BLA neuronal encoding at fear memory retrieval caused by astrocyte hM3Dq-actuation with the extinction impairment produced by this manipulation (see Fig. 2e and Extended Data Fig. 12f,g). To do so, we performed another microendoscopic neuronal imaging experiment in which astrocytes were hM3Dq-actuated via CNO injection during fear retrieval/extinction training (but not again before E-Ret) and examined neuronal activity across test-stages. Using unbiased K-means clustering to identify neurons that were CS-responsive on E-Ret, but differentially so as a function of treatment, we identified a subpopulation that was CS-excited on E-Ret but to a significantly lesser extent in the CNO group than Veh controls (Fig. 3p). This relative loss of neuronal activity E-Ret suggests that disrupting astrocyte Ca^{2+} activity impaired important plastic changes in neuronal CS-encoding occurring during extinction memory formation.*

This finding led us to further characterize the subpopulation of neurons affected by astrocyte hM3Dq-actuated by retrospectively tracking their activity to the earlier test-stages. Interestingly, we found that these neurons were more likely to be CS-responsive during E-Ext and L-Ext, although again somewhat less so in the CNO group, than neurons that were CS non-modulated on E-Ret (Fig. 3q and Extended Data Fig. 12h). Thus, disrupting astrocyte Ca^{2+} activity

attenuates the activity of a subset of neurons that are persistently CS-excited on fear retrieval through extinction. While the function of these neurons is unknown, they could potentially encode the salience or other associative properties of the CS that are important for extinction memories to be effectively formed and retrieved; properties which, our data suggest, are improperly signaled when astrocyte Ca^{2+} activity is disrupted.'

A last piece of new data relevant to the question of astrocytic effects on extinction demonstrates that chemogenetically manipulating astrocytes specifically during E-Ret does not impact extinction retrieval (see response to Referee’s 3 point #3 below). Hence, the primary role of BLA astrocytes appears to be to enable extinction plasticity and the underlying adaptations in neuronal encoding, but not the retrieval of an already-formed extinction memory.

3. hM3Dq activation in astrocytes leads to reduced freezing (Fig. 3d). How can it be ruled out that the changes in neuronal CS coding upon hM3Dq activation (Fig. 3i, 3m) are not an indirect effect of changes in behavioral state (i.e. reduced freezing) that in turn affects BLA neuronal activity. I.e. the change in BLA activity is dependent on the behavioral state and not due to a direct impact of astrocytes. It is thus currently not clear how astrocytes exert their function on BLA principal neuron coding. Do the authors propose that this effect is direct (neurochemical, homeostatic, synaptic etc) or indirect, second order via a behavioral state change (modulation of freezing)?

Response: We propose that abnormal neuron coding produced by hM3Dq activation in astrocytes results from a direct functional alteration in astrocyte-neuron interactions and is not merely a product of reduced freezing. We show that BLA astrocyte processes are present in the perisynaptic space (Extended Data Fig. 9), where they are positioned to modulate synaptic signaling BLA principal neuron coding; and, we now show that BLA principal neurons are required for fear retrieval and extinction in a new experiment in which synaptic transmission in glutamatergic BLA neurons was perturbed (see new data tetanus toxin light chain shown in

Extended Data Fig. 9a-c and also pasted below). We discuss this issue and these data in the revised manuscript as follows: ‘Astrocytes are integrated within neuronal networks through synaptic contacts^{26,31,47,48} (including in BLA, Extended Data Fig. 9a,b) and synaptic transmission at glutamatergic BLA neurons is required for both fear memory retrieval and extinction (Extended Data Fig. 10a-g). Thus, one way in which astrocytes could support memory and extinction is by modulating neuronal coding in this region.’

We also point out that, while neuronal coding was not measured in these experiments, it is still pertinent to note that hM3Dq activation did not alter shock-related freezing (Extended Data Fig. 7d) or CS-related freezing when CNO was given prior to (Extended Data Fig. 7e) or immediately after (Extended Data Fig. 7f) fear conditioning or prior to extinction retrieval (Extended Data Fig. 8g). At the very least these negative data show that hM3Dq activation did not have non-specific effects on freezing.

4. As the authors report correctly, astrocyte signaling has been previously implicated in memory-related brain function. The authors argue that they revealed a novel mechanism through which astrocytes modulate threat-related memory and BLA coding. However, the mechanism how astrocytes exert their effects on neuronal function in BLA is not addressed (see also direct vs. indirect effects in Major Point 3) and remains unknown.

Response: Whereas there is a growing literature indicating amygdala-mediated memories and associated behavioral states are underpinned by dynamic coding by neuronal population representations in the BLA. And, as the Referee points out, there is accumulating data from

elegant studies implicating astrocytes in memory-related brain function (e.g., Sun et al. 2024, Chung et al. 2025, Williamson et al. 2025). However, these studies do not show how astrocytes could influence memory via modulation of neuronal coding in behaving animals. Our findings provide novel evidence addressing this outstanding question and provide an important advance in bridging the astrocyte and neuronal literatures by providing the clearest evidence to-date of the effect of astrocytes on memory-related neuronal coding. We emphasize this key question in the Introduction as follows: *‘While these findings support a role for astrocytes in mediating fear, there remains a critical gap in understanding whether astrocytes modulate memory-related neuronal coding and plasticity via population-level neural representations and their behaviorally-mediating circuit outputs.’* And highlight the novel advance provided by our findings in the Conclusion, as follows: *‘The instantiation, retrieval and extinction of memories for previously-experienced threats has traditionally been viewed as exclusively neuronal processes. Prior studies measuring neuronal activity in the BLA have shown that fear states are associated with neural representations in this brain region¹⁻⁷. In parallel, there is a growing body of evidence pointing to an important role of astrocytes in these critical survival functions. The current findings reconcile these two lines of research.’*

To additionally go on to resolve the molecular mechanisms involved would, we feel, be beyond the scope of a single study. Notwithstanding, should the word ‘mechanism’ convey the impression that our neural circuit/systems level study aims to address astrocyte-neuron interactions at the molecular/signaling level, we have removed use of this word in this context in the revised manuscript. We also add discussion of potential molecular mechanisms in the Conclusion, as follows: *‘An important avenue for future work will be elucidating the molecular mechanisms through which astrocytes modulate fear states and accompanying neuronal representations in BLA and other brain regions. There is currently evidence that, on exposure to salient events, neurotransmitters and neuromodulators, including glutamate, adenosine and norepinephrine (NE), engage astrocytes to prime and support neuronal networks^{23,34,36,58-63}. For instance, NE signals through astrocyte-adrenergic receptors to induce Ca² activity, prime astrocytic responses to other neurotransmitters and alter synaptic efficacy, thereby mediating neuronal and behavioral adaptations to repeated unproductive actions and arousing stimuli^{17,23,33,34,64,65}. Thus, NE-driven BLA astrocyte responses to threat are one possible factor driving the neuronal computations underlying the behavioral effects see, though it is likely that other astrocyte-modulating neurotransmitters are also integral to these processes. Further work will be needed to unravel the complex milieu of mechanisms involved²⁵.’*

Minor:

5. In one set of experiments, hM4Di is used to inhibit astrocytes, while other experiments use CalEx. The authors provide a rationale for using CalEx (to avoid stress), but the use of different inhibition methods weakens the comparability of the data, especially considering that the mechanism of hM4Di is unknown.

Response: Our perspective is that because, as the Referee points out, the mechanism of hM4Di and other DREADD effects on astrocytes is poorly understood, using CalEx served as a mechanistically more well-described approach to manipulating astrocytes and thereby reference and complement our more novel DREADD data. We also point out that, unlike many extant studies in the literature, we validate the effects of all these various manipulations on *in vivo* astrocyte Ca^{2+} signaling, including during our key behavioral assays.

6. Do astrocytes follow neuronal activity or vice versa? A latency analysis (even if just as mean CS / US responses in different animals) could shine light on this.

Response: Thanks for directing us to address this question, which we have addressed with the new simultaneous astrocyte-neuron photometry experiment described in response to point #1 above. To reiterate, these new data are now shown in Fig. 1j,k (for E-Ext, and pasted below) and Extended Data Fig. 3, and discussed in the revised manuscript as follows: ‘*Additionally, while we observed similar stage-related Ca^{2+} activity changes in neurons (Extended Data Fig. 3a-h), simultaneous photometry performed in BLA astrocytes and neurons in different hemispheres of the same animals indicated comparatively slower-to-peak, but longer lasting, US and CS-related Ca^{2+} responses in astrocytes (Fig. 1j,k and Extended Data Fig. 3e-h).*’

7. Suppl Fig 1e, heat map: the calcium responses over trials during extinction 1 (Ext 1) and fear renewal (F-Ren) exhibit very similar dynamics, i.e. they decay very quickly with increasing numbers of trials. However, the freezing pattern is remarkably stronger and consistent during F-Ren. If BLA astrocyte calcium events are threat predictors, shouldn't the calcium responses in the heat map also be consistent during fear renewal instead of decaying, given that freezing is stable?

Response: A parsimonious explanation for the similar Ca^{2+} dynamics during Ext 1 and F-Ren, despite differing freezing, is that astrocyte Ca^{2+} activity in other brain regions contributes to freezing during F-Ren (in much the same way that neuronal activity in cortical and hippocampal regions is involved in F-Ren, e.g., <https://pubmed.ncbi.nlm.nih.gov/29403033/>, <https://pubmed.ncbi.nlm.nih.gov/27060752/>, <https://pubmed.ncbi.nlm.nih.gov/27773481/>).

8. Ca^{2+} transients, e.g. Fig. 2f: The difference between the global Ca^{2+} wave and the quantification of Ca^{2+} transients is not quite clear to the reader, yet essential for the conclusions of the paper as the transients are quantified. It would be important to demonstrate a few examples of Ca^{2+} transient in the control and CNO data.

Response: Thanks for the suggestion. In Fig. 2f and 2l (also pasted below), we now show examples of the Ca^{2+} transients in the control and CNO data.

9. In Fig 2c, traces last ca. 30 min. Following an initial Ca^{2+} peak, the signal decays close to baseline. The authors mention that behavioral testing starts after 30 minutes, so it would be important to plot the activity in 2c and 2i beyond 30 min.

Response: Please see Extended Data Fig. S8f (also pasted below) for activity plotted for 2 hours post-CNO. These data show that Ca^{2+} activity remains decayed after Gq manipulation, and Ca^{2+} events remain elevated after Gi manipulation, during the post-CNO period corresponding to when behavioral experiments are performed. The relevant text has been modified to make this clear: ‘*Replicating these effects in BLA, we expressed hM3Dq in BLA astrocytes and used in vivo cyto-GCaMP6f photometry and observed that clozapine-N-oxide (CNO)-injection markedly increased Ca^{2+} activity at ~10 minutes but thereafter decreased and remained lower for at least 2 hours (Fig. 2c and Extended Data Fig. 6a-e, 8f).*’

10. During fear conditioning, it seems that the CS leads to a continuous activation of astrocytes (ED Fig. S3b). During early extinction the CS responses decay during the tone (ED Fig. S3e). Maybe this is a misperception due to different scaling but if not, this difference in the time course of the CS responses during conditioning and early extinction is interesting.

Response: Perhaps the Referee mistook the US response (rightmost trace after the second dashed line in the graph pasted below) in (what is now) ED Fig. S4d as the CS response (which is the leftmost trace); note the labels above the trace and the line demarcating the periods of CS and US presentation. The response to the CS here is much lesser (although still significantly above chance) than the response to the US.

11. Fig. 3i,j vs. Fig. 3m. Inhibition of astrocyte Ca^{2+} signaling leads to a lower dimensional state-space of neuronal CS responses on the population level, yet, the ability to decode decreases. This seems counter-intuitive. Would a lower dimensional representation not lead to higher decodability?

Response: An interesting point. An argument could be made for a negative, positive or no relationship between dimensionality and decodability. On the one hand, a relationship might be expected between a lower dimensional representation and higher decodability to the extent that lower dimensionality reflects encoding of fewer variables and may therefore be ‘simpler’ to decode. On the other hand, with increased dimensionality, neuronal activity can span more state-space and increase the probability that decoders will perform well. Another perspective is that the two variables measure different properties of activity and should not necessarily be expected to be related, in that dimensionality estimates how variance (whether it is related to the to-be-decoded variable or not) is spread across multiple dimensions, whereas decoding essentially considers information in 1 dimension (the dimension containing information about the to-be-decoded variable). In cases where lower dimensionality is evident in the context of impaired behavioral performance, as we see with reduced memory retrieval following disruption of astrocyte calcium signaling, the lower dimensional CS-related information carried by the neuronal population could reflect the impoverishment (and hence poorer decodability) of that information. This is just one interpretation of a complex question and we avoid making strong claims about the meaning of the relationship between the two findings in the manuscript.

12. Fig. 1b,k: The GFAP, S100b staining is not visible. It would be easier for the reader to judge the data if the channels would be split.

Response: The merged-channel images shown in Fig. 1b,k are now also shown unmerged in Extended Data Fig. S4a (also pasted below), thanks.

13. Fig. 1e,g: The horizontal labels for the CS and US periods seem to be misaligned / broken-up.

Response: Apologies for any confusion but the horizontal lines in Fig. 1e,g depict periods during CS and US presentation where Ca^{2+} activity is statistically different from chance, as determined by permutation tests. This has been clarified in the Fig 1 legend (and elsewhere in the manuscript, where relevant) as follows: ‘Horizontal lines above traces in e,g,k denote permutation test-determined significant difference from chance for astrocytes (dark green), neurons (light green), or between astrocytes and neurons (red).’

14. Fig. 1n: ROI 6 appears twice in the ROI map.

Response: This has been corrected, thank you for catching it.

15. Fig. 2e: A correction for multiple comparisons would seem appropriate for the statistical tests on CS blocks.

Response: We have performed a correction for multiple comparisons and accordingly adjusted the significance level for the statistical tests performed on the 10 CS blocks shown in Fig. 2e (and for consistency, Fig. 2k). The key group differences at block 1 remain significant after correction.

16. Fig 2f: The Ca^{2+} traces are reported as Ext. 1 and Ext. 2. However, the behavioral data is measured in 1 session. There seems to be a disconnect.

Response: A single extinction session was used for behavioral experiments, and a longer inter-CS interval (60 seconds, versus 5 seconds) was used for fiber photometry experiments to ensure calcium activity recovered between CS presentations; necessitating splitting the extinction sessions into two because a single long session (~1 hour) would fatigue the mouse and potentially cause technical artifacts such as cable tangling and photobleaching. We show that both the single and double extinction procedures both produce robust extinction; for example, compare Fig. 1d and the controls in Fig. 2c.

17. Fig. 3h: Did the authors perform a statistical test to reach the stated conclusion?

Response: The stated conclusion of '*CNO-injected animals had fewer positively-modulated neurons (Fig. 3h,i)*' (now shown in Fig. 3h) is based on a chi-square test result. Please note that all statistical test results can be found in Supplemental Table S1.

18. Fig. 3f: The CS presentation periods are unclear. Error bars / variance bars are missing.

Response: The period shown is the entire 30-second of CS-presentation period (plus pre-CS 5 second) (the scale incorrectly showed 10 min and has been corrected to 10 sec). These are single neuron/single CS presentation examples – hence no errors bars. The legend has been modified to make these two points clearer as follows: '*Examples of CS-related neuronal Ca²⁺ activity on F-Ret (entire 30-second CS-presentation period shown).*'

19. Fig. 3n: It is currently unclear if the result holds true across animals. A statistical test should be performed to demonstrate that the results are not just driven by 1 mouse.

Response: To clarify, the values shown in Fig. 3n are not from one mouse per group. Rather, they reflect an 'Information per neuron' metric that fits the decoding curves (shown in Fig. 3n, which are generated by randomly subsampling the requisite number of neurons from the entire neuronal pool, and therefore highly unlikely to be driven by a single mouse) using a saturating function to estimate the rate at which decoding accuracy increases with the addition of each neuron. This generates a single value per group computed from the entire imaged population, and hence cannot be statistically compared across groups. For a published example of this analysis, see Figure 6K in Tang et al. 2022 <https://pmc.ncbi.nlm.nih.gov/articles/PMC9608360/>. We have modified the corresponding text as follows: '*Using a saturating function to calculate the rate at which*

decoding accuracy increased with the addition of each neuron⁴⁹, we estimated the CS-related information carried by individual BLA neurons was lower in the CNO (0.20) than Veh (0.31) group.'

20. The text should be revised for typos and clarity, as well as incorrect referencing of figure panels, e.g. lines 206-207, 209-210, 230, 138, 212.

Response: Many thanks for catching these typos. Lines 206-207: 'assess' has been added. Lines 209-210: 'disrupting' changed to 'disrupted.' Line 230: 'that' has been added. Line 138: reference now made to Fig. 2f. Line 212: reference now made to Extended Data Fig. S10d,e.

Referee 2

Overall, this is a novel and impressive study on the role of astrocyte activity in the amygdala on fear memory coding. They report that astrocytic activity tracks fear responses in the amygdala, and correlates with fear conditioning, extinction, and reinstatement. They show that bidirectional modulation of astrocyte function during extinction learning modulates fear extinction learning, and memory, in a dissociable manner. They then show the effect of this astrocytic manipulation on neuronal ensemble activity with miniscope recordings. They found that amygdala astrocyte signaling maintains the neuronal representations that correlates with memory recall, as measured by population decoding analysis. Lastly, through impressive multi-electrode array based recording of amygdala-prelimbic cortex projections, the authors show that astrocytic activity in the amygdala modulate CS-induced responses of the prelimbic region. The study is novel and important. The question is highly relevant to a broad audience. The methods are integrative and appropriate. The experiments are well controlled and the analysis is rigorous. The Figures are well constructed (though in the wrong order, which was confusing....). The study is balanced, very elegant, and concise. The conclusions are very concise, but clear and well justified.

Response: We thank the Referee for recognizing our study as being 'novel and important' and 'highly relevant to a broad audience.'

1. However, I am surprised that the authors have not used their experimental preparation to investigate the role of astrocyte activity in forming memories at the time of training. Indeed, I would also wonder how this process effects the formation of new engrams. Further, it would be interested to test what astrocyte disruption does to existing fear engrams. This could be important

and valuable, because it would test whether the ensemble decoding changes discovered by the authors is in fact functionally important for real engram coding from the perspective of the animal (and not just from the perspective of the investigator).

Response: We agree that the question of the potential role of amygdala astrocyte activity in forming memories - and associated neuronal engrams - at training is an interesting and important one that remains to be fully clarified. We demonstrate that disrupting BLA astrocyte calcium activity during conditioning (via activating hM3Dq prior to conditioning) did not alter freezing during training or subsequent retrieval/extinction (Extended Data Fig. 7e and pasted below). We therefore decided not to follow-up with experiments to examine the potential interplay between astrocytes and neuronal coding during memory formation and instead focused on examining coding during fear retrieval. Nonetheless, there remains scope to approach this question in different and potentially revealing ways using other methods in the future.

2. Lastly, I am not confident that the authors genetic tools are astrocyte specific. This could be better justified and explained. These are just two broad suggestions for improvement and clarification. The authors may have other improvements they want to prioritize or substitute for the above when considering the other Referees' comments.

Response: We thank the Referee once again for the highly positive assessment of our study and the valuable suggestions for improvements. Regarding the viral tools and transgenic mouse line (for 2-photon calcium recordings) used to monitor and manipulate astrocyte activity, these were carefully chosen based on published studies detailing their development and use in other regions (e.g., see Yu et al., 2020 <https://www.nature.com/articles/s41583-020-0264-8>), and in most cases further validated by us for specificity and efficacy in BLA. We show the astrocytic and neuronal expression of these tools throughout the manuscript and additionally examined the effects of

astrocytes chemogenetic manipulation on *ex vivo* electrophysiological neuronal recordings and *in vivo* neuronal fiber photometry experiments. We are therefore confident in the tools we use in this study.

Referee 3

This study by Bukalo and colleagues seeks to deepen our understanding of how astrocytes contribute to the processes of memory formation, extinction, and recall in learned threat associations. The research contains many technically impressive and conceptually intriguing aspects. However, I believe the current manuscript lacks clarity in presenting its findings, and its novelty could be enhanced by addressing some unclear aspects in the results. Below are my main considerations:

Response: We are grateful to the Referee for considering our study to be ‘technically impressive and conceptually intriguing’ and have sought to carefully address the aspects considered unclear with new experimentation, analyses and revisions to the figures and text.

1. Fiberphotometry and Single-Cell Tracking in the BLA: The authors first examine overall astrocytic activity in the basolateral amygdala (BLA) using fiber photometry, finding a pronounced response to the conditioned stimulus (CS) during early extinction and fear renewal phases, which aligns with elevated freezing behaviors. Subsequently, they apply endoscopic methods to analyze single-cell activity in the same region using the same behavioral protocol. However, a significant opportunity is missed here: additional analyses could provide richer insights into BLA neuronal reorganization during these phases. Specifically, longitudinal tracking of individual cells or examining single-cell responses and contributions to the CS would add depth. Instead, the authors primarily confirm the fiber photometry results without further exploiting these data. For instance, in Fig. 1n, numerous cells appear silent—do they have distinct roles in these dynamics? Do they remain inactive at all times, or are they conditionally activated? A closer analysis here could greatly enhance the paper’s novelty by offering more than what is already known about astrocytes' contribution to memory processes.

Response: Thank you – we agree that potential functional heterogeneity across cells is an important and interesting question. Therefore, to address the Referee’s question about longitudinal tracking of individual cells, we have now examined, from our 2P astrocyte data, the CS-responsivity of individual astrocyte events across test-phases (also please see response to

Referee 1's point #2 regarding our new cross-test-stage 1P neuronal imaging data). As the Referee suspected, this revealed subsets of events that become conditionally active (comparing pre-conditioning (Pre-Con) to fear retrieval/early extinction), and other subsets that subsequently either lose or maintain CS-responsivity across extinction (while yet another subset remains silent across test-phases). These new analyses are a valuable addition – thank you for the suggestion! – and are included as Fig. 1q (also pasted below) and discussed as follows: *‘Notably, however, this high-resolution imaging approach was able to show that not all astrocyte events developed conditional-activity from Pre-Con to E-Ext and, moreover, revealed subsets of events exhibiting either decreasing, increasing, sustained, or absent CS-responsivity across extinction (E-Ext-L-Ext) (Fig. 1q, Extended Data Fig. 4j-m). These findings, together with recent evidence of experience-dependent changes within specific astrocyte ensembles in BLA and hippocampus^{26,31,40}, imply heterogeneous subpopulations of astrocytes acquire and/or update memory-relevant information.’*

2. Chemogenetic Manipulation of BLA Astrocytes: The authors use chemogenetics to manipulate BLA astrocytes, but I find it hard to be convinced that CNO exposure inactivates the targeted population within 30 minutes. Furthermore, the chemogenetic effect on fiber photometry data is unclear: there are discrepancies between Z-scores (which need clarification within the methods and the text explanation), area under the curve (AUC), and cumulative calcium transients. This raises questions about whether data interpretation might vary depending on the analysis approach. Does the calcium signal uniformly increase or decrease across these methods? What effect is observed beyond the initial 30 minutes? A later testing time post-CNO injection might yield clearer data, particularly concerning effects on extinction retrieval (E-Ret; see next point).

Response: First, regarding the CNO time course. Our apologies if this was not clearer, but behavioral testing and associated Ca^{2+} imaging/photometry/single unit recordings were conducted beginning *30 minutes after CNO injection*. Hopefully, this assuages the Referee's concern about testing effects more immediately after CNO. To make this clearer in the manuscript, we have modified the experimental schematics in Fig. 2d, 2j to state 'CNO 30 min pre-Extinction.' Also note that we show the time course of CNO effects in Gq (and Gi) expressing animals are shown over 30 minutes after CNO injection in Fig. 2c and 2i, and over 120 minutes in Extended Data Fig. S8d (also see our response to point #9 of Referee 1). These latter data (also pasted below) show that calcium transients begin to change around 10-20 minutes after CNO is injected.

Additionally, to further address the Referee's question, we have performed 2 new *in vivo* fiber photometry experiments measuring CS-related BLA astrocyte Ca^{2+} activity in Gq-expressing animals either 3 minutes or 10 minutes after CNO injection. These experiments show that astrocyte conditioned CS-related Ca^{2+} activity was unaffected 3 minutes after CNO but, was already dampened 10 minutes after CNO. Hence, CNO begins to exert its effects on astrocyte Ca^{2+} quite rapidly. These new data are now presented in Extended Data Fig. 6a-d (pertinent graphs also pasted below) and we note the time course controls in the revised manuscript text as follows: '*Replicating these effects in BLA, we expressed hM3Dq in BLA astrocytes and used in vivo cyto-GCaMP6f photometry and observed that clozapine-N-oxide (CNO)-injection markedly increased Ca^{2+} activity within ~10 minutes but thereafter decreased and remained lower for at least 2 hours (Fig. 2c and Extended Data Fig. 6a-e, 8f).*'

CS-related activity tested 3 minutes after CNO injection:

CS-related activity tested 10 minutes after CNO injection:

Second, regarding the Z-scores calculation, this is clarified in the methods and Figures legend text, e.g., for Fig. 1 as follows: ‘*Fluorescence values z-scored to the 5-second pre-CS period (photometry) or entire recording session (2-photon imaging)*’. As measured by calcium transients, the calcium signal uniformly increases with hM4D actuation and decreases with hM3D actuation (as noted in the previous point, both changes begin ~10 minutes after CNO injection. The AUC values (i.e., the summed calcium activity during a given period) are lower with both manipulations. The differential effects on these measures is likely due to their differing signaling effects: hM3D actuation (which most of our experiments utilize) produces a potent release of calcium that rapidly depletes calcium from endoplasmic reticulum stores and thereby results in reduced calcium activity in our behaviorally-relevant timepoints, whereas hM4D actuation causes an increase in calcium transient frequency that at the same time appears to limit large fluctuations in calcium activity, resulting in an attenuated over AUC response within a given time window. These putative actions in BLA are broadly consistent with data in other brain regions (e.g., <https://pmc.ncbi.nlm.nih.gov/articles/PMC7968927/>, <https://pubmed.ncbi.nlm.nih.gov/30801845/>, <https://pubmed.ncbi.nlm.nih.gov/31031006/>). This aside, the most pertinent result for the

interpretation of our behavioral data is that, during fear retrieval/extinction, cumulative calcium events are depressed by hM3D actuation (Fig. 2f and also pasted below, leftmost panel) and hM4D amplified by hM4D actuation (Fig. 2l and also pasted below, rightmost panel).

3. Unresolved E-Ret Dynamics: The findings related to extinction retrieval (E-Ret) remain ambiguous. Testing E-Ret with specific chemogenetic or optogenetic manipulations could help clarify astrocyte involvement at this time point. Additionally, including E-Ret data in calcium recordings (e.g., Fig. 2f) might improve understanding of astrocytic roles during retrieval.

Response: Regarding the question about E-Ret data in calcium recordings, please see the following point. We have followed the Referee’s suggestion to test E-Ret with specific chemogenetic manipulations by conducting new experiments in which hM3Dq or hM4Di was expressed in BLA astrocytes and CNO administered *prior to E-Ret (only)*. Interestingly, the results of these experiments show that neither manipulation affected freezing on E-Ret. These data suggest that BLA astrocytes are not necessary for the retrieval of extinction memory, in contrast to their importance to extinction memory formation. These valuable new findings are added as Extended Data Fig. S8g,h (also pasted below) and discussed as follows: ‘*hM4Di-actuation during fear retrieval/extinction training also increased freezing on (CNO-free) E-Ret – indicative of a deficit in extinction memory formation. Of note, the hM3Dq-actuation experiment above produced a similar extinction memory deficit (Fig. 2e), whereas the two chemogenetic-manipulations producing opposite effects on fear retrieval and neither manipulation affecting extinction memory when CNO was given prior to E-Ret only (Extended Data Fig. S8g,h). These observations indicate that extinction is sensitive to perturbations – whether increases or decreases – in BLA astrocyte activity and thereby interfere with plastic adaptations in activity supporting memory formation extinction.*’

4. Differential Effects of Chemogenetic Activation: Chemogenetic activation appears to exert opposite effects on extinction related to Chemo-inactivation. However, both manipulations lead to an increase in freezing behavior during E-Ret. This intriguing result, however, would benefit from additional analyses to explore its underlying mechanisms. Calcium recordings, once again, could enhance interpretability.

Response: We agree that the finding that both chemogenetic manipulations lead to an increase in freezing during E-Ret, despite producing opposite effects on fear retrieval/early extinction, is an intriguing result. Calcium recordings were performed during E-Ret and are shown in Extended Data Fig. S7b and S8b (also pasted below). These data indicate that hM3Dq (below left, yellow line=hM3Dq, black line=control) and hM4Di (below right, blue line=hM4Di, black line=control) during extinction both lead to attenuated astrocyte activity on E-Ret, reflecting the similarity of the behavioral deficits in two experiments. A parsimonious interpretation for these effects relates to the Referee's point below; that is that both manipulations disturb the dynamic range of activity in astrocytes (as now shown in Fig. 1q,r) and associated neuronal plasticity (as now shown in Fig. 3p-r) required for extinction memory formation – resulting in a deficit on extinction retrieval after either manipulation. We explicitly discuss this possibility in the revised manuscript text as follows: *'Pre-Ext hM4Di-actuation also increased freezing during (CNO-free) E-Ret – indicative of a deficit in extinction memory formation – and attenuated CS-related Ca²⁺ activity during this test-stage. This latter effect is notable given hM3Dq-actuation produced a similar extinction deficit and blunted CS-related Ca²⁺ response on E-Ret (see Fig. 2e and Extended Data Fig. 7b), even though these two manipulations had opposite effects on fear retrieval and neither affected extinction memory when CNO was only given prior to E-Ret (Extended Data Fig. 8g,h). The convergence of these effects on extinction retrieval, suggests that extinction is highly sensitive to perturbations – whether increases or decreases – in astrocyte Ca²⁺ activity and, by extension, imply BLA astrocytes have an important role in supporting the plastic adaptations underlying extinction memory formation.'*

5. Calcium Recordings Across Extinction Phases: The calcium data from chemogenetically activated astrocytes reveal a generally flat activation profile across extinction sessions, lacking the modulation or decline observed in vehicle-treated animals. It appears that activity decreases during the initial extinction sessions (Ext 1 and 2) compared to vehicle-treated animals. This suggests that hM4D treatment dampens astrocytic activity rather than simply increasing it, as the authors suggest, which may impair normal extinction dynamics by preventing adaptive modulation of activity. What insights do we gain from this manipulation? Additionally, to better understand the fiber photometry responses and differences from the vehicle group, I recommend showing the zero-value calibration for Z-scores on the heatmap in Fig. 2l.

Response: As requested, we now show the zero-value calibration for Z-scores on the heatmap in Fig. 2l (and in Fig. 2f, for consistency in within the same figure). As noted in the previous point, we agree with the possibility of impaired adaptation of activity contributing to impaired extinction. We maintain that this rigidity occurs despite there being an increase in astrocyte events during the initial extinction trials (measuring fear retrieval/early extinction) (as shown in Extended Data Fig. S8c and also pasted below).

6. Analysis of Treated vs. Untreated Mice: The authors then examine differences between vehicle- and CNO-hM3D-treated mice during fear conditioning and retrieval, employing exciting and persuasive analysis techniques. However, interpreting the significance of comparing baseline astrocyte activity with chemogenetically manipulated activity is challenging, particularly given the widespread activity reduction shown in Fig. 2f. As expected with this reduced activity, the analysis reveals lower dimensionality, fewer units responding to the CS, and overall decreased informational content. This comparison, however, may fall short in illuminating the specific contributions of BLA astrocytes to memory processes. Instead, comparing single-cell activation patterns, dimensionality, and decoding capabilities across stages—such as early extinction, late extinction, retrieval, and renewal—might yield more meaningful insights.

Response: To clarify, the examination of ‘differences between vehicle- and CNO-hM3D-treated mice during fear conditioning and retrieval’ is an analysis of neuronal activity, whereas ‘the widespread activity reduction shown in Fig. 2f’ is an analysis of astrocyte activity. Therefore, we do not necessarily expect the ‘lower dimensionality, fewer units responding to the CS, and overall decreased informational content [in neurons]’ to follow from reduced astrocyte activity. Indeed, this was a major question we originally posed (i.e., does loss of astrocyte activity affect neuronal coding?) and answered by showing that neuronal coding during memory retrieval was indeed disrupted when BLA astrocytes were manipulated. Additionally, on the basis of this and Referee 1’s request to examine changes/plasticity across stages, we now show new data demonstrating impaired neuronal coding during extinction retrieval in CNO-astrocyte-hM3D-treated mice (please see response to Referee 1, point #2 above). We thank both Referees for guiding us to extend the study’s findings with these valuable additional results.

7. Correlation of Freezing Levels with Decoder Coefficients: In Fig. 4a-c, the authors suggest a correlation between freezing and decoder coefficients driven by vehicle data. To validate this, they should display separate graphs and statistical analyses for vehicle-only data.

Response: As requested, we now separate the two groups and associated statistical analyses for the vehicle and CNO data in what is now Fig. 3o (also pasted below).

8. Notable Findings Using Optogenetics and Multi-Electrode Recordings: The most compelling findings involve optogenetic and multi-electrode recordings, which demonstrate how BLA astrocytes influence projections affecting prelimbic (PL) neuronal dynamics. These results highlight novel insights into how BLA contributes to PL recruitment during extinction and how optogenetic astrocyte inactivation impacts this process. This approach holds substantial promise for further studies on the memory-related dynamics of this pathway.

Response: Thank you. We agree that the novel insights provided by these data add appreciably to the study's findings.

9. In summary, this study shows considerable potential but requires additional data analysis to clarify some unresolved dynamics (e.g., E-Ret) and to integrate single-cell identification. Additionally, BLA-PL dynamics analysis remains preliminary and could benefit from further investigation. I am enthusiastic about this research but believe the current version is still preliminary and requires further analysis and interpretation.

Response: We appreciate the Referee's enthusiasm for the work and hope we have addressed valuable requests for further analysis and interpretation to improve the manuscript.

Minor points:

10. Fig. 2f: If this uses the same protocol as Fig. 2a, consider omitting the redundant drawing to avoid confusion.

Response: We have removed Fig. 2f, as suggested.

11. Fig. 3n: Can the authors statistically assess group differences and variability?

Response: To clarify, the Information per neuron metric fits the decoding curves shown in Fig. 3n using a saturating function to estimate the rate at which decoding accuracy increases with the addition of each neuron. This generates a single value per group computed from the entire imaged population, and hence cannot be statistically compared across groups. For a published example of this analysis, see Figure 6K in Tang et al. 2022 <https://pmc.ncbi.nlm.nih.gov/articles/PMC9608360/>. We have modified the corresponding text as follows: *‘Using a saturating function to calculate the rate at which decoding accuracy increased with the addition of each neuron⁴⁹, we estimated the CS-related information carried by individual BLA neurons was lower in the CNO (0.20) than Veh (0.31) group’* to clarify this metric. Please also see our response to Referee 1’s point #19 above.

12. Fig. 3a: The term “activation” could be misleading; “manipulation,” as used in Fig. 2, may be clearer.

Response: We agree ‘activation’ could be misleading and now use the term conventionally used in reference to DREADDs -‘actuation.’

Sincerely,

Andrew Holmes and Olena Bukalo

Andrew Holmes, PhD
Chief, Laboratory of Behavioral and Genomic Neuroscience
National Institute on Alcohol Abuse and Alcoholism
National Institutes of Health
5625 Fishers Lane, 2N09, Rockville MD 20592
Telephone: +1 301 402 3519

October 2, 2025

[REDACTION]

Based on the very positive evaluation of our revised submission, we are submitting a newly revised version of our manuscript entitled ‘*Astrocytes enable amygdala neural representations supporting memory*,’ by Bukalo et al. The newly revised manuscript continues to be improved as a result of the review process and we would like to thank you and the Referees for the positive assessment of the work throughout. We have addressed each of the remaining comments and modified the text – indicated by tracked changes – as detailed below.

Referee 1

The authors have addressed major issues. A few minor issues remain.

Response: We appreciate the Referee’s valuable remaining thoughts and suggestions.

1. In general: It would be preferable if the authors would choose data graphing methods that reveal the distribution of the data instead of bar plots.

Response: To reveal the distribution of the data, the individual data points are now overlaid on the bar plots and elsewhere (e.g., Fig. 3i) we use violin plots to better depict distribution.

2. Fig. 1d. Did a post-hoc test reveal significant differences between E-Ext., L-Ext, E-Ret?

Response: Yes, *post hoc* tests (conducted after repeated measures one-way ANOVA showed a significant effect of test-phase) indicated significant differences between E-Ext and L-Ext and E-Ext and E-Ret. The significant differences are stated in the text as follows: ‘*CS-elicited freezing was high on early extinction (E-Ext, first extinction trial-block), i.e., during fear memory retrieval, then decreased significantly by late extinction (L-Ext, final extinction trial-block) and retrieval (E-Ret), before returning to pre-extinction levels on retesting in the conditioning context (fear renewal, F-Ren) (Fig. 1d),*’ and the figure legend as follows: ‘*d, Freezing across testing-phases (1-way repeated measures ANOVA and Šídák’s post hoc tests, *P<0.0001 versus first extinction trial-block, n=17 mice).*’ The statistical results are also provided in Supplementary Table 1 and the Fig. 1d now includes asterisks over L-Ext and E-Ret (please see below).

3. Fig. 1f. Label ambiguous. Does the star indicate significant difference between CS and shock? I assume that the CS and US are always recorded in the same animal. Would this not warrant a paired test design?

Response: Yes, the star indicates a significant difference between CS and shock and has been reanalyzed using a paired t-test given they were recorded in the same animals. The figure legend has been modified as follows: ‘*f, AUC shock-related Ca²⁺ activity during F-Con (2-tailed paired t-test versus CS, *P<0.001, n=17 mice).*’

4. Fig. 1g. Graphing issues of the green CS bar. It appears discontinuous. Similar general issue in several other figures.

Response: The green horizontal bar in Fig. 1g does not indicate the CS presentation, but rather timepoints significantly different from chance (analyzed via permutation test). This is described in the legend as follows: ‘*Horizontal lines above traces in e,g,k denote permutation test-*

determined significant difference from chance for astrocytes (dark green), neurons (light green), or between astrocytes and neurons (red).’

5. Fig. 1i. Vertical and horizontal error bars are missing on the individual summary data points.

Response: We have added horizontal and vertical error bars (SEM) to the individual summary data points in Fig. 1i (please see below).

6. Fig. 1k. The stack bar design deviates from the general bar design of the other figures and is harder to read.

Response: We have replaced the stacked bar with an interleaved bar plot to maintain consistency with the other figures and improve readability (please see below).

7. Fig. 1q. It is not clear how the different plasticity types were determined (minimum threshold for change magnitude, statistical test?).

Response: The determination of plasticity types has been added to the Methods, as follows: ‘CS modulated neurons were statistically defined based on the proportion of null values greater than

the observed value ($P < 0.05$) on a given test stage. These results were used to segregate cross test stage co-registered neurons into subsets exhibiting increasing, decreasing, sustained, or consistent non-responsivity to the CS as a function of fear conditioning (Pre-Con to F-Ret).’

8. Fig. 2. It is not immediately clear to the reader how the control was setup. Are these mCherry-only expressing mice or was only GCaMP6f expressed in these animals?

Response: To clarify the control setup for the reader, the Fig. 2 legend states: ‘Controls in *e,k* expressed mCherry and in *c,f,i,l* expressed mCherry+*GfaABC1D* GCaMP6f.’

9. Fig. 3a. The microscope is not mentioned in the methods. An experimental timeline similar to 2d would help the reader to understand the experimental design.

Response: The microscope is mentioned in the methods as follows: ‘BLA neurons were imaged using a miniature microscope (*nVistaTM 3.0*, Inscopix) and *IDAS HD* software (*Inscopix*) at a frame rate of 20 Hz with an LED power of 10–60% (0.9–1.7 mW at the objective, 475 nm), analogue gain 1, 1080 × 1080 pixels.’ An experimental timeline similar to 2d has been added to Fig. 3c and p (see below).

10. Fig. 3b. Is the GCaMP7f image a miniature microscope image or an ex vivo confocal image. This should be clearly labelled in the figure.

Response: The figure legend has been modified as follows: ‘Ex vivo confocal image of *hM3Dq*-mCherry expression in astrocytes (*GFAP*) and neurons (*NeuN*).’

11. Fig. 3b. How many neurons were imaged per mouse / day?

Response: The corresponding legend (now Fig. 3e) states: ‘*Cross-stage percent change of conditionally CS-excited, inhibited, sustained, and non-responsive neurons (Chi-squared tests, *P<0.05, Veh n=197 neurons/8 mice, CNO n=187 neurons/7 mice).*’

12. Fig. 3c,f,r: Unless I misunderstood the methods, the baseline of the traces should average around 0. However, this doesn’t seem to be the case in the example traces.

Response: The baseline of these traces does not average around 0 because the 1P imaging data were normalized to the entire imaged session rather than the 5-second pre-CS period depicted (in accord with precedent from prior publications). This is stated in the Methods as follows: ‘*To compute CS-related Ca²⁺ responses, time-normalized AUC fluorescence values (z-scored to the entire session) were calculated for each stimulus presentation,*’ and the end of the Fig. 3 legend as follows: ‘*Fluorescence values z-scored to the entire recording period.*’

13. Fig. ED 11,a,f: actuation vs. activation.

Response: We have changed this to ‘actuation’ (please see below) for consistency with the other figures and the text – thanks for catching it.

14. Fig. 3n, comment to Minor Point 19. Using pseudo-populations seems unnecessary given that the authors should be able to record from large numbers of BLA neurons in individual mice. The analysis would benefit from a demonstration of the robustness of the results within animals.

Response: We acknowledge the point but believe the statistical approach we took – i.e., taking the mean of 50 decoder models, each ran using neurons sampled randomly from the entire population – helps ensure the results are robust across animals. We have added this aspect of the analysis to the Fig. 3n legend to ensure it is transparent to the reader, as follows: ‘*Decoding CS-presentation as a function of neuronal ensemble size (data at each ensemble size are the average decoder performance obtained after randomly selecting neurons subsamples 50x) (Veh n=7 mice, CNO n=7 mice).*’

15. Fig. 3o. The analysis balances for the number of animals. Was the number of neurons also balanced to train the pseudo-population decoders on the same number of neurons pooled from all mice? In general, I would propose to test the decoder accuracy / single neuron coefficients on an animal-by-animal basis and relate the decoding results of each mouse to its behavior.

Response: We appreciate the suggestion to consider this alternate analytical approach (i.e., testing the decoder in an animal-wise manner), but have elected to retain the existing approach to be consistent with the other decoding approaches used (e.g., decoding by population size). The number of neurons were approximated, although not exactly matched given we found (not shown) that exact matching did not change the decoding results and so used the entire population available in the 7 mice/group to avoid arbitrarily removing 13 neurons from the Veh group. The number of animals and neurons is stated in the Fig. 3o legend as follows: ‘*Relationship (Pearson’s correlation) between CS-related freezing levels on F-Ret and mouse-averaged decoder coefficient values (Veh n=214 neurons/n=7 mice, CNO n=227 neurons/n=7 mice).*’ And in the Methods as follows: ‘*CS-presentation was decoded from neuronal activity during F-Ret as described under Decoding CS as a function of population-size, after approximating the number of neurons and animals per treatment group (Veh n=214 neurons/n=7 mice, CNO n=227 neurons/n=7 mice). The resultant decoder coefficient values for each neuron were averaged by mouse and the mouse-averaged values correlated with the mouse’s corresponding F-Ret CS-related freezing level.*’

16. Fig. 4h. “Moreover, the degree to which CS-responsivity on E-Ext was diminished by CalEx correlated with the reduction in CS-related freezing (Fig. 4h).” It is unclear how the figure panel 4h supports this statement.

Response: We have modified the text as follows: ‘*Additionally, higher freezing-related activity in BLA-PL, relative to non-tagged, neurons was absent in CalEx-expressing animals (Fig. 4h).*’

Referee 2

1. I have reviewed manuscript with fresh eyes, including carefully going through the other two Reviewers' comments. Overall, there can be no doubt that the authors have substantially improved their body of work with new experiments, new analyses, and improvements to the manuscript text. This effort is more than what is more than sufficient for justifying publication in a high impact journal. Many of these experiments represent incremental improvements, but are valuable nonetheless. Some of the experiments are quite substantial in impact, owing partly to the suggestions of the other Reviewers' comments (but not mine). I believe the core messages from the initial submission were sufficiently novel and robust enough to be published in a high impact journal, but the new additions give the manuscript and extra edge. However, on an aesthetic level I believe the original arrangement of the main Figures was better, and far more easy to read and comprehend. I believe this matters to scientific dissemination, and is certainly more important than the "we did a lot of work in review" maxim. I think that some of the content placement makes sense in review, but less to a reader of the published version. However, this is for the Editors and Authors to decide.

Response: We have retained the revised arrangement of the main Figures (with the exception of Fig. 1r and q being swapped with one another), given we feel that Fig. 3 and 4 make most sense as separate figures, as they each use their own approach (1P imaging versus neuronal recordings).

2. The Extended Figures are only stored as single files rather than in the Supplementary Information, which was a little irritating in review. The authors have satisfactorily answered both of my major points during the first round of review. I have no further comments there. I emphasize that I am not working in astrocyte biology and I have no particular stance in this sub-field, but I believe the authors have done an excellent job and I don't think this manuscript should be restrained from publication any further. However, I would like to see a couple of sentences in the Discussion on how engram specificity might be modulated by astrocytic activity.

Response: We refer to this possibility in the Discussion, within the constraints of the text word limits, as follows: ‘In parallel, there is growing evidence of astrocytic contributions to BLA-

mediated fear, e.g., via fear memory-related astrocytic engrams³¹ that may in turn facilitate the selection of specific memory-encoding neuronal engrams⁵⁵.

3. Reading through the other Reviewers' comments, here are my further comments. Reviewer 1: Reviewer 1 Point 1 is an fair parametric criticism. The authors have engaged with it extensively through these two experiments. I particularly admire the bihemispheric approach, and I think this makes a strong impact on the reader. Reviewer 1 Point 2 seems to be the most valuable criticism of the entire review process. The authors engage with this point extremely well, thereby also somewhat nullifying one of my own comments. The authors show that astrocytic activity is necessary for the learning-induced CS-specificity of BLA neurons. The new experiments in Fig. 3p-r are very welcome to this effect, as they show clear and robust evidence that astrocytic activity is crucial for extinction learning via BLA ensemble responsiveness. Fig. 3r reads a bit like cherry picking, but I don't think that is a major problem here. Reviewer 1 Point 3 is an important scientific point, and the authors go to a large effort to engage with it. Indeed, some crucial new experiments have been added. I believe the TeNT experiment is very important. However, I also think that one could argue that these synaptic inputs are simply necessary for indirect effects that behavioral changes could have on ensemble plasticity. Nevertheless, I emphasize that there is no definitive experiment that can fully answer this criticism, and indeed the criticism itself assumes a scenario that while parametrically possible, does not have a plausible mechanistic basis itself. In other words, the authors have a stronger theoretical basis for their interpretation than the Reviewer, and indeed I would be very surprised if differences in freezing behavior would cause suggest a robust plasticity effect. I think the solution here is for the authors to simply state this caveat in the Discussion.

Response: We agree that the possibility astrocytic effects on neuronal coding are fully explained by effects on freezing is further mitigated by the additional experiments and have not addressed it further in the revised Discussion.

Reviewer 1 Point 4 is also important. It is difficult to resolve these issues in such a concise publishing format, but the authors' changes make good sense. I believe the authors have identified a strong 'mechanism' at an ensemble level. There may well be enabling molecular mechanisms at the level of astrocyte/neuron interactions, but that is a different question. Further, it would be inappropriate to investigate a molecular mechanism of this phenomenon in tandem with such detailed and frontier analysis of BLA neuronal activity and coding. I believe it makes much more

sense for a study that is already highly integrative (in terms of methods) to focus on one crucial level of analysis to explain behavior. Reviewer 1 Minor Points are all well managed, and I think Extended Data Fig. S8f is particularly important. Reviewer 3: Reviewer 3 Point 1 is well addressed by the 2-P analysis and this enhances the manuscript. Reviewer 3 Point 2 is clearly in part a communication issue, but the new experiments are welcome. The time courses of CNO effect are, in my opinion, very valuable. The authors explanations for their FP analysis is clear and reasonable. Reviewer 3 Point 3 is a fair criticism, and the authors respond with a clear argument based on new and previously existing data. The lack of acute effect on extinction memory persistence is clear. However, if the authors wanted to push this further they could of course look at reinstatement, but I think is well outside of the current scope. Reviewer 3 Point 4 is perhaps expecting more symmetry than one can reasonably expect from biological systems. However, the Reviewer is fair to point it out. The authors experiments and interpretation adequately deal with this criticism overall. I think the authors deserve credit for showing the behavioral effects as they are, and not trying to force them to fit a rigid structure. Reviewer 3 Point 5 This is an insightful criticism, and a point of caution that I missed. But the authors response and analysis is a clear counter to this view. Reviewer 3 Point 6 There seems to be have been a communication issue here, but the authors' response is more than appropriate, and I believe sufficient. Reviewer 3 Point 7 Minor but valuable addition to the manuscript. Reviewer 3 Point 8 Fully agree. Reviewer 3 Point 9 I agree, and the authors have substantially improved their thesis. Reviewer 3 Minor Points are all well managed.

Response: We appreciate the Referee's very careful and considered assessment of the revision and our responses to the first round of reviews.

Referee 3

The authors have adequately addressed all of my previous concerns. They have improved the manuscript significantly through the addition of new experiments and analyses, as well as by providing clear and satisfactory explanations where necessary. I am satisfied with the revisions and find the current version of the manuscript suitable for publication.

Response: We thank the Referee for the valuable comments and suggestions made during the course of the review.

Sincerely,

Andrew Holmes and Olena Bukalo